# MIXTURE OF WEAK & STRONG EXPERTS ON GRAPHS

**Hanqing Zeng** *
Meta AI
zengh@meta.com

**Hanjia Lyu** *
University of Rochester
hlyu5@ur.rochester.edu

**Diyi Hu**
University of Southern California
diyihu@usc.edu

**Yinglong Xia**
Meta AI
yxia@meta.com

**Jiebo Luo**
University of Rochester
jluo@cs.rochester.edu

## ABSTRACT

Realistic graphs contain both (1) rich self-features of nodes and (2) informative structures of neighborhoods, jointly handled by a Graph Neural Network (GNN) in the typical setup. We propose to decouple the two modalities by Mixture of weak and strong experts (Mowst), where the weak expert is a light-weight Multi-layer Perceptron (MLP), and the strong expert is an off-the-shelf GNN. To adapt the experts' collaboration to different target nodes, we propose a "confidence" mechanism based on the dispersion of the weak expert's prediction logits. The strong expert is conditionally activated in the low-confidence region when either the node's classification relies on neighborhood information, or the weak expert has low model quality. We reveal interesting training dynamics by analyzing the influence of the confidence function on loss: our training algorithm encourages the specialization of each expert by effectively generating soft splitting of the graph. In addition, our "confidence" design imposes a desirable bias toward the strong expert to benefit from GNN's better generalization capability. Mowst is easy to optimize and achieves strong expressive power, with a computation cost comparable to a single GNN. Empirically, Mowst on 4 backbone GNN architectures show significant accuracy improvement on 6 standard node classification benchmarks, including both homophilous and heterophilous graphs (https://github.com/facebookresearch/mowst-gnn).

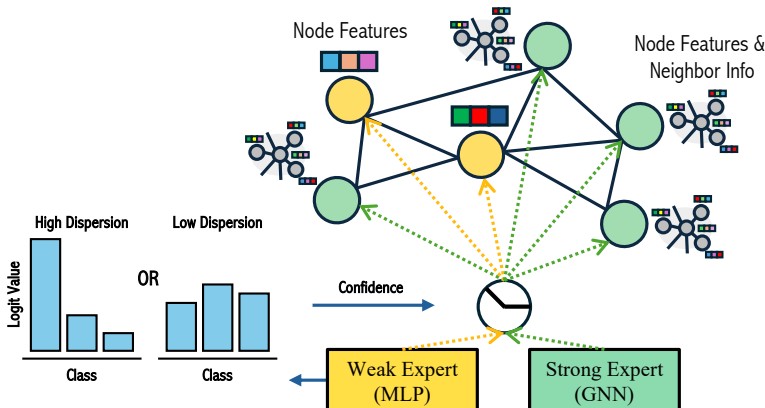

Figure 1: Design overview of Mowst. The full system is composed of a weak expert, a strong expert, and a gating module. Diverse collaboration behaviors between the weak & strong experts emerge as a result of the gating module's coordination. The gating function, which can be either manually defined or automatically learned (via an additional compact MLP), calculates a confidence score based on the dispersion of *only* the weak expert's prediction logits. The confidence score varies across different target nodes depending on the experts' relative strength on the local graph region. The score also directly controls how each expert's own logits are combined into the system's final prediction.

---

*Equal contribution

# 1 INTRODUCTION

A main challenge in graph learning lies in the data complexity. Realistic graphs contain non-homogeneous patterns such that different parts of the graph may exhibit different characteristics. For instance, locally homophilous and locally heterophilous regions may co-exist in one graph (Zhu et al., 2021); depending on local connectivity, graph signals may be mixed in diverse ways quantified by node-level assortativity (Suresh et al., 2021); the number of graph convolution iterations should be adjusted based on the topology of the neighborhood surrounding each target node (Zhang et al., 2021). However, many widely-used GNNs have a fundamental limitation since they are designed based on *global* properties of the graph. For instance, GCN (Kipf & Welling, 2016) and SGC (Wu et al., 2019) perform signal smoothing using the full-graph Laplacian; GIN (Xu et al., 2019) simulates $k$-hop subgraph isomorphism test with the same $k$ on all target nodes; GraphSAGE (Hamilton et al., 2017) and GAT (Veličković et al., 2018) aggregates features from $k$-hop neighbors, again with a global $k$. There is large potential to improve GNN's capacity by diversified treatments on a per-node basis.

Model capacity can be enhanced in several ways. One solution is to develop more advanced layer architectures for a single GNN (Brody et al., 2022; Zheng et al., 2023; Joshi et al., 2023), with the aim of enabling the model to automatically adapt to the unique characteristics of different target nodes. The other way is to incorporate existing GNN models into a Mixture-of-Experts (MoE) system (Jacobs et al., 1991; Hu et al., 2022), considering that MoE has effectively improved model capacity in many domains (Masoudnia & Ebrahimpour, 2014; Du et al., 2022; Lepikhin et al., 2021). In this study, we follow the MoE design philosophy, but take a step back to mix a simple Multi-Layer Perceptron (MLP) with an off-the-shelf GNN – an intentionally imbalanced combination unseen in traditional MoE. The main motivation is that MLP and GNN models can specialize to address the two most fundamental modalities in graphs: the feature of a node itself, and the structure of its neighborhood. The MLP, though much weaker than the GNN, can play an important role in various cases. For instance, in homophilous regions where nodes features are similar, leveraging an MLP to focus on the rich features of individual nodes may be more effective than aggregating neighborhood features through a GNN layer. Conversely, in highly heterophilous regions, message passing could introduce noise, potentially causing more harm than good (Zhu et al., 2020a). The MLP expert can help "clean up" the dataset (§2.4) for the GNN, enabling the strong expert to focus on the more complicated nodes whose neighborhood structure provides useful information for the learning task.

An additional advantage of our "weak-strong" duo is that incorporating a lightweight and easily optimized MLP helps mitigate the issues of computation costs and optimization difficulties commonly associated with traditional GNN or MoE systems. The lack of recursive neighborhood aggregation in an MLP not only makes its computation orders of magnitude cheaper than a GNN (Wu et al., 2019; Zhang et al., 2022) (even when optimized by sampling techniques (Zeng et al., 2021; Shi et al., 2023)), but it also completely avoids issues such as oversmoothing (Chen et al., 2020a; Li et al., 2018) and oversquashing (Topping et al., 2022; Alon & Yahav, 2021). While the traditional MoE approaches can partially address the computation challenges by activating a subset of expert through various gating modules (Shazeer et al., 2017; Nie et al., 2022), *sparse* gating may further complicate optimization due to the introduction of discontinuities and intentional noise (Fedus et al., 2022).

**Contributions.** We propose Mixture of weak and strong experts (`Mowst`) on graphs, where the MLP and GNN experts specialize in the feature and structure modalities. To encourage collaboration without complicated gating, we propose a "confidence" mechanism to control the contribution of each expert on a per-node basis (Figure 1). Unlike recent MoE designs where experts are fused in each layer (Du et al., 2022; Lepikhin et al., 2021), our experts execute independently through their last layer and are then mixed based on a confidence score calculated by the dispersion of the MLP's prediction logits. Our mixing mechanism has the following properties. First, since the confidence score solely depends on the MLP's output, the system is inherently biased. This means that under the same training loss, the predictions of the GNN are more likely to be accepted than those of the MLP (§2.3). Such bias is desirable due to GNN's better generalization capability (Yang et al., 2023), and the extent of bias can be learned via the confidence function. Second, our model-level (rather than layer-level) mixture simplifies the optimization and enhances explainability. Through theoretical analysis of the optimization behavior (§2.3), we uncover various interesting collaboration modes between the two experts (§2.3). In the *specialization* mode, the system dynamically creates a soft splitting of the graph nodes based on not only the nodes' characteristics but also the two experts' relative model quality. After splitting, the experts each specialize on the nodes they own. In the

*denoising* mode, the GNN expert dominates on almost all nodes after convergence. The MLP overfits a small set of noisy nodes, effectively removing them from the GNN's training set and thus enabling the GNN to further fine-tune. The above advantages of `Mowst`, together with the theoretically high expressive power, come at the cost of minor computation overhead. In experiments, we extensively evaluate `Mowst` on 4 types of GNN experts and 6 standard benchmarks covering both homophilous and heterophilous graphs. We show consistent accuracy improvements over state-of-the-art baselines.

## 2 `Mowst`

Our discussion mainly focuses on the 2-expert `Mowst` while §2.7 shows the many-expert generalization. The key challenge is to design the mixture module considering the subtleties in the interactions between the imbalanced experts. On the one hand, the weak expert should be *cautiously* activated to avoid accuracy degradation. On the other hand, for nodes that can be truly mastered by the MLP, the weak expert should meaningfully contribute rather than being overshadowed by its stronger counterpart. §2.1 and §2.2 present the overall model. §2.3 and §2.4 analyze the collaboration behaviors. §2.5 discusses a variant of `Mowst`. §2.6 analyzes the expressive power and computation cost.

**Notations.** Let $\mathcal{G}(\mathcal{V}, \mathcal{E}, \boldsymbol{X})$ be a graph with node set $\mathcal{V}$ and edge set $\mathcal{E}$. Each node $v$ has a length-$f$ raw feature vector $\boldsymbol{x}_v$. The node features can be stacked as $\boldsymbol{X} \in \mathbb{R}^{|\mathcal{V}| \times f}$. Denote $v$'s ground truth label as $\boldsymbol{y}_v \in \mathbb{R}^n$, where $n$ is the number of classes. Denote the model prediction as $\boldsymbol{p}_v \in \mathbb{R}^n$.

### 2.1 `Mowst`

---

**Algorithm 1** `Mowst` inference

**Input:** $\mathcal{G}(\mathcal{V}, \mathcal{E}, \boldsymbol{X})$; target node $v$
**Output:** prediction of $v$
Run the trained MLP expert on $v$
Get prediction $\boldsymbol{p}_v$ & confidence $C(\boldsymbol{p}_v) \in [0, 1]$
**if** random number $q \in [0, 1]$ has $q < C(\boldsymbol{p}_v)$ **then**
  Predict $v$ by MLP's prediction $\boldsymbol{p}_v$
**else**
  Run the trained GNN expert on $v$
  Predict $v$ by GNN's prediction $\boldsymbol{p}'_v$
**end if**

---

**Algorithm 2** `Mowst` training

**Input:** $\mathcal{G}(\mathcal{V}, \mathcal{E}, \boldsymbol{X})$; training labels $\{\boldsymbol{y}_v\}$
Initialize MLP & GNN weights as $\boldsymbol{\theta}_0$ & $\boldsymbol{\theta}'_0$
**for** round $r = 1$ until convergence **do**
  Fix GNN weights $\boldsymbol{\theta}'_{r-1}$
  Update MLP weights to $\boldsymbol{\theta}_r$ by gradient descent on $L_{\texttt{Mowst}}$
until convergence
  Fix MLP weights $\boldsymbol{\theta}_r$
  Update GNN weights to $\boldsymbol{\theta}'_r$ by gradient descent on $L_{\texttt{Mowst}}$
until convergence
**end for**

---

**Inference.** Algorithm 1 is executed on an already-trained `Mowst`: on target $v$, the two experts each execute independently until they generate their own predictions $\boldsymbol{p}_v$ and $\boldsymbol{p}'_v$. The mixture module either accepts $\boldsymbol{p}_v$ with probability $C(\boldsymbol{p}_v)$ (in which case we do not need to execute the GNN at all), or discards $\boldsymbol{p}_v$ and uses $\boldsymbol{p}'_v$ as the final prediction. We use a random number $q$ to simulate the MLP's activation probability $C(\boldsymbol{p}_v)$, where $C(\boldsymbol{p}_v)$ reflects how confident the MLP is in its prediction. For instance, in binary classification, $\boldsymbol{p}_v = [0, 1]^\mathsf{T}$ and $[0.5, 0.5]^\mathsf{T}$ correspond to cases where the MLP is certain and uncertain about its prediction, with $C(\boldsymbol{p}_v)$ being close to 1 and 0, respectively.

**Training.** The training should minimize the *expected* loss incurred in inference. In Algorithm 1, the MLP incurs loss $L(\boldsymbol{p}_v, \boldsymbol{y}_v)$ with probability $C(\boldsymbol{p}_v)$. Therefore, the overall loss on expectation is:

$$L_{\texttt{Mowst}} = \frac{1}{|\mathcal{V}|} \sum_{v \in \mathcal{V}} \big( C(\boldsymbol{p}_v) \cdot L(\boldsymbol{p}_v, \boldsymbol{y}_v) + (1 - C(\boldsymbol{p}_v)) \cdot L(\boldsymbol{p}'_v, \boldsymbol{y}_v) \big) \tag{1}$$

where $\boldsymbol{p}_v = \texttt{MLP}(\boldsymbol{x}_v; \boldsymbol{\theta})$ and $\boldsymbol{p}'_v = \texttt{GNN}(\boldsymbol{x}_v; \boldsymbol{\theta}')$ are the experts' predictions; $\boldsymbol{\theta}$ and $\boldsymbol{\theta}'$ are the experts' model parameters. $L_{\texttt{Mowst}}$ is fully differentiable. While we could simultaneously optimize both experts via standard gradient descent, Algorithm 2 shows a "training in turn" strategy: we fix one expert's parameters while optimizing the other. Training each expert in turn enables the experts to fully optimize themselves despite having different convergence behaviors due to the distinct model architectures. If $C$ is learnable (§2.2), we update its parameters together with MLP's $\boldsymbol{\theta}$.

**Intuition.** We demonstrate via a simple case study how $C$ moderates the two experts. Suppose at some point of training, the two experts have the same loss on some $v$. In case 1, $v$'s self features are sufficient for a good prediction. Then MLP can make $\boldsymbol{p}_v$ more certain to lower its $L(\boldsymbol{p}_v, \boldsymbol{y}_v)$ and increase $C$. The improved MLP's loss then contributes more to $L_{\texttt{Mowst}}$ than GNN's loss, thus

improving the overall $L_{\mathtt{Mowst}}$. In case 2, $v$'s self-features are insufficient for further reducing MLP's loss. If the MLP makes $\boldsymbol{p}_v$ more certain, $L_{\mathtt{Mowst}}$ will deteriorate since an increased $C$ will up-weight the contribution of a worse $L\left(\boldsymbol{p}_v, \boldsymbol{y}_v\right)$. Otherwise, if the MLP acts in the reverse way to even degrade $\boldsymbol{p}_v$ to random guess, $L\left(\boldsymbol{p}_v, \boldsymbol{y}_v\right)$ will be worse but $C$ reduces to 0, leaving $L_{\mathtt{Mowst}}$ unaffected even if MLP is completely ignored. Case 2 shows the bias of $\mathtt{Mowst}$ towards the GNN. However, as shown in case 1, the weak MLP can still play a significant role in nodes where it specializes effectively.

## 2.2 Confidence function

In this subsection, we omit the subscript $v$. We formally categorize the class of the confidence function $C$. Consider the single-task, multi-class node classification problem, where $n$ is the total number of classes. The input to $C$, $\boldsymbol{p}$, belongs to the standard $(n-1)$-simplex, $\mathcal{S}_{n-1}$. *i.e.*, $\sum_{1 \le i \le n} p_i = 1$ and $p_i \ge 0$, for $i = 1, \ldots, n$. The output of $C$ falls between 0 and 1. Since $C$ reflects the certainty of the MLP's prediction, we decompose it as $C = G \circ D$, where $\circ$ denotes function composition. Here, $D$ takes $\boldsymbol{p}$ as input and computes its dispersion, and $G$ is a real-valued scalar function that maps the dispersion to a confidence score. We define $G$ and $D$ as follows:

**Definition 2.1.** $D : \mathcal{S}_{n-1} \mapsto \mathbb{R}$ *is continuous and quasiconvex where* $D\left(\frac{1}{n}\mathbf{1}\right) = 0$ *and* $D\left(\boldsymbol{p}\right) > 0$ *if* $\boldsymbol{p} \ne \frac{1}{n}\mathbf{1}$. $G : \mathbb{R} \mapsto \mathbb{R}$ *is monotonically non-decreasing where* $G\left(0\right) = 0$ *and* $0 < G\left(x\right) \le 1, \forall x > 0$.

**Proposition 2.2.** $C = G \circ D$ *is quasiconvex.*

$D$'s definition states that only a prediction as bad as a random guess receives 0 dispersion. Typical dispersion functions such as *variance* and *negative entropy* (see §D.1.1 for definitions) belong to our class of $D$. Definition of $G$ states that $C$ does not decrease with a higher dispersion, which is reasonable as dispersion reflects the certainty of the prediction. In training, we can fix $C$ by manually specifying both $D$ and $G$. Alternatively, we can use a lightweight neural network (*e.g.,* MLP) to learn $G$, where the input to $G$'s MLP is the variance and negative entropy computed by $D$ (§3).

## 2.3 Interaction Between Experts

When optimizing the training loss in Equation 1, confidence $C$ will accumulate on some nodes while diminish on the others. Different distributions of $C$ correspond to different ways that the two experts can *specialize* and *collaborate*. In the following, we theoretically reveal three factors that control the value of $C$: the richness of self-feature information, the relative strength between the two experts, and the shape of the confidence function. The analysis on the experts' relative strength also reveals why the confidence-based gate is *biased*. Since both $C$ and the MLP loss are functions of the MLP prediction $\boldsymbol{p}_v$, we analyze the optimal $\boldsymbol{p}_v$ that minimizes $L_{\mathtt{Mowst}}$ given a *fixed* GNN expert.

We set up the following optimization problem. We first divide the graph into $m$ disjoint node partitions, where nodes are in the same partition, $\mathcal{M}_i$, if and only if they have *identical* self-features. For all nodes in $\mathcal{M}_i$, an MLP generates the same prediction denoted as $\hat{\boldsymbol{p}}_i$. On the other hand, predictions for different partitions, $\hat{\boldsymbol{p}}_i$ and $\hat{\boldsymbol{p}}_j$ ($i \ne j$), can be *arbitrarily* different since an MLP is a universal approximator (Hornik et al., 1989). The above properties convert Equation 1 from an optimization problem on $\boldsymbol{\theta}$ (MLP's weights) to the one on $\{\hat{\boldsymbol{p}}_i\}$ (MLP's predictions). Consider nodes $u$ and $v$ in the same partition $\mathcal{M}_i$. If they have different labels $\boldsymbol{y}_u \ne \boldsymbol{y}_v$, no MLP can distinguish $u$ from $v$ because the model lacks necessary neighborhood information. Let $\boldsymbol{\alpha}_i$ be the label distribution for $\mathcal{M}_i$. *i.e.,* $\alpha_{ij} \in (0, 1)$ portion of nodes in $\mathcal{M}_i$ have label $j$, and $\sum_{1 \le j \le n} \alpha_{ij} = 1$. We are now ready to derive the optimization problem as follows (see §E.1 for detailed steps):

$$\{\hat{\boldsymbol{p}}_i^*\} = \arg\min_{\{\hat{\boldsymbol{p}}_i\}} L_{\mathtt{Mowst}} = \arg\min_{\{\hat{\boldsymbol{p}}_i\}} \sum_{1 \le i \le m} C\left(\hat{\boldsymbol{p}}_i\right) \cdot \left(\hat{L}_{\boldsymbol{\alpha}_i}\left(\hat{\boldsymbol{p}}_i\right) - \mu_i\right) \tag{2}$$

where $\hat{L}_{\boldsymbol{\alpha}_i}\left(\hat{\boldsymbol{p}}_i\right) = \frac{1}{|\mathcal{M}_i|} \sum_{v \in \mathcal{M}_i} L\left(\hat{\boldsymbol{p}}_i, \boldsymbol{y}_v\right) = -\sum_{1 \le j \le n} \alpha_{ij} \cdot \log \hat{p}_{ij}$ is the average MLP cross-entropy loss on nodes in $\mathcal{M}_i$; $\mu_i = \frac{1}{|\mathcal{M}_i|} \sum_{v \in \mathcal{M}_i} L\left(\boldsymbol{p}_v', \boldsymbol{y}_v\right)$ is a constant representing the average loss of the *fixed* GNN on the $\mathcal{M}_i$ nodes. We first summarize the properties of the loss as follows:

**Proposition 2.3.** *Given* $\boldsymbol{\alpha}_i$, $\hat{L}_{\boldsymbol{\alpha}_i}\left(\hat{\boldsymbol{p}}_i\right)$ *is a convex function of* $\hat{\boldsymbol{p}}_i$ *with unique minimizer* $\hat{\boldsymbol{p}}_i^* = \boldsymbol{\alpha}_i$. *Let* $\Delta\left(\boldsymbol{\alpha}\right) = \hat{L}_{\boldsymbol{\alpha}}\left(\boldsymbol{\alpha}\right)$ *be a function of* $\boldsymbol{\alpha}$, $\Delta$ *is a concave function with unique maximizer* $\boldsymbol{\alpha}^* = \frac{1}{n}\mathbf{1}$.

$\Delta(\boldsymbol{\alpha}_i)$ reflects the *best possible* performance of any MLP on a group $\mathcal{M}_i$. For optimization problem 2, Theorem 2.4 quantifies the positions of the minimizer $\hat{\boldsymbol{p}}_i^*$ and the corresponding values of $C$. In case 3 ($\Delta(\boldsymbol{\alpha}_i) < \mu_i$), it is impossible to derive the exact $\hat{\boldsymbol{p}}_i^*$ without knowing the specific $C$. Therefore, we bound $\hat{\boldsymbol{p}}_i^*$ via "sub-level sets", which characterize the shapes of the loss and confidence functions. The last sentence of Theorem 2.4 shows the "tightness" of the bound.

**Theorem 2.4.** *Suppose $C = G \circ D$ follows Definition 2.1. Denote $\mathscr{L}_{\boldsymbol{\alpha}_i}^{\mu_i} = \{\hat{\boldsymbol{p}}_i \mid \hat{L}_{\boldsymbol{\alpha}_i}(\hat{\boldsymbol{p}}_i) = \mu_i\}$ and $\mathscr{L}_{\boldsymbol{\alpha}_i}^{<\mu_i} = \{\hat{\boldsymbol{p}}_i \mid \hat{L}_{\boldsymbol{\alpha}_i}(\hat{\boldsymbol{p}}_i) < \mu_i\}$ as the level set and strict sublevel set of $\hat{L}_{\boldsymbol{\alpha}_i}$. Denote $C^{<\mu_i} = \{\hat{\boldsymbol{p}}_i \mid C(\hat{\boldsymbol{p}}_i) < \mu_i\}$ as the strict sublevel set of $C$. For a given $\boldsymbol{\alpha}_i \neq \frac{1}{n}\mathbf{1}_n$, the minimizer $\hat{\boldsymbol{p}}_i^*$ satisfies:*

- *If $\Delta(\boldsymbol{\alpha}_i) > \mu_i$, then $\hat{\boldsymbol{p}}_i^* = \frac{1}{n}\mathbf{1}_n$ and $C(\hat{\boldsymbol{p}}_i^*) = 0$.*

- *If $\Delta(\boldsymbol{\alpha}_i) = \mu_i$, then $\hat{\boldsymbol{p}}_i^* = \boldsymbol{\alpha}_i$ or $\hat{\boldsymbol{p}}_i^* = \frac{1}{n}\mathbf{1}$.*

- *If $\Delta(\boldsymbol{\alpha}_i) < \mu_i$, then $\hat{\boldsymbol{p}}_i^* \in \mathscr{L}_{\boldsymbol{\alpha}_i}^{<\mu_i} - C^{<\mu'}$ where $\mu' = C(\boldsymbol{\alpha}_i) \leq C(\hat{\boldsymbol{p}}_i^*) \leq G\left(\max_{\hat{\boldsymbol{p}}_i \in \mathscr{L}_{\boldsymbol{\alpha}_i}^{\mu_i}} D(\hat{\boldsymbol{p}}_i)\right)$. Further, there exists $C$ such that $\hat{\boldsymbol{p}}_i^* = \boldsymbol{\alpha}_i$, or $\hat{\boldsymbol{p}}_i^*$ is sufficiently close to the level set $\mathscr{L}_{\boldsymbol{\alpha}_i}^{\mu_i}$.*

Proposition 2.3 and Theorem 2.4 jointly reveal factors affecting the MLP's contribution: (1) Node feature information, $\boldsymbol{\alpha}_i$: $\boldsymbol{\alpha}_i$ being closer to $\frac{1}{n}\mathbf{1}$ implies that node features are less informative. Thus, confidence $C$ will be lower since it is harder to reduce the MLP loss (Proposition 2.3); (2) Relative strength of experts, $\Delta(\boldsymbol{\alpha}_i) - \mu_i$: when the best possible MLP loss $\Delta(\boldsymbol{\alpha}_i)$ cannot beat the GNN loss $\mu_i$, the MLP simply learns to *give up* on the node group $\mathcal{M}_i$ by generating random guess $\hat{\boldsymbol{p}}_i^* = \frac{1}{n}\mathbf{1}$. Otherwise, the MLP will beat the GNN by learning a $\hat{\boldsymbol{p}}_i^*$ in some neighborhood of $\boldsymbol{\alpha}_i$ and obtaining positive $C$. The worse the GNN (*i.e.,* larger $\mu_i$), the easier to obtain a more dispersed $\hat{\boldsymbol{p}}_i^*$ (*i.e.,* enlarged $\mathscr{L}_{\boldsymbol{\alpha}_i}^{<\mu_i}$), and so the easier to achieve a larger $C$ (*i.e.,* increased $\max_{\hat{\boldsymbol{p}}_i \in \mathscr{L}_{\boldsymbol{\alpha}_i}^{\mu_i}} D(\hat{\boldsymbol{p}}_i)$); (3) Shape of the confidence function $C$: the sub-level set, $C^{<\mu'}$, constrains the range of $\hat{\boldsymbol{p}}_i^*$. Additionally, changing $G$'s shape can push $C(\hat{\boldsymbol{p}}_i^*)$ towards the lower bound $C(\boldsymbol{\alpha}_i)$ or the upper bound $G\left(\max_{\hat{\boldsymbol{p}}_i \in \mathscr{L}_{\boldsymbol{\alpha}_i}^{\mu_i}} D(\hat{\boldsymbol{p}}_i)\right)$. See §D.1 for specific $G$ constructions and all proofs.

Point (1) is a data property. Point (2) leads to diverse training dynamics (§2.4). Additionally, it reflects Mowst's inherent bias: a better GNN completely dominates the MLP by $1 - C(\hat{\boldsymbol{p}}_i^*) = 1$, while a better MLP only softly deprecates the GNN with $C(\hat{\boldsymbol{p}}_i^*) \leq 1$. Such bias is by design since we only cautiously activate the weak expert: (a) A GNN is harder to optimize, so even if it temporarily performs badly during training, we still preserve a certain weight $1 - C$ hoping that it later catches up with the MLP; (b) Since a GNN generalizes better on the test set (Yang et al., 2023), we prefer it when the two experts have similar training performance. Point (3) justifies our learnable $G$ in §2.2.

### 2.4 TRAINING DYNAMICS

**Specialization via data splitting.** At the beginning of training ($r = 1$, Algorithm 2), a randomly initialized GNN approximately produces random guesses. Therefore, the MLP generates a distribution of $C$ over all training nodes purely based on the importance of self-features. Denote the subset of nodes with large $1 - C$ as $\mathcal{S}_{\mathrm{GNN}}$. When it is the GNN's turn to update its model parameters, the GNN optimizes $\mathcal{S}_{\mathrm{GNN}}$ better due to the larger loss weight. In the subsequent round ($r = 2$), the MLP will struggle to outperform the GNN on many nodes, especially on $\mathcal{S}_{\mathrm{GNN}}$. According to Theorem 2.4, the MLP will completely ignore such challenging nodes by setting $C = 0$ on $\mathcal{S}_{\mathrm{GNN}}$, thus simplifying the learning task for the weak expert. When MLP converges again, it specializes better on a smaller subset of the training set, and the updated $C$ distribution leads to a clearer data partition between the two experts. This iterative process continues, with each additional round of "in-turn training" reinforcing better specialization between the experts.

**Denoised fine-tuning.** This is a special case of "specialization" where nodes are dominantly assigned to the GNN expert. This case can happen for two reasons: (1) GNN is intrinsically stronger; (2) Our confidence-based gate favors the GNN (§2.3). Let $\mathcal{S}_{\mathrm{MLP}}$ and $\mathcal{S}_{\mathrm{GNN}}$ represent the sets of nodes assigned to the two experts when the training of Mowst has *almost converged*. According to the "relative strength" analysis in §2.3, $\mathcal{S}_{\mathrm{MLP}}$ may only constitute a small set of *outlier* nodes with a high level of structural noises. Such $\mathcal{S}_{\mathrm{MLP}}$ does not capture a meaningful distribution of the training data, and thus the MLP overfits when it optimizes on $\mathcal{S}_{\mathrm{MLP}}$ and ignores $\mathcal{S}_{\mathrm{GNN}}$. What's interesting is that

now MLP's role switches from true specialization to noise filtering: Without the MLP, $\mathcal{S}_{\text{MLP}}$ and $\mathcal{S}_{\text{GNN}}$ are both in the GNN's training set. While the size of $\mathcal{S}_{\text{MLP}}$ is small, it may generate harmful gradients significant enough to make the GNN stuck in sub-optimum. With the MLP, $\mathcal{S}_{\text{MLP}}$ is eliminated from the GNN's training set, enabling the GNN to further fine-tune on $\mathcal{S}_{\text{GNN}}$ without the negative impact from $\mathcal{S}_{\text{MLP}}$. The improvement is not attributable to the MLP's performance on $\mathcal{S}_{\text{MLP}}$, but rather to the enhancement of the GNN's model quality after training on a cleaner dataset.

## 2.5 Mowst$^\star$

We propose a model variant, Mowst$^\star$, by modifying the training loss of Mowst. In Equation 1, we compute the losses of the two experts separately, and then combine them via $C$ to obtain the overall loss $L_{\text{Mowst}}$. In Equation 3, we first use $C$ to combine the predictions of the two experts, $\boldsymbol{p}_v$ and $\boldsymbol{p}'_v$, and then calculate a single loss for $L^\star_{\text{Mowst}}$. Note that the cross entropy loss, $L(\cdot, \boldsymbol{y}_v)$ is convex, and the weighted sum based on $C$ also corresponds to a convex combination. We thus derive Proposition 2.5, which states that Mowst$^\star$ is theoretically better than Mowst due to a lower loss.

$$L^\star_{\text{Mowst}} = \frac{1}{|\mathcal{V}|} \sum_{v \in \mathcal{V}} L\big( C\left(\boldsymbol{p}_v\right) \cdot \boldsymbol{p}_v + \left(1 - C\left(\boldsymbol{p}_v\right)\right) \cdot \boldsymbol{p}'_v, \ \boldsymbol{y}_v \big) \tag{3}$$

**Proposition 2.5.** *For any function $C$ with range in $[0, 1]$, $L_{Mowst}$ upper-bounds $L^\star_{Mowst}$.*

Since Mowst$^\star$ only has a single loss term, we no longer employ the "in-turn training" of Algorithm 2. Instead, we directly differentiate $L^\star_{\text{Mowst}}$ and update the MLP, GNN, and the learnable $C$ altogether. During inference, Mowst$^\star$ computes both experts and predicts via $C\left(\boldsymbol{p}_v\right) \cdot \boldsymbol{p}_v + \left(1 - C\left(\boldsymbol{p}_v\right)\right) \cdot \boldsymbol{p}'_v$.

**Mowst *vs.* Mowst$^\star$.** Both variants have similar collaboration behaviors (*e.g.,* §2.4), primarily driven by the "weak-strong" design choice and the confidence mechanism. Regarding trade-offs, Mowst may be easier to optimize as its "in-turn training" fully decouples the experts' different model architectures, while Mowst$^\star$ has a theoretically lower loss. Both variants can be practically useful.

## 2.6 Expressive power and computation complexity

We jointly analyze Mowst and Mowst$^\star$ due to their commonalities. Since the MLP and GNN experts execute independently before being combined, it is intuitive that our system can express each expert alone by simply disabling the other expert. In the following theoretical results, the term "expressive power" is used in accordance with its standard definition (Lu et al., 2017), which characterizes the neural network's ability to approximate functions. The formal definition and proof are in §D.2. In Proposition 2.6, we construct a specific $G$ so that the confidence function acts as a binary gate between experts. Theorem 2.7 provides a stronger result on the GCN architecture (Kipf & Welling, 2016): when neighbors are noisy (or even adversarial), their aggregation brings more harm than benefit. An MLP is a simple yet effective way to eliminate such neighbor noises. The proof share inherent connections with the empirically observed "denoising" behavior in §2.4.

**Proposition 2.6.** *Mowst and Mowst$^\star$ are at least as expressive as the MLP or GNN expert alone.*

**Theorem 2.7.** *Mowst-GCN and Mowst$^\star$-GCN are more expressive than the GCN expert alone.*

For computation complexity, we consider the average cost of predicting one node in inference. For sake of discussion, we consider the GCN architecture and assume that both experts have the same number of layers $\ell$ and feature dimension $f$. In the worst case, both experts are activated (as noted in Algorithm 1, we can skip the strong expert with probability $C$). The cost of the GCN is lower bounded by $\Omega\left(f^2 \cdot (\ell + b_{\ell-1})\right)$, where $b_{\ell-1}$ is the average number of $(\ell-1)$-hop neighbors. The cost of the MLP is $O\left(f^2 \cdot \ell\right)$. On large graphs, the neighborhood size $b_{\ell-1}$ can grow exponentially with $\ell$, so realistically, $b_{\ell-1} \gg \ell$. This means the worst-case cost of Mowst or Mowst$^\star$ is similar to that of a vanilla GCN. The conclusion still holds if we use an additional, lightweight MLP to compute the confidence $C$ (since such an MLP only approximates a scalar function $G$, it is not expensive).

## 2.7 Mixture of progressively stronger experts

Suppose we have a series of $k$ *progressively stronger* experts where expert $i$ is stronger than $j$ if $i > j$ (*e.g.,* 3 experts consisting of MLP, SGC (Wu et al., 2019) and GCN (Kipf & Welling, 2016)).

Mowst and Mowst* can be generalized following a *recursive* formulation. Let $L_i^v$ be the loss of expert $i$ on node $v$, $L_{\geq i}^v$ be the loss of a Mowst sub-system consisting of experts $i$ to $k$, and $C_i^v$ be the confidence of expert $i$ computed by $i$'s prediction on node $v$. Equation 1 then generalizes to $L_{\text{Mowst}} = \frac{1}{|\mathcal{V}|} \sum_{v \in \mathcal{V}} L_{\geq 1}^v$ and $L_{\geq i}^v = C_i^v \cdot L_i^v + (1 - C_i^v) \cdot L_{\geq i+1}^v$. During inference, similar to Algorithm 1, the strong expert will be activated only when all previous weaker experts are bypassed due to their low confidence. The case for Mowst* can be derived similarly. See §E.2 for the final form of $L_{\text{Mowst}}$. Due to the recursive nature of adding new experts, all our analyses and observations in previous sections still hold. For instance, the expressive power is ensured as follows:

**Proposition 2.8.** *Mowst and Mowst* are at least as expressive as any expert alone.*

## 3 EXPERIMENTS

Table 1: Mowst outperforms baselines under the same number of layers and hidden dimension. Values with '†', '‡' and '††' are from Hu et al. (2020), Lim et al. (2021), and Wang et al. (2023). For each graph, we show the **best** and second best results, and **absolute gains** against the GNN counterparts (*e.g.,* Mowst(*)-GCN *vs.* GCN and GraphMoE-GCN). All results are averaged over 10 runs.

| | Flickr | ogbn-products | ogbn-arxiv | Penn94 | pokec | twitch-gamer |
|---|---|---|---|---|---|---|
| MLP | 46.93 ±0.00 | 61.06† ±0.08 | 55.50† ±0.23 | 73.61‡ ±0.40 | 62.37‡ ±0.02 | 60.92‡ ±0.07 |
| GAT | 52.47 ±0.14 | OOM | 71.58 ±0.17 | 81.53‡ ±0.55 | 71.77‡ ±6.18 | 59.89‡ ±4.12 |
| GPR-GNN | 53.23 ±0.14 | 72.41 ±0.04 | 71.10 ±0.22 | 81.38‡ ±0.16 | 78.83‡ ±0.05 | 61.89‡ ±0.29 |
| AdaGCN | 48.96 ±0.06 | 69.06 ±0.04 | 58.45 ±0.50 | 74.42 ±0.58 | 55.92 ±0.35 | 61.02 ±0.14 |
| GCN | 53.86 ±0.37 | 75.64† ±0.21 | 71.74† ±0.29 | 82.17 ±0.04 | 76.01 ±0.49 | 62.42 ±0.53 |
| GCN-skip | 52.98 ±0.00 | - | 69.56 ±0.00 | 76.58 ±0.53 | 73.46 ±0.04 | 61.05 ±0.23 |
| GraphMoE-GCN | 53.03 ±0.14 | 73.90 ±0.00 | 71.88†† ±0.32 | 81.61 ±0.27 | 76.99 ±0.10 | 62.76 ±0.22 |
| Mowst(*)-GCN | 54.62 ±0.23 | 76.49 ±0.22 | **72.52** ±0.07 | 83.19 ±0.43 | 77.28 ±0.08 | 63.74 ±0.23 |
| | **(+0.76)** | **(+0.85)** | **(+0.64)** | **(+1.02)** | **(+0.29)** | **(+0.83)** |
| GIN | 53.71 ±0.35 | - | 69.39 ±0.56 | 82.68 ±0.32 | 53.37 ±2.15 | 61.76 ±0.60 |
| Mowst(*)-GIN | **55.48** ±0.32 | - | 71.43 ±0.26 | **84.56** ±0.31 | 76.11 ±0.39 | 64.32 ±0.34 |
| | **(+1.77)** | | **(+2.04)** | **(+1.88)** | **(+22.74)** | **(+2.56)** |
| GIN-skip | 52.70 ±0.00 | - | 71.28 ±0.00 | 80.32 ±0.43 | 76.29 ±0.51 | 64.27 ±0.25 |
| Mowst(*)-GIN-skip | 53.19 ±0.31 | - | 71.79 ±0.23 | 81.20 ±0.55 | **79.70** ±0.23 | **64.91** ±0.22 |
| | **(+0.49)** | | **(+0.51)** | **(+0.88)** | **(+3.41)** | **(+0.64)** |
| GraphSAGE | 53.51 ±0.05 | 78.50† ±0.14 | 71.49† ±0.27 | 76.75 ±0.52 | 75.76 ±0.04 | 61.99 ±0.30 |
| GraphMoE-SAGE | 52.16 ±0.13 | 77.79 ±0.00 | 71.19 ±0.15 | 77.04 ±0.55 | 76.67 ±0.08 | 63.42 ±0.23 |
| Mowst(*)-SAGE | 53.90 ±0.18 | **79.38** ±0.44 | 72.04 ±0.24 | 79.07 ±0.43 | 77.84 ±0.04 | 64.38 ±0.14 |
| | **(+0.39)** | **(+0.88)** | **(+0.55)** | **(+2.03)** | **(+1.33)** | **(+1.05)** |

**Setup.** We evaluate Mowst(*) on a diverse set of benchmarks, including 3 homophilous graphs (Flickr (Zeng et al., 2020), ogbn-arxiv and ogbn-products (Hu et al., 2020)) and 3 heterophilous graphs (Penn94, pokec and twitch-gamer (Lim et al., 2021)). Following the literature, we perform node classification using the "accuracy" metric, with standard training / validation / test splits. The graph sizes range from 89K (Flickr) to 2.4M (ogbn-products) nodes. See also §A.1.

BASELINES. GCN (Kipf & Welling, 2016), GraphSAGE (Hamilton et al., 2017), GAT (Veličković et al., 2018), and GIN (Xu et al., 2019) are the most widely used GNN architectures achieving state-of-the-art accuracy on both homophilous (Hu et al., 2020; Shi et al., 2023) and heterophilous (Lim et al., 2021; Zhu et al., 2021; Platonov et al., 2023) graphs. In addition, GPR-GNN (Chien et al., 2021) decouples the aggregation of neighbors within different hops, which effectively addresses the challenges on heterophilous graphs (Lim et al., 2021; Platonov et al., 2023). We also compare with H₂GCN (Zhu et al., 2020b), the state-of-the-art heterophilous GNN. Additionally, we construct variants of the baselines by adding skip connections. Since GraphSAGE and H₂GCN have already implemented skip connections in their original design, we thus only integrate skip connection into GCN and GIN. Due to resource constraints, we exclude these variants in our experiments on the largest ogbn-products graph. AdaGCN (Sun et al., 2021) and GraphMoE (Wang et al., 2023) are the state-of-the-art ensemble and MoE models, both combining multiple GNNs.

HYPERPARAMETERS. For Flickr, ogbn-products and ogbn-arxiv, we follow the original literature (Hu et al., 2020; Zeng et al., 2021) to set the number of layers as 3 and hidden dimension as 256, for all the baselines as well as for both the MLP and GNN experts of Mowst(*). Regarding Penn94, pokec and twitch-gamer, the authors of the original paper (Lim et al., 2021) searched

for the best network architecture for each baseline independently. We follow the same protocol and hyperparameter space for our baselines and `Mowst(*)`. We set an additional constraint on `Mowst(*)` to ensure fair comparison under similar computation costs: we first follow Lim et al. (2021) to determine the number of layers $\ell$ and hidden dimension $d$ of the vanilla GNN baselines, and then set the same $\ell$ and $d$ for the corresponding `Mowst` models. We use another MLP to implement the learnable $G$ function (§2.2). To reduce the size of the hyperparameter space, we set the number of layers and hidden dimension of $G$'s MLP the same as those of the experts. See §A.2 for the hyperparameter space (*e.g.,* learning rate, dropout) and the grid-search methodology.

**Comparison with state-of-the-art.**   As shown in Table 1, `Mowst(*)` consistently achieves significant accuracy improvement on all datasets. It also performs the best compared to all the other baselines. Moreover, we perform a comparison within two sub-groups, one with GCN as the backbone architecture including GCN, GraphMoE-GCN and `Mowst(*)-GCN` and the other with GraphSAGE including GraphSAGE, GraphMoE-SAGE and `Mowst(*)-SAGE`. In both sub-groups, `Mowst(*)` achieves significant accuracy improvement over the second best. The accuracy improvement is consistently observed across both homophilous and heterophilous graphs, showing that the decoupling of the self-features and neighbor structures, along with the denoising effect of the weak expert are generally beneficial.

Table 2 presents the experimental results for $H_2$GCN and `Mowst` on three heterophilous graphs. $H_2$GCN consistently outperforms the GCN baseline, demonstrating its ability to effectively handle heterophilous neighborhoods. More importantly, when compared to $H_2$GCN, `Mowst(*)-H₂GCN` achieves a *significant and consistent improvement* in accuracy across all three graphs. This shows that `Mowst` can sub-

Table 2: Comparison with $H_2$GCN on heterophilous graphs.

|  | Penn94 | pokec | twitch-gamer |
|---|---|---|---|
| GCN | 82.17 ±0.04 | 76.01 ±0.49 | 62.42 ±0.53 |
| `Mowst(*)-GCN` | 83.19 ±0.43 | 77.28 ±0.08 | 63.74 ±0.23 |
|  | **(+1.02)** | **(+0.29)** | **(+0.83)** |
| $H_2$GCN | 82.71 ±0.67 | 80.89 ±0.16 | 65.70 ±0.20 |
| `Mowst(*)-H₂GCN` | **83.39** ±0.43 | **83.02** ±0.30 | **66.03** ±0.16 |
|  | **(+0.68)** | **(+2.13)** | **(+0.33)** |

stantially enhance the performance of a state-of-the-art heterophilous GNN like $H_2$GCN, with the help of a relatively simple expert such as a standard MLP.

**MLP expert *vs.* skip connections.**   We compare with the "skip connection" variants of the GNN baselines. Since the accuracy of GIN-skip is close to that of `Mowst(*)-GIN` on `ogbn-arxiv`, `pokec` and `twitch-gamer`, we additionally run `Mowst(*)-GIN-skip`. We observe that `Mowst(*)-GIN` *consistently improves* the accuracy of the GIN baseline by a *large margin*. Effects of skip connections on GCN, GraphSAGE, GIN and $H_2$GCN clearly indicate that the MLP expert integrated by `Mowst` enhances model capacity in a *unique* and *valuable* way. The MLP expert is not simply providing a shortcut to bypass certain neighborhood aggregation operations. Rather, it meaningfully interact with the GNN expert to let the full system personalize on each target node. Note that there is no need to run `Mowst(*)-GCN-skip` since `Mowst(*)-GCN` already outperforms GCN-skip significantly w.r.t. both accuracy and computation cost. The analysis on computation cost can be found in §E.3.4.

**`Mowst` *vs.* `Mowst*`.**   Using the GCN architecture as an example, we compare three model variants: (1) `Mowst`, (2) `Mowst*` and (3) `Mowst` (joint), a vanilla version of `Mowst` that performs training by simultaneously updating the two experts from the gradients of

Table 3: Comparison of test set accuracy.

|  | Flickr | pokec | twitch-gamer |
|---|---|---|---|
| `Mowst-GCN (joint)` | 53.47±0.36 | 76.62±0.11 | 63.44±0.22 |
| `Mowst-GCN` | **54.62**±0.23 | 77.12±0.09 | **63.74**±0.23 |
| `Mowst*-GCN` | 53.94±0.37 | **77.28**±0.08 | 63.59±0.11 |

the whole $L_{\texttt{Mowst}}$ (Equation 1). `Mowst` (joint) does not follow the "in-turn training" strategy discussed in Algorithm 2. From Table 3, we observe that `Mowst` (joint) achieves lower test accuracy on all three graphs. Intuitively, since the two experts may have difference convergence rates, updating one expert with the other one fixed may help stabilize training and improve the overall convergence quality. Moreover, `Mowst` or `Mowst*` may outperform the other variant depending on the property of the graph, which is consistent with our trade-off analysis in §2.5. `Mowst*` can theoretically achieve a lower loss, while `Mowst` may be easier to optimize. In practice, both variants can be useful.

**Training dynamics.**   The two behaviors described in §2.4 may be observed on both `Mowst` and `Mowst*`. For demonstration, we visualize each behavior using one model variant. The empirical findings on "denoised fine-tuning" are provided in §C.2.

SPECIALIZATION VIA DATA SPLITTING. We track the evolution of the confidence score $C$ during `Mowst*-GCN` training. In Figure 2, for each benchmark, we plot $C$'s distribution corresponding to different model snapshots. The distribution on the upper side corresponds to the earlier stage of training, while the distribution on the lower side reflects the weighting between the two experts after convergence. In each distribution, the height shows the percentage of nodes

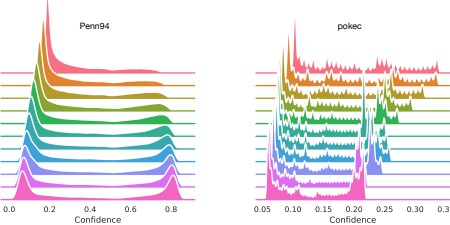

Figure 2: Evolution of the $C$ distribution.

assigned with such $C$. On `Penn94`, the GNN dominates initially. The MLP gradually learns to specialize on a significant portion of the data. Eventually, the whole graph is clearly split between the two experts, indicated by the two large peaks around 0 and 0.8 (corresponding to nodes assigned to GNN and MLP, respectively). For `pokec`, the specialization is not so strong. When training progresses, the MLP becomes less confident when the GNN is optimized better – nodes initially having larger $C$ gradually reduce $C$ to form a peak at around 0.2. For different graphs, the two experts will adapt their extent of collaboration by minimizing the confidence-weighted loss.

## 4 RELATED WORK

Our work is closely related to Mixture-of-Experts, model ensemble methods on graphs, as well as the decoupling models for feature and structure information. Please refer to §B for further discussion.

**Mixture-of-Experts.** MoE can effectively increase model capacity (Masoudnia & Ebrahimpour, 2014; Yuksel et al., 2012). The key idea is to localize / specialize different expert models to different partitions of the data. Typically, experts have comparable strengths. Thus, the numerous gating functions in the literature have a symmetric form w.r.t. all experts without biasing towards a specific one (*e.g.*, variants of softmax (Jordan & Jacobs, 1994; Shazeer et al., 2017; Puigcerver et al., 2023) In comparison, `Mowst` deliberately breaks the balance among experts via a biased gating. Another concern in MoE is the increased computation cost. Even though efficient dense gating designs exist (Nie et al., 2022), sparse gating is still the most common technique to save computation by conditionally deactivating some experts. However, it is widely observed that such sparsity causes training instability (Fedus et al., 2022; Shazeer et al., 2017; Puigcerver et al., 2023). Under our confidence-based gating, `Mowst` is both easy to train (§2.4) and computationally efficient (§2.6). Finally, many recent designs incorporate MoE in each layer of a large model (Riquelme et al., 2021; Du et al., 2022; Lepikhin et al., 2021), whereas `Mowst` mixes only at the output layers. We fully decouple the experts with imbalanced strengths in consideration of their different training dynamics.

**MoE & model ensemble on graphs.** GraphDIVE (Hu et al., 2022) proposes mixture of multi-view experts to address the class-imbalance issue in graph classification. GraphMoE (Wang et al., 2023) introduces mixing multiple GNN experts via the top-$K$ sparse gating (Shazeer et al., 2017). Zhou & Luo (2019) explore a similar idea of mixing multiple GNNs with existing gating functions. AdaGCN (Sun et al., 2021) proposes a model ensemble technique for GNNs, based on the classic AdaBoost algorithm (Freund et al., 1999). In addition, AM-GCN (Wang et al., 2020b) performs graph convolution on the decoupled topological and feature graphs to learn multi-channel information.

## 5 CONCLUSION

In this study, we introduce a novel mixture-of-experts design to combine a weak MLP expert with a strong GNN expert to effectively decouple the self-feature and neighbor structure modalities on graphs. The "weak-strong" collaboration emerges under a gating mechanism based on the weak expert's prediction confidence. We theoretically and empirically demonstrate intriguing training dynamics that evolve based on the properties of the graph and the relative strengths of the experts. We show significant accuracy improvement over state-of-the-art methods across both homophilous and heterophilous graphs. Future work will focus on a comprehensive evaluation of the generalized `Mowst` framework, which progressively mixes stronger experts, as outlined in §2.7. Additionally, exploring whether the "weak-strong" combination can serve as a general MoE design paradigm with benefits extending *beyond graph learning* is an interesting direction for further research.

ACKNOWLEDGMENTS

We are grateful to Jingyang Lin, Dr. Wei Zhu, and Dr. Wei Xiong for their constructive suggestions. Luo was supported in part by NSF Award #2238208.

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

**Table of Contents (Appendix)**

## A  MORE DETAILS ON EXPERIMENTS

### A.1  DATASET DESCRIPTION & STATISTICS

Table 4 provides a summary of the basic statistics for the six benchmark graphs. The `Flickr` dataset is originally introduced in Zeng et al. (2020), while `ogbn-products` and `ogbn-arxiv` are first proposed in Hu et al. (2020). The `Penn94`, `pokec`, and `twitch-gamer` datasets are originally proposed in Lim et al. (2021). A brief overview of the dataset construction is as follows:

Table 4: **Statistics of the datasets.**

|  | # nodes | # edges | # classes | # node features |
|---|---|---|---|---|
| Flickr | 89,250 | 899,756 | 7 | 500 |
| ogbn-products | 2,449,029 | 61,859,140 | 47 | 100 |
| ogbn-arxiv | 169,343 | 2,332,486 | 40 | 128 |
| Penn94 | 41,554 | 1,362,229 | 2 | 5 |
| pokec | 1,632,803 | 30,622,564 | 2 | 65 |
| twitch-gamer | 168,114 | 6,797,557 | 2 | 7 |

`Flickr`, proposed by Zeng et al. (2020), is a graph where each node represents an image uploaded to Flickr. An undirected edge is drawn between two nodes if the corresponding images share common properties, such as the same geographic location, same gallery, or were commented on by the same user. The node features are 500-dimensional bag-of-words representations of the images.

`ogbn-arxiv`, a paper citation network curated by Hu et al. (2020), consists of nodes representing arXiv papers and directed edges indicating citation relationships. The 128-dimensional node features are derived by averaging the embeddings of words in the paper's title and abstract, with embeddings generated by a word2vec model trained on the MAG corpus (Wang et al., 2020a).

`ogbn-products`, also curated by Hu et al. (2020), represents an Amazon product co-purchasing network. Each node corresponds to an Amazon product, and edges between nodes indicate that the two products are frequently purchased together. The node features are 100-dimensional bag-of-words vectors of the product descriptions.

`Penn94` (Lim et al., 2021; Traud et al., 2012) is a Facebook 100 network from 2005 representing a friendship network of university students. Each node represents a student, with node features including major, second major/minor, dorm/house, year, and high school.

`pokec` (Lim et al., 2021; Leskovec & Krevl, 2014) is the friendship graph of a Slovak online social network. Nodes represent users, and edges represent directed friendship relations. Node features include profile information such as geographical region, registration time, and age.

`twitch-gamer` (Lim et al., 2021; Sarkar & Rózemberczki, 2021) is a connected undirected graph representing relationships between accounts on the streaming platform Twitch. Each node represents

a Twitch user, and an edge indicates mutual followers. Node features include the number of views, creation and update dates, language, lifetime, and whether the account is dead.

## A.2 HYPERPARAMETERS

We perform grid search within the entire hyperparameter space defined as follows. Note that for GPR-GNN (Chien et al., 2021) and AdaGCN (Sun et al., 2021), we use the same grid search as reported in their original papers.

- MLP, GCN, GCN-skip, GraphSAGE, GAT, GIN, GIN-skip: learning rate $lr \in \{.1, .01, .001\}$ for all graphs. For all the homophilous graphs, the dropout ratio is searched in $\{.1, .2, .3, .4, .5\}$, hidden dimension is set as 256 and the number of layers is 3 according to Hu et al. (2020). For the heterophilous graphs, we follow the same hyperparameter space as Lim et al. (2021).
- GPR-GNN: $lr \in \{.01, .05, .002\}$, $\alpha \in \{.1, .2, .5, .9\}$, hidden dimension $\in \{16, 32, 64, 128, 256\}$.
- AdaGCN: $lr \in \{.01, .005, .001\}$, number of layers $\in \{5, 10\}$, dropout ratio $\in \{.0, .5\}$.
- GraphMoE: $lr \in \{.1, .01, .001\}$, dropout ratio $\in \{.1, .2, .3, .4, .5\}$, number of experts $\in \{2, 8\}$.
- H$_2$GCN: $lr \in \{.01, .001\}$, dropout ratio $\in \{.1, .3, .5\}$.
- `Mowst` and `Mowst`$^\star$: The learning rate and dropout ratio of the MLP expert and the GNN expert are kept the same. The girds for them are $lr \in \{.1, .01, .001\}$, dropout ratio $\in \{.1, .2, .3, .4, .5\}$. As for the gating module, we also conduct a grid search to find the optimal combination of the learning rate and dropout ratio. The grid for them are $lr \in \{.1, .01, .001\}$, dropout ratio $\in \{.1, .2, .3, .4, .5\}$.

## A.3 IMPLEMENTATION DETAILS

The implementations for `Mowst` and the GNN baselines are built using PyTorch and the PyTorch Geometric library. The code for GraphMoE,[1] AdaGCN,[2] and H$_2$GCN[3] are sourced from the official repositories associated with their respective original papers. The GPR-GNN implementation is adapted from the repository provided by Lim et al. (2021). All models, including our models and the baselines, are trained on NVIDIA A100 GPUs with 80GB of memory. Training is conducted using full-batch gradient descent with the Adam optimizer (Kingma & Ba, 2014), without resorting to neighborhood or subgraph sampling techniques.

## A.4 PRETRAINING

In our experiments, we find that pretraining the individual experts before the training process of `Mowst` (as described in Algorithm 2) can also improve the optimization. There are four pretraining scenarios for `Mowst`: pretraining only the weak expert, pretraining only the strong expert, pretraining both experts, and not pretraining either expert.

## A.5 CRITERIA FOR FAIR COMPARISON

For a fair experimental comparison between `Mowst` and its corresponding vanilla GNN baseline (*e.g.,* `Mowst-GCN` and GCN), two potential criteria can be considered:

- **Model size**: `Mowst` and its corresponding vanilla GNN baseline should have the same number of parameters.
- **Computation cost**: `Mowst` and its corresponding vanilla GNN baseline should have similar computation cost.

**The above two criteria may not be simultaneously satisfied**. Consider a scenario where both the MLP and GNN experts in `Mowst` have the same number of layers and hidden dimensions $f$. If the

---

[1]`https://github.com/VITA-Group/Graph-Mixture-of-Experts`
[2]`https://github.com/datake/AdaGCN`
[3]`https://github.com/GemsLab/H2GCN`

vanilla GNN baseline is configured to have an architecture identical to Mowst's GNN expert, then Mowst will have more model parameters due to the additional MLP expert. However, Mowst will have a computation cost comparable to that of the vanilla GNN baseline (detailed derivation provided in §E.3.4). In this case, *the "computation cost" criterion is met, but the "model size" criterion is not*.

Alternatively, we can enhance the vanilla GNN baseline by adding a skip connection to each layer, where the skip connection performs a linear transformation with an $f \times f$ weight matrix. In essence, this approach constructs the baseline by integrating each layer of the MLP expert with each layer of the GNN. As a result, the modified GNN and Mowst have the same number of parameters. However, this configuration violates the "computation cost" criterion. As detailed in §E.3.4, the computation cost of a GNN with a skip connection is significantly higher than that of the original GNN without a skip connection, and therefore also higher than that of Mowst. Thus, *the "model size" criterion is met, but the "computation cost" criterion is not*.

Given that no setting can be perfectly fair, we need to prioritize one criterion over the other. In our experiments, we choose to emphasize the "**computation cost**" criterion, considering the significant challenges GNNs face in terms of high computation costs, rather than large model sizes. As highlighted in §E.3.3, the computation cost of GNNs can be particularly high due to the well-known "neighborhood explosion" phenomenon (Chen et al., 2018; Hamilton et al., 2017; Ying et al., 2018; Zeng et al., 2020). This leads to GNN models typically being shallow (with 2 to 3 layers), resulting in a **small model size**. For instance, the notable work PinSAGE (Ying et al., 2018) employs a 2-layer GraphSAGE model with a 2,048 hidden dimension for Pinterest recommendations, resulting in a total model size of only 84MB under 32-bit floating point representation. Despite its compact size, the computational cost of running such a GNN is substantial due to neighborhood explosion, requiring 3 days on 16 GPUs to train PinSAGE. This unique challenge in graph learning highlights that **a GNN can incur a high computation cost even with a small model size**.

Therefore, when comparing Mowst with its corresponding vanilla GNN baseline, we ensure a similar computation cost by setting the architecture of Mowst's GNN expert to be identical to the vanilla GNN baseline. It is important to note that we still conduct experiments on the GNN baselines augmented with skip connections (§A.7), but these experiments are primarily for architectural exploration.

## A.6  FURTHER DISCUSSION ON THE COMPARISON WITH THE STATE-OF-THE-ART ON HETEROPHILOUS GRAPHS

Heterophilous graphs are among the types of graphs that can benefit from our techniques, since it is commonly believed that heterophilous neighborhoods are noisy for GNN's aggregation.[4] In this section, we present additional experimental results showing that **even on top of a state-of-the-art heterophilous GNN, Mowst can still significantly improve its accuracy with the help from a weak expert** which is just a standard MLP.

We compare with the state-of-the-art heterophilous GNN baseline, $H_2$GCN (Zhu et al., 2020b). $H_2$GCN integrates model components that are specifically optimized for aggregating heterophilous neighbors. It separates ego and neighbor embeddings, recognizing that a node's self-features may differ significantly from its neighbors' aggregated embeddings in heterophilous settings. Additionally, it leverages higher-order neighborhood aggregation to glean information from wider, potentially homophily-dominant contexts, providing more relevant signals. Finally, $H_2$GCN merges intermediate representations to capture both local and global graph information.

Table 2 presents the experimental results for $H_2$GCN and Mowst on three heterophilous graphs. The experimental setup and hyperparameter search methodology are consistent with those described in §3 and §A.2, respectively. Specifically, for both $H_2$GCN and Mowst(*)-$H_2$GCN, we follow the hyperparameter space defined by Lim et al. (2021). Similar to Table 1, we add an additional constraint on the model architecture of Mowst: after identifying the optimal $H_2$GCN, we replicate its architecture to construct the GNN expert of Mowst. The MLP expert of Mowst is then configured with the same number of layers and hidden dimensions as the $H_2$GCN expert. Table 2 clearly shows the following:

- **Effectiveness of the $H_2$GCN baseline:**  $H_2$GCN achieves significantly higher accuracy than the GCN, GraphSAGE and GIN baselines. This shows that $H_2$GCN's architecture can handle heterophilous neighborhoods very well.

---

[4]However, note that heterophilous graphs are ***not*** the only type of graphs we target.

- **Performance boost by** `Mowst(*)-H2GCN`**:** Compared with H$_2$GCN, `Mowst(*)-H2GCN` further *improves the accuracy significantly and consistently* on the three graphs. The average accuracy improvement is 1.05.

## A.7 Further discussion on the comparisons with GIN & skip connection-based GNNs

In addition to the three `Mowst` variants in Table 1 and Table 2, we further construct `Mowst` on GIN and its variants, and show that `Mowst` significantly boosts the baseline accuracy across all five datasets (due to resource constraints, we exclude the largest `ogbn-products` graph when running experiments in this section). We further add skip connection to the GIN and GCN baselines, and show that

- Skip connection can help boost the baseline performance in some cases, but can also result in accuracy degradation.
- When skip connection improves the baseline, building `Mowst` with the "skip-connection GNN" expert can further boost the accuracy significantly.

Among the four GNN backbone architectures on which we have built `Mowst`, **GraphSAGE and H$_2$GCN already incorporate skip connections in their native architectural definitions**. Therefore, we only need to implement the skip-connection variants for GCN and GIN. The implementation details are as folllows:

Consider a node $v$. Let its output embedding of layer $i$ be $\boldsymbol{x}_i^v$. Let the set of neighbors of $v$ be $\mathcal{N}^v$ (excluding $v$ itself).

Without the skip connection, a GCN layer $i$ performs the following:

$$\boldsymbol{x}_i^v = \sigma\left(\boldsymbol{W}_i \cdot \sum_{u \in \mathcal{N}^v \cup \{v\}} \alpha_{uv} \boldsymbol{x}_{i-1}^u\right) \tag{4}$$

where $\boldsymbol{W}_i$ is the weight matrix, $\alpha_{uv}$ is a scalar weight determined by the graph structure and $\sigma$ is the non-linear activation (*e.g.,* ReLU).

With the skip connection, a GCN-skip layer $i$ performs the following:

$$\boldsymbol{x}_i^v = \sigma\left(\boldsymbol{W}_i' \cdot \boldsymbol{x}_{i-1}^v + \boldsymbol{W}_i \cdot \sum_{u \in \mathcal{N}^v \cup \{v\}} \alpha_{uv} \boldsymbol{x}_{i-1}^u\right) \tag{5}$$

Without the skip connection, a GIN layer $i$ performs the following:

$$\boldsymbol{x}_i^v = \sigma\left(h_{\boldsymbol{\Theta}}\left((1+\epsilon) \cdot \boldsymbol{x}_{i-1}^v + \sum_{u \in \mathcal{N}^v} \boldsymbol{x}_{i-1}^u\right)\right) \tag{6}$$

where $\epsilon$ is a learnable scalar, and $h_{\boldsymbol{\Theta}}(\cdot)$ is a 2-layer MLP with ReLU activation in the hidden layer (but not in the last layer), with $\boldsymbol{\Theta}$ denoting the learnable parameters of such an MLP.

With the skip connection, a GIN-skip layer $i$ performs the following:

$$\boldsymbol{x}_i^v = \sigma\left(\boldsymbol{W}_i' \cdot \boldsymbol{x}_{i-1}^v + h_{\boldsymbol{\Theta}}\left((1+\epsilon) \cdot \boldsymbol{x}_{i-1}^v + \sum_{u \in \mathcal{N}^v} \boldsymbol{x}_{i-1}^u\right)\right) \tag{7}$$

In Table 1, we compare GCN with GCN-skip, and GIN with GIN-skip. We observe that GCN-skip consistently degrades the performance of GCN, while GIN-skip improves the accuracy of GIN on `ogbn-arxiv`, `pokec` and `twitch-gamer` significantly. There is no need to run `Mowst(*)-GCN-skip` since `Mowst(*)-GCN` already outperforms GCN-skip by a large margin with respect to both accuracy and computation cost (see analysis of computation cost in §E.3.4). On the other hand, since the accuracy of GIN-skip is close to that of `Mowst(*)-GIN` on `ogbn-arxiv`, `pokec` and `twitch-gamer`, we run `Mowst(*)-GIN-skip` for additional comparisons. We summarize our findings as follows:

- Adding the skip connection to a GNN layer can either enhance or degrade accuracy.

- `Mowst(*)-GIN` **consistently achieves a substantial improvement in accuracy over the GIN baseline**. Notably, on `pokec`, the GIN baseline exhibits poor convergence. Adding an MLP expert of `Mowst` dramatically improves the convergence quality.

- `Mowst(*)-GIN-skip` **further boosts the accuracy of GIN-skip significantly**.

- The results of skip connections on GCN (Table 1), GraphSAGE (Table 1), GIN (Table 1), and $H_2$GCN (Table 2) demonstrate that: (1) `Mowst` is a general framework that is **applicable to various GNN architecture variants**, and (2) the contributions of the MLP expert and skip connection are different, with the weak MLP expert providing **unique value in enhancing the quality of the GNN model**.

**Remark.** Based on the reasoning in §A.5, the following pairs in Table 1 satisfy the fair comparison criteria: "GCN *vs.* `Mowst(*)-GCN`", "GIN *vs.* `Mowst(*)-GIN`" and "GIN-skip *vs.* `Mowst(*)-GIN-skip`".

## B  EXTENDED RELATED WORK

In addition to the related work (§4) in Mixture-of-Experts and ensemble methods on graphs, our work is also closely related to the **decoupling of feature and structure information and heterophilous GNNs**. GraphSAGE (Hamilton et al., 2017), GCNII (Chen et al., 2020b) and $H_2$GCN (Zhu et al., 2020b) use various forms of residual connections to facilitate self-feature propagation, with $H_2$GCN especially excelling on heterophilous graphs. GPR-GNN (Chien et al., 2021), MixHop (Abu-El-Haija et al., 2019) and SIGN (Frasca et al., 2020) blend information from various hops away via different weighting strategies. NDLS (Zhang et al., 2021) performs adaptive sampling to customize per-node receptive fields. GLNN (Zhang et al., 2022) distills structural information into an MLP. These models decouple the feature and structure information within one model, while `Mowst` does so explicitly via separate experts.

## C  ADDITIONAL EXPERIMENTS

### C.1  ALTERNATIVE CHOICE FOR WEAK & STRONG EXPERTS

In §2.3, §2.4, and §2.6, we have discussed the advantages of designing `Mowst` with an MLP as the weak expert and a GNN as the strong expert. Here, we explore an alternative approach and its trade-offs. A question one would naturally ask is: Should we keep the architecture of the two experts the same, and construct the weak & strong variants only by varying the architectural parameters. For instance, the two experts could be a shallow and a deep GNN, respectively.

We first affirm that the current setting of `Mowst` is already following the above proposal, and then conduct additional experiments by replacing an MLP with a shallow GCN.

### C.1.1  MLP AS A SHALLOW GNN

We first show that MLP and GNN are **not** fundamentally two different architectures. In fact, an MLP is **exactly** a 0-hop GNN when the GNN's layer architecture follows popular designs such as GraphSAGE (Hamilton et al., 2017), GCN (Kipf & Welling, 2016) or GIN (Xu et al., 2019).

**Notations.** Let $\boldsymbol{x}_i^v$ denote the embedding vector of node $v$ output by layer $i$, $\boldsymbol{W}_i$ the weight parameter matrix of layer $i$, and $\mathcal{N}^v$ the set of neighbor nodes of $v$. Let $\sigma\left(\cdot\right)$ represent the non-linear activation function.

**GraphSAGE.** Each layer $i$ performs $\boldsymbol{x}_i^v = \sigma\left(\boldsymbol{W}_i^1 \cdot \boldsymbol{x}_{i-1}^v + \boldsymbol{W}_i^2 \cdot \sum_{u \in \mathcal{N}^v} \boldsymbol{x}_{i-1}^u\right)$. In a 0-hop version, where the neighbor set $\mathcal{N}^v = \emptyset$, the operation simplifies to $\boldsymbol{x}_i^v = \sigma\left(\boldsymbol{W}_i^1 \cdot \boldsymbol{x}_{i-1}^v\right)$, which is identical to the operation of an MLP layer.

**GCN.** Each layer $i$ performs $\boldsymbol{x}_i^v = \sigma\left(\boldsymbol{W}_i \cdot \left(\alpha_{vv}\boldsymbol{x}_{i-1}^v + \sum_{u \in \mathcal{N}^v} \alpha_{uv}\boldsymbol{x}_{i-1}^u\right)\right)$, where $\alpha_{uu}$ and $\alpha_{uv}$ are constants pre-defined by the graph structure. If we perform 0-hop aggregation and ignore the graph structure, then $\alpha_{uu}$ and $\alpha_{uv}$ become 1 and $\mathcal{N}^v = \emptyset$. The 0-hop GCN performs $\boldsymbol{x}_i^v = \sigma\left(\boldsymbol{W}_i \cdot \boldsymbol{x}_{i-1}^v\right)$, which is again identical to the operation of an MLP layer.

A similar process can be applied to other architectures such as GIN (Xu et al., 2019), H$_2$GCN (Zhu et al., 2020b), and GAT (Veličković et al., 2018) to reduce them to a standard MLP.

Therefore, following our design in §3, our weak MLP expert is **already consistent with** our strong GNN expert from the perspective of layer architecture.

### C.1.2  RESULTS ON MIXTURE OF SHALLOW-DEEP GCNS

We further construct a variant of `Mowst(*)-GCN` by replacing the weak MLP expert with a shallow GCN. In Table 5, we refer to this variant as "Weak-Strong GCN". The hidden dimension of the two experts in Weak-Strong GCN are the same as that of `Mowst(*)-GCN`. The "weak" expert of the "Weak-Strong GCN" is a 2-layer GCN while the "strong" expert is a 3-layer GCN for `Flickr`, `ogbn-arxiv`, and `ogbn-products`. For `Penn94`, `pokec`, and `twitch-gamer`, the "weak" expert is a 1-layer GCN, and the "strong" expert is a 2-layer GCN. The detailed experimental settings and hyperparameter search methodology are the same as Table 1.

We observe the following from Table 5:

- Adding a weak, shallow GCN to the strong GCN is overall still beneficial.
- The accuracy gains fluctuate across datasets. For example, "Weak-Strong GCN" achieves significant accuracy improvement on `twitch-gamer`, but has accuracy degradation on `Flickr`.
- The overall accuracy gain from "Weak-Strong GCN" is lower than that from the original `Mowst(*)-GCN`.

Table 5: Test set accuracy comparison among the baseline GCN and two variants of `Mowst`. "Weak-Strong GCN" consists of a 2-layer GCN as the weak expert and a 3-layer GCN as the strong expert. For each graph, we highlight the **best** accuracy. The experimental setting and hyperparameter search methodology are the same as Table 1.

| | Flickr | ogbn-products | ogbn-arxiv | Penn94 | pokec | twitch-gamer |
|---|---|---|---|---|---|---|
| MLP | 46.93 ±0.00 | 61.06[†] ±0.08 | 55.50[†] ±0.23 | 73.61[‡] ±0.40 | 62.37[‡] ±0.02 | 60.92[‡] ±0.07 |
| Weak-Strong MLP | 46.87 ±0.16 | 61.24 ±0.10 | 55.98 ±0.10 | 73.78 ±0.32 | 62.44 ±0.05 | 60.68 ±0.12 |
| GCN | 53.86 ±0.37 | 75.64[†] ±0.21 | 71.74[†] ±0.29 | 82.17 ±0.04 | 76.01 ±0.49 | 62.42 ±0.53 |
| Weak-Strong GCN | 54.19 ±0.27 | 74.47 ±0.24 | 71.92 ±0.19 | 82.38 ±0.32 | 76.51 ±0.08 | 63.36 ±0.25 |
| Mowst(*)-GCN | **54.62** ±0.23 | **76.49** ±0.22 | **72.52** ±0.07 | **83.19** ±0.43 | **77.28** ±0.08 | **63.74** ±0.23 |

We perform the following trade-off analysis on the "Weak-Strong GCN" variant. First of all, according to §2, our `Mowst` design is not restricted to a specific expert architecture. Therefore, theoretically "Weak-Strong GCN" still improves the model capacity of the baseline GCN. While upgrading the weak expert from an MLP to a shallow GCN has the potential to further increase the model capacity (*e.g.,* "Weak-Strong GCN" on `twitch-gamer`), `Mowst(*)-GCN` may still be more favorable than "Weak-Strong GCN". We reveal the reasons by analyzing the *specialization* challenge in "Weak-Strong GCN" (see §2.4).

Recall that in the original `Mowst` design, when the two experts specialize, the MLP expert would specialize to nodes with rich self-features but high level of structural noises. The GNN expert would

specialize to nodes with a large amount of useful structure information. Following the above design consideration, in "Weak-Strong GCN", the 2-layer GCN expert will specialize to nodes with rich information in the 2-hop neighborhood, but noisy connections in the 3-hop neighborhood. However, in realistic graphs, a shallow neighborhood (e.g., 2-hop) already contains most of the useful structural information (Zeng et al., 2021) (consequently, the accuracy boost by upgrading a 2-layer GNN to a 3-layer one is much smaller than the boost by upgrading an MLP to a GNN). As a result, the 3-hop noises in a 3-layer GNN can be much less harmful than the 1-hop or 2-hop noises. Thus, it would be challenging for the 2-layer GCN to find nodes that it can specialize on. The intuitive explanation is that, the functionality of the 2-layer GCN expert overlaps significantly with the 3-layer GCN expert, and thus the benefits reduces when mixing two experts that are similar.

WEAK-STRONG *vs.* STRONG-STRONG. We implement a "strong-strong" variant of `Mowst*` consisting of two GNN experts with identical network architecture. Figure 3 shows the convergence curves of the test set accuracy during training, with GCN as the strong experts' architecture. On both `Penn94` and `ogbn-arxiv`, we observe that the convergence quality of the "strong-strong" variant is much worse. Specifically, on `Penn94`, the "strong-strong" model does not identify a suitable collaboration mode between the two GNNs, causing the collapse of the accuracy curve. On `ogbn-arxiv`, the collapse of the "strong-strong" model does not happen. Yet it is still clear that (1) training a

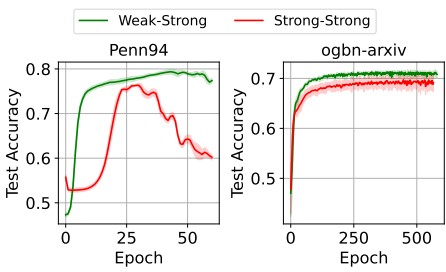

Figure 3: Test accuracy comparison between "weak-strong" and "strong-strong".

"weak-strong" model is more stable, indicated by the much smaller variance, and (2) the accuracy of the "strong-strong" variant is significantly lower. Lastly, note that the $x$-axis denotes number of epochs. The per-epoch execution time for "strong-strong" is also significantly longer, indicating an even longer overall convergence time than the "weak-strong" model.

## C.2 DENOISED FINE-TUNING

In Figure 4, we examine two distinct node groups and visualize their positions in a shared latent space using t-SNE (Van der Maaten & Hinton, 2008) applied to the second-to-last layer of the GNN expert. The first node group comprises nodes that elicit high confidence from the MLP expert (nodes with top-25% $C$). The second group includes the remaining nodes. Green and blue dots represent the initial positions of the group 1 and 2 nodes at the end of training round $r$. Red and orange dots represent the final positions of the group 1 and 2 nodes at the end of training round $r + 10$ (*e.g.,* a node starting from a green dot will end at a red dot). Notably, we observe that a significant portion of nodes remains relatively stable throughout the training process, and thus their start and end positions almost completely overlap. We refrain from visualizing them, as their GNN's predictions

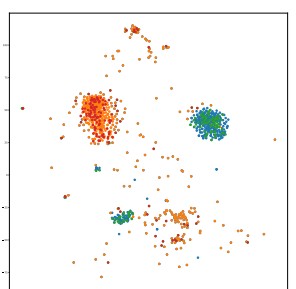

Figure 4: t-SNE visualization on GNN embeddings for `Flickr`.

remain unaltered. Instead, we only visualize nodes whose movement (Euclidean distance) exceeds the $75^{th}$ percentile. In Figure 4, the blue and green nodes are clustered together, indicating that the GNN cannot differentiate them. After the MLP filters out the green nodes of high $C$, the GNN is then able to better optimize the blue nodes even if doing so will increase the GNN loss on the green ones (since the loss on green nodes is down-weighted). Eventually, the GNN pushes the blue nodes to a better position belonging to their true class (the shift of the green nodes is a side effect, as the GNN cannot differentiate the green from the blue ones).

# D PROOF AND DERIVATION

## D.1 PROOFS RELATED TO MODEL OPTIMIZATION

**Definition D.1.** *(Convex function (Bertsekas et al., 2003)) Let $\mathcal{D}$ be a convex subset of $\mathbb{R}^n$. A function $f : \mathcal{D} \mapsto \mathbb{R}$ is convex if $f\left(\lambda \cdot x + (1 - \lambda) \cdot y\right) \leq \lambda \cdot f(x) + (1 - \lambda) \cdot f(y)$, $\forall x, y \in \mathcal{D}$ and $\forall \lambda \in [0, 1]$.*

**Definition D.2.** *(Quasiconvex function (Boyd & Vandenberghe, 2004)) Consider a real-valued function $f$ whose domain $\mathcal{D}$ is convex. Function $f$ is quasiconvex if for any $\gamma \in \mathbb{R}$, its $\gamma$-sublevel set $\{x \in \mathcal{D} \mid f(x) \leq \gamma\}$ is convex. Equivalently, $f$ is quasiconvex if, for all $x, y \in \mathcal{D}$ and $\lambda \in [0, 1]$, we have $f\left(\lambda \cdot x + (1 - \lambda) \cdot y\right) \leq \max\{f(x), f(y)\}$.*

**Definition D.3.** *(Strictly quasiconvex function) A real-valued function $f$ defined on a convex domain $\mathcal{D}$ is strictly quasiconvex if, for all $x, y \in \mathcal{D}$, $x \neq y$ and $\lambda \in (0, 1)$, we have $f\left(\lambda \cdot x + (1 - \lambda) \cdot y\right) < \max\{f(x), f(y)\}$.*

### D.1.1 PROOF OF PROPOSITION 2.2

**Proposition D.4.** *(Originally Proposition 2.2) $C = G \circ D$ is quasiconvex.*

*Proof.* The proof directly follows Agrawal & Boyd (2020), which states that if $g : \mathcal{D} \mapsto \mathbb{R}$ is a quasiconvex function and $h$ is a non-decreasing real-valued function on the real line, then $f = h \circ g$ is quasiconvex. By Definition 2.1, we have $C = G \circ D$ is quasiconvex.

The only thing remains to be shown is the quasiconvexity of typical dispersion functions. We show the convexity of the variance and negative entropy functions by following Definition D.1. Next, we complete the proof since any convex function is also quasiconvex (based on Definitions D.1 and D.2: if $f$ is convex, then $f\left(\lambda x + (1 - \lambda) y\right) \leq \lambda f(x) + (1 - \lambda) f(y) \leq \lambda \max\{f(x), f(y)\} + (1 - \lambda) \max\{f(x), f(y)\} = \max\{f(x), f(y)\}$).

To show the convexity of the dispersion function $D(\boldsymbol{p})$, we first note that its domain is a $(n-1)$-simplex (*i.e.,* $\sum_{1 \leq i \leq n} p_i = 1$ and $p_i \geq 0$ for $i = 1, 2, \ldots, n$), which is a convex set. Next, for specific $D$:

**Variance function** is defined as $D(\boldsymbol{p}) = \sum_{1 \leq i \leq n} \left(p_i - \frac{1}{n}\right)^2$. For two points $\boldsymbol{p}$ and $\boldsymbol{p}'$ and $\lambda \in [0, 1]$, it is easy to show that $D\left(\lambda \boldsymbol{p} + (1 - \lambda) \boldsymbol{p}'\right) \leq \lambda D(\boldsymbol{p}) + (1 - \lambda) D(\boldsymbol{p}')$ (we can follow a similar treatment as Equation 16 by noting that scalar function $d(x) = \left(x - \frac{1}{n}\right)^2$ is convex).

**Negative entropy function** is defined as $D(\boldsymbol{p}) = c + \sum_{1 \leq i \leq n} p_i \cdot \log p_i$, where $c$ is just a constant to satisfy Definition 2.1 that $D\left(\frac{1}{n}\mathbf{1}\right) = 0$. It is a well-known result that the negative entropy is convex. See the proof in JoramSoch (2020). $\qquad\square$

### D.1.2 PROOF OF PROPOSITION 2.3

**Proposition D.5.** *(Originally Proposition 2.3) Given $\boldsymbol{\alpha}_i$, $\hat{L}_{\boldsymbol{\alpha}_i}(\hat{\boldsymbol{p}}_i)$ is a convex function of $\hat{\boldsymbol{p}}_i$ with unique minimizer $\hat{\boldsymbol{p}}_i^* = \boldsymbol{\alpha}_i$. Let $\Delta(\boldsymbol{\alpha}) = \hat{L}_{\boldsymbol{\alpha}}(\boldsymbol{\alpha})$ be a function of $\boldsymbol{\alpha}$, $\Delta$ is a concave function with unique maximizer $\boldsymbol{\alpha}^* = \frac{1}{n}\mathbf{1}$.*

*Proof.* We can formulate the problem of finding the minimizer $\hat{\boldsymbol{p}}_i^*$ of $\hat{L}_{\boldsymbol{\alpha}_i}(\hat{\boldsymbol{p}}_i)$ as

$$\min_{\hat{\boldsymbol{p}}_i} \quad -\sum_{1 \leq j \leq n} \alpha_{ij} \cdot \log \hat{p}_{ij} \tag{8}$$

$$\text{s.t.} \quad \hat{p}_{ij} \geq 0, \text{ for } j = 1, 2, \ldots n \tag{9}$$

$$\sum_{1 \leq j \leq n} \hat{p}_{ij} = 1 \tag{10}$$

The Larangian is

$$\mathcal{L}\left(\hat{\boldsymbol{p}}_i, \gamma, \lambda_1, \lambda_2, ..., \lambda_n\right) = \sum_{1 \leq j \leq n} \alpha_{ij} \cdot \log \hat{p}_{ij} + \gamma \left(\sum_{1 \leq j \leq n} \hat{p}_{ij} - 1\right) + \sum_{1 \leq j \leq n} \lambda_j \hat{p}_{ij} \quad (11)$$

Applying the KKT condition (Bertsekas, 2009), we have

$$\begin{cases} \alpha_{ij} \cdot \dfrac{1}{\hat{p}_{ij}^*} + \gamma^* + \lambda_j^* = 0, & \text{for } j = 1, 2, ..., n \quad \text{(Stationarity)} \\ \lambda_j^* \cdot \hat{p}_{ij}^* = 0, & \text{for } j = 1, 2, ..., n \quad \text{(Complementary slackness)} \\ \hat{p}_{ij}^* \geq 0, & \text{for } j = 1, 2, ..., n \quad \text{(Primal feasibility)} \\ \displaystyle\sum_{1 \leq j \leq n} \hat{p}_{ij}^* = 1 & \text{(Primal feasibility)} \\ \lambda_j^* \geq 0 & \text{for } j = 1, 2, ..., n \quad \text{(Dual feasibility)} \\ \gamma^* \geq 0 & \text{(Dual feasibility)} \end{cases} \quad (12)$$

Solving the system of equations in 12, we have $\hat{p}_{ij}^* = \alpha_{ij} / \left(\sum_{1 \leq j \leq n} \alpha_{ij}\right)$. Since $\boldsymbol{\alpha}_i$ forms a distribution, we further have $\hat{p}_{ij}^* = \alpha_{ij}$, *i.e.*,

$$\hat{\boldsymbol{p}}_i^* = \boldsymbol{\alpha}_i \quad (13)$$

Since $\hat{L}_{\boldsymbol{\alpha}_i}$ is also strictly convex (Proposition D.7), its minimizer is unique.

In particular, $\Delta\left(\boldsymbol{\alpha}_i\right) = \hat{L}_{\boldsymbol{\alpha}_i}\left(\boldsymbol{\alpha}_i\right) = -\sum_{1 \leq j \leq n} \alpha_{ij} \cdot \log \alpha_{ij} = \mathcal{H}\left(\boldsymbol{\alpha}_i\right)$. It is easy to show that the entropy function $\mathcal{H}$ is strictly concave w.r.t. the probability mass function and attains its maximum when $\boldsymbol{\alpha}_i^* = \frac{1}{n}\mathbf{1}$ (JoramSoch, 2020).

$\square$

### D.1.3 PROOFS RELATED TO THEOREM 2.4

We first observe that the optimization problem defined in Equation 2 can be fully decomposed:

$$\min_{\{\hat{\boldsymbol{p}}_i\}} \sum_{1 \leq i \leq m} C\left(\hat{\boldsymbol{p}}_i\right) \cdot \left(\hat{L}_{\boldsymbol{\alpha}_i}\left(\hat{\boldsymbol{p}}_i\right) - \mu_i\right) = \sum_{1 \leq i \leq m} \min_{\hat{\boldsymbol{p}}_i} \left(C\left(\hat{\boldsymbol{p}}_i\right) \cdot \left(\hat{L}_{\boldsymbol{\alpha}_i}\left(\hat{\boldsymbol{p}}_i\right) - \mu_i\right)\right) \quad (14)$$

It is evident that the original optimization problem (Equation 2) can be decomposed into $m$ independent sub-problems:

$$\hat{\boldsymbol{p}}_i^* = \arg\min_{\hat{\boldsymbol{p}}_i} \left(C\left(\hat{\boldsymbol{p}}_i\right) \cdot \left(\hat{L}_{\boldsymbol{\alpha}_i}\left(\hat{\boldsymbol{p}}_i\right) - \mu_i\right)\right), \qquad 1 \leq i \leq m \quad (15)$$

Therefore, in the following, we **omit the subscript** $i$.

**Definition D.6.** *(Extreme points (Bertsekas et al., 2003)) Given a nonempty convex set $\mathcal{D}$, a point $x \in \mathcal{D}$ is an extreme point if there do not exist $y \in \mathcal{D}$ and $z \in \mathcal{D}$ with $y \neq x$ and $z \neq x$, and a scalar $\lambda \in (0, 1)$, such that $x = \lambda y + (1 - \lambda) z$.*

**Proposition D.7.** *The loss function $\hat{L}_{\boldsymbol{\alpha}}\left(\hat{\boldsymbol{p}}\right)$ is strictly convex. For all $\mu$, the sub-level set $\mathcal{L}_{\boldsymbol{\alpha}}^{\leq \mu}$ is convex and compact.*

*Proof.* For strict convexity, first of all, the domain of $\hat{L}_{\boldsymbol{\alpha}}$, the $(n-1)$-simplex, is convex. Then consider two points $\hat{\boldsymbol{p}}$ and $\hat{\boldsymbol{p}}'$ in $\hat{L}_{\boldsymbol{\alpha}}$'s domain and some $\lambda \in (0, 1)$:

$$\hat{L}_{\boldsymbol{\alpha}}\left(\lambda \cdot \hat{\boldsymbol{p}} + (1-\lambda) \cdot \hat{\boldsymbol{p}}'\right) = -\sum_{1 \le i \le n} \alpha_i \cdot \log\left(\lambda \cdot \hat{p}_i + (1-\lambda) \cdot \hat{p}_i'\right)$$

$$\overset{(a)}{<} \sum_{1 \le i \le n} \alpha_i \cdot \left(-\lambda \cdot \log \hat{p}_i - (1-\lambda) \cdot \log \hat{p}_i'\right)$$

$$= \lambda \cdot \left(-\sum_{1 \le i \le n} \alpha_i \cdot \log \hat{p}_i\right) + (1-\lambda) \cdot \left(-\sum_{1 \le i \le n} \alpha_i \cdot \log \hat{p}_i'\right)$$

$$= \lambda \cdot \hat{L}_{\boldsymbol{\alpha}}\left(\hat{\boldsymbol{p}}\right) + (1-\lambda) \cdot \hat{L}_{\boldsymbol{\alpha}}\left(\hat{\boldsymbol{p}}'\right) \tag{16}$$

where "$\overset{(a)}{<}$" is due to the strict convexity of the $\log$ function. This proves that $\hat{L}_{\boldsymbol{\alpha}}$ is strictly convex. Then it directly follows that all its sub-level sets are convex.

Note that $\hat{L}_{\boldsymbol{\alpha}}$ is continuous (and so lower semicontinuous as well). By Proposition 1.2.2 of Bertsekas et al. (2003), all sublevel sets of $\hat{L}_{\boldsymbol{\alpha}}$ are closed. Further, since $\hat{L}_{\boldsymbol{\alpha}}$ is defined on the $(n-1)$-simplex, $\mathcal{L}_{\boldsymbol{\alpha}}^{\le \mu}$ is also bounded. Therefore, $\mathcal{L}_{\boldsymbol{\alpha}}^{\le \mu}$ is compact (compact means closed and bounded). $\qquad \square$

**Proposition D.8.** *The set of extreme points of $\mathcal{L}_{\boldsymbol{\alpha}}^{\le \mu}$ equals $\mathcal{L}_{\boldsymbol{\alpha}}^{\mu}$.*

*Proof.* Denote $\mathcal{P}$ as the set of extreme points of $\mathcal{L}_{\boldsymbol{\alpha}}^{\le \mu}$. We first show that $\mathcal{L}_{\boldsymbol{\alpha}}^{\mu} \subseteq \mathcal{P}$. Consider a point $\hat{\boldsymbol{p}}_0 \in \mathcal{L}_{\boldsymbol{\alpha}}^{\mu}$. Suppose there exist some other points $\hat{\boldsymbol{p}}_1, \hat{\boldsymbol{p}}_2 \in \mathcal{L}_{\boldsymbol{\alpha}}^{\le \mu}$ (with $\hat{\boldsymbol{p}}_1 \ne \hat{\boldsymbol{p}}_0$ and $\hat{\boldsymbol{p}}_2 \ne \hat{\boldsymbol{p}}_0$), such that $\lambda \hat{\boldsymbol{p}}_1 + (1-\lambda) \hat{\boldsymbol{p}}_2 = \hat{\boldsymbol{p}}_0$ for some $\lambda \in (0,1)$. By strict convexity shown in Proposition D.7, we have $\mu = \hat{L}_{\boldsymbol{\alpha}}\left(\hat{\boldsymbol{p}}_0\right) = \hat{L}_{\boldsymbol{\alpha}}\left(\lambda \hat{\boldsymbol{p}}_1 + (1-\lambda) \hat{\boldsymbol{p}}_2\right) < \lambda \hat{L}_{\boldsymbol{\alpha}}\left(\hat{\boldsymbol{p}}_1\right) + (1-\lambda) \hat{L}_{\boldsymbol{\alpha}}\left(\hat{\boldsymbol{p}}_2\right)$. This implies that $\hat{L}_{\boldsymbol{\alpha}}\left(\hat{\boldsymbol{p}}_1\right) > \mu$ or $\hat{L}_{\boldsymbol{\alpha}}\left(\hat{\boldsymbol{p}}_2\right) > \mu$, which contradicts with the condition that $\hat{\boldsymbol{p}}_1, \hat{\boldsymbol{p}}_2 \in \mathcal{L}_{\boldsymbol{\alpha}}^{\le \mu}$. Thus, any point $\hat{\boldsymbol{p}}_0 \in \mathcal{L}_{\boldsymbol{\alpha}}^{\mu}$ is an extreme point of $\mathcal{L}_{\boldsymbol{\alpha}}^{\le \mu}$.

Now we show $\mathcal{P} \subseteq \mathcal{L}_{\boldsymbol{\alpha}}^{\mu}$. Consider a point $\hat{\boldsymbol{p}}_0 \in \mathcal{L}_{\boldsymbol{\alpha}}^{< \mu}$. For any $\mu \in \mathbb{R}$, we always have $\hat{p}_{0i} > 0$, for all $i = 1, 2, \ldots, n$ (otherwise, $\hat{L}_{\boldsymbol{\alpha}}\left(\hat{\boldsymbol{p}}_0\right) \to \infty$). This means that we can find a vector $\boldsymbol{\epsilon}$, such that (1) $\sum_{1 \le i \le n} \epsilon_i = 0$, (2) $|\epsilon_i| < \hat{p}_{0i}$, for all $1 \le i \le n$, and (3) $\hat{L}_{\boldsymbol{\alpha}}\left(\hat{\boldsymbol{p}}_0 + \boldsymbol{\epsilon}\right) \le \mu$ and $\hat{L}_{\boldsymbol{\alpha}}\left(\hat{\boldsymbol{p}}_0 - \boldsymbol{\epsilon}\right) \le \mu$. Point (1) and (2) ensure that $\hat{\boldsymbol{p}}_0 + \boldsymbol{\epsilon}$ and $\hat{\boldsymbol{p}}_0 - \boldsymbol{\epsilon}$ are both with the domain of $\hat{L}_{\boldsymbol{\alpha}}$. For Point (3), we can always find small enough $\epsilon_i$ due to the continuity of the loss function $\hat{L}_{\boldsymbol{\alpha}}$. We thus find two points, $\hat{\boldsymbol{p}}_1 = \hat{\boldsymbol{p}}_0 + \boldsymbol{\epsilon} \in \mathcal{L}_{\boldsymbol{\alpha}}^{\le \mu}$ and $\hat{\boldsymbol{p}}_2 = \hat{\boldsymbol{p}}_0 - \boldsymbol{\epsilon} \in \mathcal{L}_{\boldsymbol{\alpha}}^{< \mu}$, where $\hat{\boldsymbol{p}}_0 = \frac{1}{2}\hat{\boldsymbol{p}}_1 + \frac{1}{2}\hat{\boldsymbol{p}}_2$. Thus, $\hat{\boldsymbol{p}}_0 \in \mathcal{L}_{\boldsymbol{\alpha}}^{< \mu}$ is not an extreme point of $\mathcal{L}_{\boldsymbol{\alpha}}^{\le \mu}$.

In summary, $\mathcal{P} = \mathcal{L}_{\boldsymbol{\alpha}}^{\mu}$. $\qquad \square$

**Proposition D.9.** *(Krein-Milman Theorem (Bertsekas et al., 2003)) Let $\mathcal{D}$ be a nonempty convex subset of $\mathbb{R}^n$. If $\mathcal{D}$ is compact, then $\mathcal{D}$ is equal to the convex hull of its extreme points.*

**Proposition D.10.** *(Caratheodory's Theorem (Bertsekas, 2009)) Let $\mathcal{D}$ be a nonempty subset of $\mathbb{R}^n$. Every point from the convex hull of $\mathcal{D}$ can be represented as a convex combination of $x_1, \ldots, x_m$ from $\mathcal{D}$, where $m \le n+1$ and $x_2 - x_1, \ldots, x_m - x_1$ are linearly independent.*

**Proposition D.11.** *If $f$ is a quasiconvex function defined on $\mathcal{D}$, then for all $x_1, \ldots, x_m \in \mathcal{D}$ and $\lambda_1 \ge 0, \ldots, \lambda_m \ge 0$ such that $\sum_{1 \le i \le m} \lambda_i = 1$, we have $f\left(\sum_{1 \le i \le m} \lambda_i x_i\right) \le \max_{1 \le i \le m}\{f(x_i)\}$.*

**Proposition D.12.** *If $f$ is a strictly quasiconvex function defined on $\mathcal{D}$, then for (1) $x_1, \ldots, x_m \in \mathcal{D}$ such that $x_1 \ne x_2 \ldots \ne x_m$ and $x_2 - x_1, \ldots, x_m - x_1$ are linearly independent, and (2) $\lambda_1 > 0, \ldots, \lambda_m > 0$ such that $\sum_{1 \le i \le m} \lambda_i = 1$, we have $f\left(\sum_{1 \le i \le m} \lambda_i x_i\right) < \max_{1 \le i \le m}\{f(x_i)\}$.*

*Proof.* For brevity, we only show the proof for Proposition D.12 as the proof for Proposition D.11 is similar (and easier). We prove by induction. The base case of $m = 2$ automatically holds by Definition D.2.

For $m > 2$, note that

$$\sum_{1 \le i \le m} \lambda_i x_i = \lambda_1 x_1 + \left( \sum_{2 \le i \le m} \lambda_i \right) \cdot \left( \sum_{2 \le j \le m} \frac{\lambda_j}{\sum_{2 \le i \le m} \lambda_i} x_j \right) \tag{17}$$

Let $\lambda'_j = \frac{\lambda_j}{\sum_{2 \le i \le m} \lambda_i}$. Apparently, $\sum_{2 \le j \le m} \lambda'_j = 1$ and $\lambda'_j > 0$ for all $2 \le j \le m$. In addition, $x_3 - x_2, \ldots, x_m - x_2$ are linearly independent (To show this: $\sum_{2 \le j \le m} \beta_j (x_j - x_1) = \sum_{3 \le j \le m} \beta_j (x_j - x_2) + \left( \sum_{2 \le j \le m} \beta_j \right) (x_2 - x_1)$. If $x_3 - x_2, \ldots, x_m - x_2$ are linearly dependent, then there exists $\beta_3, \ldots, \beta_m$ such that $\sum_{3 \le j \le m} \beta_j (x_j - x_2) = 0$. Then by letting $\beta_2 = - \sum_{3 \le j \le m} \beta_j$, we have $\sum_{2 \le j \le m} \beta_j (x_j - x_1) = 0$ – contradiction with the hypothesis that $x_2 - x_1, \ldots, x_m - x_1$ are linearly independent). Therefore, by induction hypothesis, we have

$$f \left( \sum_{2 \le j \le m} \lambda'_j x_j \right) < \max_{2 \le j \le m} \{ f(x_j) \} \tag{18}$$

Let $x' = \sum_{2 \le j \le m} \lambda'_j x_j$. Since $x_2 - x_1, \ldots, x_m - x_1$ are linearly independent, we must have $\sum_{2 \le j \le m} \lambda'_j (x_j - x_1) \ne 0$. Equivalently, $\sum_{2 \le j \le m} \lambda'_j x_j = x' \ne x_1$. Thus,

$$f \left( \sum_{1 \le i \le m} \lambda_i x_i \right) = f \left( \lambda_1 x_1 + (1 - \lambda_1) x' \right) < \max \{ f(x_1), f(x') \} \tag{19}$$

$$< \max \left\{ f(x_1), \max_{2 \le i \le m} \{ f(x_i) \} \right\} \tag{20}$$

$$= \max_{1 \le i \le m} \{ f(x_i) \} \tag{21}$$

This completes the induction process and thus concludes the proof. $\qquad\square$

**Theorem D.13.** *(Originally Theorem 2.4) Suppose $C = G \circ D$ follows Definition 2.1. Denote $\mathcal{L}^\mu_{\boldsymbol{\alpha}} = \{ \hat{\boldsymbol{p}} \mid \hat{L}_{\boldsymbol{\alpha}} (\hat{\boldsymbol{p}}) = \mu \}$ and $\mathcal{L}^{<\mu}_{\boldsymbol{\alpha}} = \{ \hat{\boldsymbol{p}} \mid \hat{L}_{\boldsymbol{\alpha}} (\hat{\boldsymbol{p}}) < \mu \}$ as the level set and strict sublevel set of $\hat{L}_{\boldsymbol{\alpha}}$. Denote $\mathcal{C}^{<\mu} = \{ \hat{\boldsymbol{p}} \mid C (\hat{\boldsymbol{p}}) < \mu \}$ as the strict sublevel set of $C$. For a given $\boldsymbol{\alpha} \ne \frac{1}{n} \mathbf{1}_n$, the minimizer $\hat{\boldsymbol{p}}^*$ satisfies:*

- *If $\Delta (\boldsymbol{\alpha}) > \mu$, then $\hat{\boldsymbol{p}}^* = \frac{1}{n} \mathbf{1}_n$ and $C (\hat{\boldsymbol{p}}^*) = 0$.*

- *If $\Delta (\boldsymbol{\alpha}) = \mu$, then $\hat{\boldsymbol{p}}^* = \boldsymbol{\alpha}$ or $\hat{\boldsymbol{p}}^* = \frac{1}{n} \mathbf{1}$.*

- *If $\Delta (\boldsymbol{\alpha}) < \mu$, then $\hat{\boldsymbol{p}}^* \in \mathcal{L}^{<\mu}_{\boldsymbol{\alpha}} - \mathcal{C}^{<\mu'}$ where $\mu' = C (\boldsymbol{\alpha}) \le C (\hat{\boldsymbol{p}}^*) \le G \left( \max_{\hat{\boldsymbol{p}} \in \mathcal{L}^\mu_{\boldsymbol{\alpha}}} D (\hat{\boldsymbol{p}}) \right)$. Further, there exists $C$ such that $\hat{\boldsymbol{p}}^* = \boldsymbol{\alpha}$, or $\hat{\boldsymbol{p}}^*$ is sufficiently close to the level set $\mathcal{L}^\mu_{\boldsymbol{\alpha}}$.*

*Proof.* We separately consider the three cases listed by Theorem 2.4.

**Case 1.** By Proposition 2.3, we have $\hat{L}_{\boldsymbol{\alpha}} (\hat{\boldsymbol{p}}) - \mu \ge \min_{\hat{\boldsymbol{p}}} \left( \hat{L}_{\boldsymbol{\alpha}} (\hat{\boldsymbol{p}}) - \mu \right) = \hat{L}_{\boldsymbol{\alpha}} (\boldsymbol{\alpha}) - \mu = \Delta (\boldsymbol{\alpha}) - \mu > 0$ for all $\hat{\boldsymbol{p}}$. By Definition 2.1, we have $C (\hat{\boldsymbol{p}}) \ge 0$ for all $\hat{\boldsymbol{p}}$. We can thus derive the following bound: $C (\hat{\boldsymbol{p}}^*) \cdot \left( \hat{L}_{\boldsymbol{\alpha}} (\hat{\boldsymbol{p}}^*) - \mu \right) = \min_{\hat{\boldsymbol{p}}} \left( C (\hat{\boldsymbol{p}}) \cdot \left( \hat{L}_{\boldsymbol{\alpha}} (\hat{\boldsymbol{p}}) - \mu \right) \right) \ge 0$. Further, since $\hat{L}_{\boldsymbol{\alpha}} (\hat{\boldsymbol{p}}) - \mu > 0$ for all $\hat{\boldsymbol{p}}$, the lower bound $C (\hat{\boldsymbol{p}}^*) \cdot \left( \hat{L}_{\boldsymbol{\alpha}} (\hat{\boldsymbol{p}}^*) - \mu \right) = 0$ is achieved if and only if $C (\hat{\boldsymbol{p}}^*) = 0$. By Definition 2.1, the minimizer $\hat{\boldsymbol{p}}^* = \frac{1}{n} \mathbf{1}_n$.

**Case 2.** The reasoning is similar to Case 1. We can derive $C (\hat{\boldsymbol{p}}^*) \cdot \left( \hat{L}_{\boldsymbol{\alpha}} (\hat{\boldsymbol{p}}^*) - \mu \right) \ge 0$. Now to achieve the lower bound $C (\hat{\boldsymbol{p}}^*) \cdot \left( \hat{L}_{\boldsymbol{\alpha}} (\hat{\boldsymbol{p}}^*) - \mu \right) = 0$, we either need $C (\hat{\boldsymbol{p}}^*) = 0$ or $\hat{L}_{\boldsymbol{\alpha}} (\hat{\boldsymbol{p}}^*) - \mu = 0$, which means either $\hat{\boldsymbol{p}}^* = \frac{1}{n} \mathbf{1}_n$ or $\hat{\boldsymbol{p}}^* = \boldsymbol{\alpha}$.

**Case 3.** First, observe that $C\left(\hat{\boldsymbol{p}}^*\right) \cdot \left(\hat{L}_{\boldsymbol{\alpha}}\left(\hat{\boldsymbol{p}}^*\right) - \mu\right) = \min_{\hat{\boldsymbol{p}}} \left(C\left(\hat{\boldsymbol{p}}\right) \cdot \left(\hat{L}_{\boldsymbol{\alpha}}\left(\hat{\boldsymbol{p}}\right) - \mu\right)\right) \overset{\hat{\boldsymbol{p}}=\boldsymbol{\alpha}}{\leq} C\left(\boldsymbol{\alpha}\right) \cdot \left(\hat{L}_{\boldsymbol{\alpha}}\left(\boldsymbol{\alpha}\right) - \mu\right) = C\left(\boldsymbol{\alpha}\right) \cdot \left(\Delta\left(\boldsymbol{\alpha}\right) - \mu\right) < 0$ (note that we assume $\boldsymbol{\alpha} \neq \frac{1}{n}\mathbf{1}_n$). Since $C\left(\hat{\boldsymbol{p}}\right) \geq 0$ for all $\hat{\boldsymbol{p}}$, this implies that

$$C\left(\hat{\boldsymbol{p}}^*\right) > 0 \tag{22}$$

$$\hat{L}_{\boldsymbol{\alpha}}\left(\hat{\boldsymbol{p}}^*\right) - \mu < 0 \tag{23}$$

By Inequality 23, we have $\hat{\boldsymbol{p}}^* \in \mathcal{L}_{\boldsymbol{\alpha}}^{<\mu}$.

We can further constrain $\hat{\boldsymbol{p}}^*$ such that $C\left(\hat{\boldsymbol{p}}^*\right) \geq C\left(\boldsymbol{\alpha}\right)$: Suppose otherwise that $0 \leq C\left(\hat{\boldsymbol{p}}^*\right) < C\left(\boldsymbol{\alpha}\right)$. By Proposition 2.3, we have $\hat{L}_{\boldsymbol{\alpha}}\left(\hat{\boldsymbol{p}}^*\right) - \mu \geq \Delta\left(\boldsymbol{\alpha}\right) - \mu \quad \Rightarrow \quad -\left(\hat{L}_{\boldsymbol{\alpha}}\left(\hat{\boldsymbol{p}}^*\right) - \mu\right) \leq -\left(\Delta\left(\boldsymbol{\alpha}\right) - \mu\right)$. Combining the two inequalities together, we have $-C\left(\hat{\boldsymbol{p}}^*\right) \cdot \left(\hat{L}_{\boldsymbol{\alpha}}\left(\hat{\boldsymbol{p}}^*\right) - \mu\right) < -C\left(\boldsymbol{\alpha}\right) \cdot \left(\Delta\left(\boldsymbol{\alpha}\right) - \mu\right) \quad \Rightarrow \quad C\left(\hat{\boldsymbol{p}}^*\right) \cdot \left(\hat{L}_{\boldsymbol{\alpha}}\left(\hat{\boldsymbol{p}}^*\right) - \mu\right) > C\left(\boldsymbol{\alpha}\right) \cdot \left(\Delta\left(\boldsymbol{\alpha}\right) - \mu\right)$. This means that $\hat{\boldsymbol{p}}^*$ is not a minimizer – a contradiction. In other words, we must have $\hat{\boldsymbol{p}}^* \notin \mathcal{C}^{<\mu'}$ where $\mu' = C\left(\boldsymbol{\alpha}\right)$.

So far, we have proven $\hat{\boldsymbol{p}}^* \in \mathcal{L}_{\boldsymbol{\alpha}}^{<\mu} - \mathcal{C}^{<\mu'}$ where $\mu' = C\left(\boldsymbol{\alpha}\right) \leq C\left(\hat{\boldsymbol{p}}^*\right)$. Next, we further upper-bound the range of the confidence corresponding to $\hat{\boldsymbol{p}}^*$. Note that

$$C\left(\hat{\boldsymbol{p}}^*\right) \leq \sup_{\hat{\boldsymbol{p}} \in \mathcal{L}_{\boldsymbol{\alpha}}^{<\mu} - \mathcal{C}^{<\mu'}} C\left(\hat{\boldsymbol{p}}\right) \overset{(a)}{\leq} \sup_{\hat{\boldsymbol{p}} \in \mathcal{L}_{\boldsymbol{\alpha}}^{\leq\mu}} C\left(\hat{\boldsymbol{p}}\right) \overset{(b)}{=} G\left(\max_{\hat{\boldsymbol{p}} \in \mathcal{L}_{\boldsymbol{\alpha}}^{\leq\mu}} D\left(\hat{\boldsymbol{p}}\right)\right) \overset{(c)}{=} G\left(\max_{\hat{\boldsymbol{p}} \in \mathcal{L}_{\boldsymbol{\alpha}}^{\mu}} D\left(\hat{\boldsymbol{p}}\right)\right) \tag{24}$$

For "$\overset{(a)}{\leq}$" above, the reason is that $\mathcal{L}_{\boldsymbol{\alpha}}^{<\mu} - \mathcal{C}^{<\mu'} \subseteq \mathcal{L}_{\boldsymbol{\alpha}}^{\leq\mu}$. For "$\overset{(b)}{=}$", let's first consider $\sup_{\mathcal{L}_{\boldsymbol{\alpha}}^{\leq\mu}} D\left(\hat{\boldsymbol{p}}\right)$. By Proposition D.7, $\mathcal{L}_{\boldsymbol{\alpha}}^{\leq\mu}$ is compact. By Definition 2.1, $D$ is continuous. So by the extreme value theorem (Murphy, 2008), $D$ attains its maximum value, *i.e.*, there exists $\hat{\boldsymbol{p}}' \in \mathcal{L}_{\boldsymbol{\alpha}}^{\leq\mu}$ such that $D\left(\hat{\boldsymbol{p}}'\right) = \sup_{\mathcal{L}_{\boldsymbol{\alpha}}^{\leq\mu}} D\left(\hat{\boldsymbol{p}}\right) = \max_{\mathcal{L}_{\boldsymbol{\alpha}}^{\leq\mu}} D\left(\hat{\boldsymbol{p}}\right)$. In other words, $D\left(\hat{\boldsymbol{p}}'\right) \geq D\left(\hat{\boldsymbol{p}}\right)$ for all $\hat{\boldsymbol{p}} \in \mathcal{L}_{\boldsymbol{\alpha}}^{\leq\mu}$. Now since $G$ is monotonically non-decreasing (Definition 2.1), we have $G\left(D\left(\hat{\boldsymbol{p}}'\right)\right) \geq G\left(D\left(\hat{\boldsymbol{p}}\right)\right)$ for all $\hat{\boldsymbol{p}} \in \mathcal{L}_{\boldsymbol{\alpha}}^{\leq\mu}$. This means $C = G \circ D$ attains its maximum at $\hat{\boldsymbol{p}}'$ and $\sup_{\hat{\boldsymbol{p}} \in \mathcal{L}_{\boldsymbol{\alpha}}^{\leq\mu}} C\left(\hat{\boldsymbol{p}}\right) = \max_{\hat{\boldsymbol{p}} \in \mathcal{L}_{\boldsymbol{\alpha}}^{\leq\mu}} C\left(\hat{\boldsymbol{p}}\right) = G\left(\max_{\hat{\boldsymbol{p}} \in \mathcal{L}_{\boldsymbol{\alpha}}^{\leq\mu}} D\left(\hat{\boldsymbol{p}}\right)\right)$. Finally, for "$\overset{(c)}{=}$", we need to show $\max_{\hat{\boldsymbol{p}} \in \mathcal{L}_{\boldsymbol{\alpha}}^{\leq\mu}} D\left(\hat{\boldsymbol{p}}\right) = \max_{\hat{\boldsymbol{p}} \in \mathcal{L}_{\boldsymbol{\alpha}}^{\mu}} D\left(\hat{\boldsymbol{p}}\right)$. By Propositions D.7, D.8 and D.9, $\mathcal{L}_{\boldsymbol{\alpha}}^{\leq\mu}$ is the convex hull of $\mathcal{L}_{\boldsymbol{\alpha}}^{\mu}$. So by Proposition D.10, for any point $\hat{\boldsymbol{p}}_0 \in \mathcal{L}_{\boldsymbol{\alpha}}^{\leq\mu}$, we can find $\hat{\boldsymbol{p}}_1, \ldots, \hat{\boldsymbol{p}}_m \in \mathcal{L}_{\boldsymbol{\alpha}}^{\mu}$ (with $m \leq n + 1$), such that $\hat{\boldsymbol{p}}_0 = \sum_{1 \leq i \leq m} \lambda_i \hat{\boldsymbol{p}}_i$ (with $\sum_{1 \leq i \leq m} \lambda_i = 1$ and $\lambda_i \geq 0$ for $i = 1, \ldots, m$). By Proposition D.11, we have $D\left(\hat{\boldsymbol{p}}_0\right) \leq \max_{1 \leq i \leq m}\{D\left(\hat{\boldsymbol{p}}_i\right)\} \leq \max_{\hat{\boldsymbol{p}} \in \mathcal{L}_{\boldsymbol{\alpha}}^{\mu}} D\left(\hat{\boldsymbol{p}}\right)$. Thus, $\max_{\hat{\boldsymbol{p}} \in \mathcal{L}_{\boldsymbol{\alpha}}^{\leq\mu}} D\left(\hat{\boldsymbol{p}}\right) = \max_{\hat{\boldsymbol{p}} \in \mathcal{L}_{\boldsymbol{\alpha}}^{\mu}} D\left(\hat{\boldsymbol{p}}\right)$. This completes the proof for Equation 24. Thus, we have proven the bound on both the minimizer $\hat{\boldsymbol{p}}^*$ and its corresponding $C\left(\hat{\boldsymbol{p}}^*\right)$.

**Case 3: Tightness of the bound.** From the above proof, we know that when $\Delta\left(\boldsymbol{\alpha}\right) < \mu$, the possible positions that the minimizer $\hat{\boldsymbol{p}}^*$ can take are determined by the two level sets $\mathcal{C}^{<\mu'}$ ($\mu' = C\left(\boldsymbol{\alpha}\right)$) and $\mathcal{L}_{\boldsymbol{\alpha}}^{<\mu}$. We now show that the bound on $\hat{\boldsymbol{p}}^*$ is tight since $\hat{\boldsymbol{p}}^*$ can be close to the two level sets under appropriate $C = G \circ D$ function. Note that our proof is based on $C$ defined by 2.1, meaning that $C$ does not need to be continuous.

First, we construct a $C$ such that $\hat{\boldsymbol{p}}^* = \boldsymbol{\alpha}$. Such a $C$ can be defined by a simple $G$ for any $D$:

$$G(x) = \begin{cases} 0, & \text{when } x \leq 0 \\ 1, & \text{when } x > 0 \end{cases} \tag{25}$$

Since $\boldsymbol{\alpha} \neq \frac{1}{n}\mathbf{1}_n$, we have $D\left(\boldsymbol{\alpha}\right) > 0$ and thus $C\left(\boldsymbol{\alpha}\right) = G\left(D\left(\boldsymbol{\alpha}\right)\right) = 1$. By conditions 22, 23, we have $C\left(\hat{\boldsymbol{p}}^*\right) \cdot \left(\hat{L}_{\boldsymbol{\alpha}}\left(\hat{\boldsymbol{p}}^*\right) - \mu\right) \geq \max C\left(\hat{\boldsymbol{p}}\right) \cdot \min\left(\hat{L}_{\boldsymbol{\alpha}}\left(\hat{\boldsymbol{p}}\right) - \mu\right) = 1 \cdot \left(\Delta\left(\boldsymbol{\alpha}\right) - \mu\right) = \Delta\left(\boldsymbol{\alpha}\right) - \mu$.

In addition, since the problem $\min\left(\hat{L}_{\boldsymbol{\alpha}}\left(\hat{\boldsymbol{p}}\right) - \mu\right)$ has a unique minimizer at $\boldsymbol{\alpha}$ (Proposition 2.3), and $C\left(\boldsymbol{\alpha}\right) = \max C\left(\hat{\boldsymbol{p}}\right) = 1$, we know that under $C$ defined by Equation 25, we have a unique minimizer $\hat{\boldsymbol{p}}^* = \boldsymbol{\alpha}$ for the overall optimization problem 15.

Next, we construct a $C$ such that $\hat{\boldsymbol{p}}^*$ is close to $\hat{L}_{\boldsymbol{\alpha}}^{\mu}$. We first define the distance between a point $\hat{\boldsymbol{p}}$ and the level set $\mathcal{L}_{\boldsymbol{\alpha}}^{\mu}$ as:

$$\texttt{dist}\left(\hat{\boldsymbol{p}}, \mathcal{L}_{\boldsymbol{\alpha}}^{\mu}\right) = \min_{\hat{\boldsymbol{p}}' \in \mathcal{L}_{\boldsymbol{\alpha}}^{\mu}} \|\hat{\boldsymbol{p}} - \hat{\boldsymbol{p}}'\| \tag{26}$$

Thus, we want to show that there exists a $C$, such that $\texttt{dist}\left(\hat{\boldsymbol{p}}^*, \mathcal{L}_{\boldsymbol{\alpha}}^{\mu}\right) < \epsilon$ for any $\epsilon > 0$.

Consider another sub-level set $\mathcal{L}_{\boldsymbol{\alpha}}^{\leq \mu-\eta}$ for some $\eta > 0$ (we also let $\eta$ to be small enough so that $\mathcal{L}_{\boldsymbol{\alpha}}^{\leq \mu-\eta}$ is non-empty). We have shown before (by combining Propositions D.7, D.8, D.9 and D.10) that any point $\hat{\boldsymbol{p}}' \in \mathcal{L}_{\boldsymbol{\alpha}}^{<\mu-\eta}$ can be expressed as a convex combination of $\hat{\boldsymbol{p}}_1, \ldots, \hat{\boldsymbol{p}}_m$, where $m \leq n+1$, $\hat{\boldsymbol{p}}_i \in \mathcal{L}_{\boldsymbol{\alpha}}^{\mu-\eta}$ and $\hat{\boldsymbol{p}}_2 - \hat{\boldsymbol{p}}_1, \ldots, \hat{\boldsymbol{p}}_m - \hat{\boldsymbol{p}}_1$ are linearly independent. Since $\hat{\boldsymbol{p}}'$ is in the strict sub-level set and $\hat{\boldsymbol{p}}_i$ are in the sub-level set, we have $\hat{\boldsymbol{p}}' \neq \hat{\boldsymbol{p}}_i$, and thus we can write $\hat{\boldsymbol{p}}' = \sum_{1 \leq i \leq m} \lambda_i \hat{\boldsymbol{p}}_i$ where $\lambda_i > 0$ and $\sum_{1 \leq i \leq m} \lambda_i = 1$.

Suppose $D$ is *strictly quasiconvex*. We can now apply Proposition D.12, such that

$$D\left(\hat{\boldsymbol{p}}'\right) = D\left(\sum_{1 \leq i \leq m} \lambda_i \hat{\boldsymbol{p}}_i\right) < \max_{1 \leq i \leq m}\left\{D\left(\hat{\boldsymbol{p}}_i\right)\right\} \tag{27}$$

The strict "$<$" means that $\mathcal{L}_{\boldsymbol{\alpha}}^{<\mu-\eta}$ cannot contain any maximizer of the optimization problem "$\max_{\hat{\boldsymbol{p}} \in \mathcal{L}_{\boldsymbol{\alpha}}^{\leq \mu-\eta}} D\left(\hat{\boldsymbol{p}}\right)$" (note that previously, we have only shown $\mathcal{L}_{\boldsymbol{\alpha}}^{\mu-\eta}$ contains the maximizer, but it is also possible for $\mathcal{L}_{\boldsymbol{\alpha}}^{<\mu-\eta}$ to contain the maximizer if $D$ is not strictly quasiconvex). Now, define $D_{\max} := \max_{\hat{\boldsymbol{p}} \in \mathcal{L}_{\boldsymbol{\alpha}}^{\leq \mu-\eta}} D\left(\hat{\boldsymbol{p}}\right) > 0$ and the corresponding maximizer (or one of the maximizers) as $\hat{\boldsymbol{p}}_{\eta}^* \in \mathcal{L}_{\boldsymbol{\alpha}}^{\mu-\eta}$. For all $\hat{\boldsymbol{p}} \in \mathcal{L}_{\boldsymbol{\alpha}}^{<\mu-\eta}$, we have $D\left(\hat{\boldsymbol{p}}\right) < D_{\max}$.

Consider a $G$ function defined as follows:

$$G = \begin{cases} 0, & \text{when } x \leq 0 \\ \beta, & \text{when } 0 < x < D_{\max} \\ 1, & \text{when } x \geq D_{\max} \end{cases} \tag{28}$$

By definition, $\hat{L}_{\boldsymbol{\alpha}}\left(\hat{\boldsymbol{p}}_{\eta}^*\right) = \mu - \eta > 0$. Also, $C\left(\hat{\boldsymbol{p}}_{\eta}^*\right) = G\left(D\left(\hat{\boldsymbol{p}}_{\eta}^*\right)\right) = 1$. As a result, $C\left(\hat{\boldsymbol{p}}_{\eta}^*\right) \cdot \left(\hat{L}_{\boldsymbol{\alpha}}\left(\hat{\boldsymbol{p}}_{\eta}^*\right) - \mu\right) = -\eta$. For any point $\hat{\boldsymbol{p}} \in \mathcal{L}_{\boldsymbol{\alpha}}^{<\mu-\eta}$, we have $C\left(\hat{\boldsymbol{p}}\right) = G\left(D\left(\hat{\boldsymbol{p}}\right)\right) = \beta$. Consequently, $\min_{\hat{\boldsymbol{p}} \in \mathcal{L}_{\boldsymbol{\alpha}}^{<\mu-\eta}} C\left(\hat{\boldsymbol{p}}\right) \cdot \left(\hat{L}_{\boldsymbol{\alpha}}\left(\hat{\boldsymbol{p}}\right) - \mu\right) = \beta \min_{\hat{\boldsymbol{p}} \in \mathcal{L}_{\boldsymbol{\alpha}}^{<\mu-\eta}}\left(\hat{L}_{\boldsymbol{\alpha}}\left(\hat{\boldsymbol{p}}\right) - \mu\right) = \beta\left(\Delta\left(\boldsymbol{\alpha}\right) - \mu\right)$. Therefore, as long as $0 < \beta < \frac{\eta}{\mu - \Delta(\boldsymbol{\alpha})}$, the minimizer $\hat{\boldsymbol{p}}^*$ of the original optimization problem 15 must satisfy $\hat{\boldsymbol{p}}^* \in \mathcal{L}_{\boldsymbol{\alpha}}^{<\mu} - \mathcal{L}_{\boldsymbol{\alpha}}^{<\mu-\eta}$ (or more precisely, $\hat{\boldsymbol{p}}^* \in \mathcal{L}_{\boldsymbol{\alpha}}^{<\mu} - \mathcal{L}_{\boldsymbol{\alpha}}^{<\mu-\eta} - C^{<\mu'}$).

Now the last issue is to determine the relationship between $\eta$ and $\epsilon$. We know that $\mu - \eta \leq \hat{L}_{\boldsymbol{\alpha}}\left(\hat{\boldsymbol{p}}^*\right) < \mu$. We utilize the idea of *path integral*. We first note that the domain of $\hat{L}_{\boldsymbol{\alpha}}\left(\hat{\boldsymbol{p}}\right)$ is the $(n-1)$-simplex, which lies within a hyperplane of $\mathbb{R}^n$. It is easy to show that the normal vector of such a hyperplane is parallel to $\mathbf{1}$. For a point $\hat{\boldsymbol{p}}$, denote $\boldsymbol{\pi}\left(\hat{\boldsymbol{p}}\right)$ as the vector by projecting the gradient $\nabla \hat{L}_{\boldsymbol{\alpha}}\left(\hat{\boldsymbol{p}}\right)$ onto the hyperplane. With some basic calculation, we derive that

$$\boldsymbol{\pi}\left(\hat{\boldsymbol{p}}\right)=\begin{bmatrix} -\frac{\alpha_1}{\hat{p}_1}+\frac{1}{n}\sum_{1\leq j\leq n}\frac{\alpha_j}{\hat{p}_j} \\ \vdots \\ -\frac{\alpha_n}{\hat{p}_n}+\frac{1}{n}\sum_{1\leq j\leq n}\frac{\alpha_j}{\hat{p}_j} \end{bmatrix} \tag{29}$$

$$=\begin{bmatrix} \left(\frac{1}{n}-1\right)\alpha_1 & \alpha_2 & \dots & \alpha_n \\ \alpha_1 & \left(\frac{1}{n}-1\right)\alpha_2 & \dots & \alpha_n \\ \vdots & & \ddots & \vdots \\ \alpha_1 & \alpha_2 & \dots & \left(\frac{1}{n}-1\right)\alpha_n \end{bmatrix}\cdot\begin{bmatrix} \hat{p}_1 \\ \hat{p}_2 \\ \vdots \\ \hat{p}_n \end{bmatrix} \tag{30}$$

The linear system of equations $\boldsymbol{\pi}\left(\hat{\boldsymbol{p}}\right)=\boldsymbol{0}$ has unique solution of $\hat{\boldsymbol{p}}=\boldsymbol{\alpha}$, since the coefficient matrix in Equation 30 has full rank. In other words, $\|\boldsymbol{\pi}\left(\hat{\boldsymbol{p}}\right)\|>0$ for all $\hat{\boldsymbol{p}}\neq\boldsymbol{\alpha}$.

Imagine we start from $\hat{\boldsymbol{p}}^*$ and traverse a path $P$ in the hyperplane. Suppose that the path always follows the direction of $\boldsymbol{\pi}\left(\hat{\boldsymbol{p}}\right)$ for all $\hat{\boldsymbol{p}}$ on the $P$. If the total length of $P$ is less than $\|\hat{\boldsymbol{p}}^*-\boldsymbol{\alpha}\|$, then regardless of what path $P$ looks like, it always holds that $\|\boldsymbol{\pi}\left(\hat{\boldsymbol{p}}\right)\|>0$ for all $\hat{\boldsymbol{p}}$ on $P$. Define $d_{\min}:=\min_{\hat{\boldsymbol{p}}\text{ on path }P}\|\boldsymbol{\pi}\left(\hat{\boldsymbol{p}}\right)\|>0$.

Now consider the path integral along $P$. The start point of $P$ is $\hat{\boldsymbol{p}}^*$. Denote $P$'s end point at $\hat{\boldsymbol{p}}'$

$$\int_P\boldsymbol{\pi}\left(\hat{\boldsymbol{p}}\right)\cdot d\hat{\boldsymbol{p}}\stackrel{(a)}{=}\int_P\nabla\hat{L}_{\boldsymbol{\alpha}}\left(\hat{\boldsymbol{p}}\right)\cdot d\hat{\boldsymbol{p}}\stackrel{(b)}{=}\hat{L}_{\boldsymbol{\alpha}}\left(\hat{\boldsymbol{p}}'\right)-\hat{L}_{\boldsymbol{\alpha}}\left(\hat{\boldsymbol{p}}^*\right) \tag{31}$$

where "$\stackrel{(a)}{=}$" is due to that $\nabla\hat{L}_{\boldsymbol{\alpha}}\left(\hat{\boldsymbol{p}}\right)-\boldsymbol{\pi}\left(\hat{\boldsymbol{p}}\right)$ is a vector perpendicular to the hyperplane (and thus to $d\hat{\boldsymbol{p}}$). "$\stackrel{(b)}{=}$" is due to that $\nabla\hat{L}_{\boldsymbol{\alpha}}\left(\hat{\boldsymbol{p}}\right)$ defines a gradient field, and the path integral can be computed by the end points and is path independent (gradient theorem).

In addition, note that

$$\int_P\boldsymbol{\pi}\left(\hat{\boldsymbol{p}}\right)\cdot d\hat{\boldsymbol{p}}\stackrel{(c)}{=}\int_P\|\boldsymbol{\pi}\left(\hat{\boldsymbol{p}}\right)\|\cdot\|d\hat{\boldsymbol{p}}\|\geq\int_P d_{\min}\cdot\|d\hat{\boldsymbol{p}}\|=d_{\min}\int_P\|d\hat{\boldsymbol{p}}\|=d_{\min}\cdot\texttt{len}\left(P\right) \tag{32}$$

where "$\stackrel{(c)}{=}$" is due to that we always traverse the path along the direction of $\boldsymbol{\pi}\left(\hat{\boldsymbol{p}}\right)$; $\texttt{len}\left(P\right)$ denotes the total length of the path $P$.

Combining 31 and 32, we have $\hat{L}_{\boldsymbol{\alpha}}\left(\hat{\boldsymbol{p}}'\right)\geq d_{\min}\cdot\texttt{len}\left(P\right)+\hat{L}_{\boldsymbol{\alpha}}\left(\hat{\boldsymbol{p}}^*\right)$. This means, if we start from a point $\hat{\boldsymbol{p}}^*$ at level set $\mathcal{L}_{\boldsymbol{\alpha}}^{\mu-\eta}$ and traverse a path with length at most $\frac{\eta}{d_{\min}}$, we will arrive at a point $\hat{\boldsymbol{p}}'$ at the level set $\mathcal{L}_{\boldsymbol{\alpha}}^\mu$. We can further derive the following bound:

$$\texttt{dist}\left(\hat{\boldsymbol{p}}^*,\mathcal{L}_{\boldsymbol{\alpha}}^\mu\right)\leq\|\hat{\boldsymbol{p}}^*-\hat{\boldsymbol{p}}'\|\leq\texttt{len}\left(P\right)\leq\frac{\eta}{d_{\min}} \tag{33}$$

In summary, for any budget $\epsilon>0$, we can construct a $C$ according to Equation 28 by choosing parameters such that $0<\eta\leq d_{\min}\cdot\epsilon$ and $0<\beta<\frac{\eta}{\mu-\Delta(\boldsymbol{\alpha})}$. $\qquad\square$

We follow the notations from §2.3 and apply Theorem 2.4 to the binary classification task, resulting in the following corollary:

**Corollary D.13.1.** *For binary classification ($n=2$), w.l.o.g, assume $\alpha_1\in[0.5,1)$. Define $\hat{L}_+(p)=\hat{L}_{\boldsymbol{\alpha}}\left([p,1-p]^\mathsf{T}\right)$ for $p\in[\alpha_1,1)$. For strictly quasiconvex $D$ and monotonically increasing $G$, $\hat{p}_1^*\in\left[\alpha_1,\hat{L}_+^{-1}(\mu)\right)$ and $C(\boldsymbol{\alpha})\leq C(\hat{\boldsymbol{p}}^*)<C\left(\left[\hat{L}_+^{-1}(\mu),1-\hat{L}_+^{-1}(\mu)\right]^\mathsf{T}\right)$ when $\Delta(\boldsymbol{\alpha})<\mu$.*

*Proof.* For binary classification, we have $\hat{\boldsymbol{p}} = [\hat{p}_1, \hat{p}_2]^\mathsf{T} = [\hat{p}_1, 1 - \hat{p}_1]^\mathsf{T}$. Therefore,

$$\hat{L}_{\boldsymbol{\alpha}}\left(\hat{\boldsymbol{p}}\right) = \hat{L}_{\boldsymbol{\alpha}}\left([\hat{p}_1, 1 - \hat{p}_1]^\mathsf{T}\right) = -\alpha_1 \cdot \log \hat{p}_1 - (1 - \alpha_1) \cdot \log\left(1 - \hat{p}_1\right) \tag{34}$$

Let $\hat{L}_\pm\left(\hat{p}\right) = -\alpha_1 \cdot \log \hat{p} - (1 - \alpha) \cdot \log\left(1 - \hat{p}\right)$. Define $\hat{L}_+\left(\hat{p}\right) = \hat{L}_\pm\left(\hat{p}\right)$ where $\hat{p} \in [\alpha_1, 1)$ and $\hat{L}_-\left(\hat{p}\right) = \hat{L}_\pm\left(\hat{p}\right)$ where $\hat{p} \in (0, \alpha_1]$. It is easy to verify that $\hat{L}_+$ monotonically increases and $\hat{L}_-$ monotonically decreases. In addition, $\hat{L}_+\left(\alpha_1\right) = \hat{L}_-\left(\alpha_1\right) = \Delta\left([\alpha_1, 1 - \alpha_1]^\mathsf{T}\right) = \Delta\left(\boldsymbol{\alpha}\right)$.

When $\Delta\left(\boldsymbol{\alpha}\right) < \mu$, then $\hat{L}_+\left(\hat{p}\right) < \mu$ if and only if $\hat{p} \in \left[\alpha_1, \hat{L}_+^{-1}\left(\mu\right)\right)$, and $\hat{L}_-\left(\hat{p}\right) < \mu$ if and only if $\hat{p} \in \left(\hat{L}_-^{-1}\left(\mu\right), \alpha_1\right]$. Thus, let $\hat{\boldsymbol{p}}_- = \left[\hat{L}_-^{-1}\left(\mu\right), 1 - \hat{L}_-^{-1}\left(\mu\right)\right]^\mathsf{T}$ and $\hat{\boldsymbol{p}}_+ = \left[\hat{L}_+^{-1}\left(\mu\right), 1 - \hat{L}_+^{-1}\left(\mu\right)\right]^\mathsf{T}$. Then $\mathcal{L}_{\boldsymbol{\alpha}}^\mu = \{\hat{\boldsymbol{p}}_-, \hat{\boldsymbol{p}}_+\}$ and $\mathcal{L}_{\boldsymbol{\alpha}}^{<\mu}$ consists of the open line segment connecting $\hat{\boldsymbol{p}}_-$ and $\hat{\boldsymbol{p}}_+$ (not including the two end points $\hat{\boldsymbol{p}}_-$ and $\hat{\boldsymbol{p}}_+$).

For $C$, since $D$ is strictly quasiconvex, then let $\boldsymbol{\alpha} = [\alpha_1, 1 - \alpha_1]^\mathsf{T}$ and $\boldsymbol{\alpha}' = [1 - \alpha_1, \alpha_1]^\mathsf{T}$, we have $D\left(\lambda \cdot \boldsymbol{\alpha} + (1 - \lambda) \cdot \boldsymbol{\alpha}'\right) < \max\{D\left(\boldsymbol{\alpha}\right), D\left(\boldsymbol{\alpha}'\right)\} = D\left(\boldsymbol{\alpha}\right) = D\left(\boldsymbol{\alpha}'\right)$ for $\lambda \in (0, 1)$ (see Definition D.3; also note that $[p, 1 - p]^\mathsf{T}$ and $[1 - p, p]^\mathsf{T}$ should always have the same dispersion). Therefore, for all $\hat{p} \in (1 - \alpha_1, \alpha_1)$, we have $D\left([\hat{p}, 1 - \hat{p}]^\mathsf{T}\right) < D\left(\boldsymbol{\alpha}\right)$. We can similarly show that for all $\hat{p} \in (0, 1 - \alpha_1) \cup (\alpha_1, 1)$, we have $D\left([\hat{p}, 1 - \hat{p}]^\mathsf{T}\right) > D\left(\boldsymbol{\alpha}\right)$. Now since $G$ is monotonically increasing, then for $\hat{p} \in (1 - \alpha_1, \alpha_1)$, we have $C\left([\hat{p}, 1 - \hat{p}]^\mathsf{T}\right) < C\left(\boldsymbol{\alpha}\right) = \mu'$. For $\hat{p} \in (0, 1 - \alpha_1) \cup (\alpha_1, 1)$, we have $C\left([\hat{p}, 1 - \hat{p}]^\mathsf{T}\right) < \mu'$. Therefore, $C^{<\mu'}$ consists of the open line segment connecting $\boldsymbol{\alpha}$ and $\boldsymbol{\alpha}'$ (not including the two end points of $\boldsymbol{\alpha}$ and $\boldsymbol{\alpha}'$).

If $1 - \alpha_1 \le \hat{L}_-^{-1}\left(\mu\right) < \alpha_1$, then the line segment connecting $\hat{\boldsymbol{p}}_-$ and $\boldsymbol{\alpha}$ overlap with $C^{<\mu'}$. Therefore, $\mathcal{L}_{\boldsymbol{\alpha}}^{<\mu} - C^{<\mu}$ equals the line segment between $\boldsymbol{\alpha}$ and $\hat{\boldsymbol{p}}_+$. *i.e.,* $\mathcal{L}_{\boldsymbol{\alpha}}^{<\mu} - C^{<\mu} = \{[\hat{p}, 1 - \hat{p}]^\mathsf{T} \mid \hat{p} \in \left[\alpha_1, \hat{L}_+^{-1}\left(\mu\right)\right)\}$.

If $0.5 > 1 - \alpha_1 > \hat{L}_-^{-1}\left(\mu\right)$, then the line segment between $\boldsymbol{\alpha}$ and $\boldsymbol{\alpha}'$ fully overlap with the line segment between $\hat{\boldsymbol{p}}_-$ and $\hat{\boldsymbol{p}}_+$. Therefore, $\mathcal{L}_{\boldsymbol{\alpha}}^{<\mu} - C^{<\mu}$ consisting of (1) a segment between $\hat{\boldsymbol{p}}_-$ and $\boldsymbol{\alpha}'$, defined by $\mathcal{S}_1 = \{[\hat{p}, 1 - \hat{p}]^\mathsf{T} \mid \hat{p} \in \left(\hat{L}_-^{-1}\left(\mu\right), 1 - \alpha_1\right]\}$, and (2) a segment between $\boldsymbol{\alpha}$ and $\hat{\boldsymbol{p}}_+$, defined by $\mathcal{S}_2 = \{[\hat{p}, 1 - \hat{p}]^\mathsf{T} \mid \hat{p} \in \left[\alpha_1, \hat{L}_+^{-1}\left(\mu\right)\right)\}$. We can further rule out segment (1) by the following analysis. For all $\hat{p} \in (0, 0.5)$, we have $\hat{L}_\pm\left(\hat{p}\right) - \hat{L}_\pm\left(1 - \hat{p}\right) = (1 - 2\alpha_1)\left(\log\left(1 - \hat{p}\right) - \log \hat{p}\right) > 0$. This implies that $\hat{L}_+^{-1}\left(\mu\right) > 1 - \hat{L}_-^{-1}\left(\mu\right) > 0.5$. As a result, for any $\hat{\boldsymbol{p}} = [\hat{p}, 1 - \hat{p}]^\mathsf{T} \in \mathcal{S}_1$, we can find a corresponding $\hat{\boldsymbol{p}}' = [1 - \hat{p}, \hat{p}]^\mathsf{T} \in \mathcal{S}_2$. Since $\hat{\boldsymbol{p}}$ and $\hat{\boldsymbol{p}}'$ have the same dispersion, then $C\left(\hat{\boldsymbol{p}}\right) = C\left(\hat{\boldsymbol{p}}'\right)$. However, $\hat{L}_{\boldsymbol{\alpha}}\left(\hat{\boldsymbol{p}}\right) = \hat{L}_\pm\left(\hat{p}\right) > \hat{L}_\pm\left(1 - \hat{p}\right) = \hat{L}_{\boldsymbol{\alpha}}\left(\hat{\boldsymbol{p}}'\right)$. Consequently, $C\left(\hat{\boldsymbol{p}}\right) \cdot \left(\hat{L}_{\boldsymbol{\alpha}}\left(\hat{\boldsymbol{p}}\right) - \mu\right) > C\left(\hat{\boldsymbol{p}}'\right) \cdot \left(\hat{L}_{\boldsymbol{\alpha}}\left(\hat{\boldsymbol{p}}'\right) - \mu\right)$. This implies that $\hat{\boldsymbol{p}}$ cannot be a minimizer. In summary, the minimizer can only fall in $\mathcal{S}_2$.

Considering the two cases, we have $\hat{p}_1^* \in \left[\alpha_1, \hat{L}_+^{-1}\left(\mu\right)\right)$ for the minimizer $\hat{\boldsymbol{p}}^*$.

Finally, let us consider the range of $C\left(\hat{\boldsymbol{p}}^*\right)$. According to Theorem 2.4, we only need to consider the upper bound $G\left(\max_{\hat{\boldsymbol{p}} \in \mathcal{L}_{\boldsymbol{\alpha}}^\mu} D\left(\hat{\boldsymbol{p}}\right)\right)$. We have shown that $\mathcal{L}_{\boldsymbol{\alpha}}^\mu = \{\hat{\boldsymbol{p}}_-, \hat{\boldsymbol{p}}_+\}$, as well as $\hat{L}_+^{-1}\left(\mu\right) > 1 - \hat{L}_-^{-1}\left(\mu\right) > 0.5$. Thus, the dispersion of $\hat{\boldsymbol{p}}_+$ is no less than that of $\hat{\boldsymbol{p}}_-$, meaning that

$$G\left(\max_{\hat{\boldsymbol{p}} \in \mathcal{L}_{\boldsymbol{\alpha}}^\mu} D\left(\hat{\boldsymbol{p}}\right)\right) = G\left(D\left(\hat{\boldsymbol{p}}_+\right)\right) = C\left(\hat{\boldsymbol{p}}_+\right) = C\left(\left[\hat{L}_+^{-1}\left(\mu\right), 1 - \hat{L}_+^{-1}\left(\mu\right)\right]^\mathsf{T}\right). \qquad \square$$

### D.1.4 PROOF OF PROPOSITION 2.5

**Proposition D.14.** *(Originally Proposition 2.5) For any function $C$ with range in $[0, 1]$, $L_{Mowst}$ upper-bounds $L_{Mowst}^\star$.*

*Proof.* We compare Equations 1 and 3. Note that the loss $L$ is a convex function. In addition, $C\left(\boldsymbol{p}_v\right) \in [0, 1]$. Thus, $C\left(\boldsymbol{p}_v\right) \cdot \boldsymbol{p}_v + (1 - C\left(\boldsymbol{p}_v\right)) \cdot \boldsymbol{p}'_v$ is a convex combination of $\boldsymbol{p}_v$ and $\boldsymbol{p}'_v$ in $L$'s domain.

Thus, by Definition D.1,

$$C\left(\boldsymbol{p}_v\right) \cdot L\left(\boldsymbol{p}_v\right) + (1 - C\left(\boldsymbol{p}_v\right)) \cdot L\left(\boldsymbol{p}'_v\right) \geq L\left(C\left(\boldsymbol{p}_v\right) \cdot \boldsymbol{p}_v + (1 - C\left(\boldsymbol{p}_v\right)) \cdot \boldsymbol{p}'_v\right) \quad (35)$$

Summing the left-hand side and right-hand side of the above inequality over all nodes $v \in \mathcal{V}$, we derive the conclusion that $L_{\texttt{Mowst}} \geq L^\star_{\texttt{Mowst}}$. $\qquad\square$

## D.2 Proofs related to expressive power

When considering a graph as the model input, we define the following:

(1) Model A is *as expressive as* model B if, for any application of model B on any graph, there exists a corresponding model A that produces identical predictions.

(2) Model A is *more expressive than* model B if (a) model A is at least as expressive as model B, and (b) there exists a graph for which a model A can be found that yields different predictions from any model B.

In the following proof, to demonstrate that one model is "as expressive as" another, we construct confidence functions such that Mowst generates exactly the same outputs as any of its experts. To show that one model is "more expressive than" another, we construct a specific graph and a corresponding Mowst-GCN for which the Mowst-GCN can correctly classify more nodes than any standalone GCN.

### D.2.1 Proof of Proposition 2.6

**Proposition D.15.** *(Originally Proposition 2.6)* Mowst *and* Mowst$^\star$ *are at least as expressive as the MLP or GNN expert alone.*

*Proof.* We complete the proof by showing a particular confidence function $C = G \circ D$ and expert configuration, that reduce Mowst to a single expert.

Define $\boldsymbol{q}_v = C\left(\boldsymbol{p}_v\right) \cdot \boldsymbol{p}_v + (1 - C\left(\boldsymbol{p}_v\right)) \cdot \boldsymbol{p}'_v$.

Further, define the following $G$:

$$G(x) = \begin{cases} 0, & \text{when } x \leq 0 \\ 1, & \text{when } x > 0 \end{cases} \quad (36)$$

**Mowst$^\star$ $\Rightarrow$ MLP expert.** Let the GNN expert in our Mowst system always generate random prediction of $\boldsymbol{p}'_v = \frac{1}{n}\mathbf{1}$, regardless of $v$. If the MLP expert does not generate a purely random guess (*i.e.,* $\boldsymbol{p}_v \neq \frac{1}{n}\mathbf{1}$), then $D\left(\boldsymbol{p}_v\right) > 0$ according to Definition 2.1. Therefore, $C\left(\boldsymbol{p}_v\right) = G\left(D\left(\boldsymbol{p}_v\right)\right) = 1$, and thus $\boldsymbol{q}_v = \boldsymbol{p}_v$. In the extreme case, where $\boldsymbol{p}_v = \frac{1}{n}\mathbf{1}$, we have $\boldsymbol{q}_v = \boldsymbol{p}'_v$. Yet, note that we configure our GNN expert to always output $\frac{1}{n}\mathbf{1}$. As a result, we still have $\boldsymbol{q}_v = \boldsymbol{p}_v = \frac{1}{n}\mathbf{1}$. This shows that Mowst$^\star$ can always produce an identical output to that of an MLP expert alone.

**Mowst$^\star$ $\Rightarrow$ GNN expert.** We configure the MLP expert such that it always generates a trivial prediction of $\boldsymbol{p}_v = \frac{1}{n}\mathbf{1}$. Consequently, $D\left(\boldsymbol{p}_v\right) = 0$ for all $v$, and $C\left(\boldsymbol{p}_v\right) = G\left(D\left(\boldsymbol{p}_v\right)\right) = 0$ for all $v$ according to Equation 36. Thus, $\boldsymbol{q}_v = \boldsymbol{p}'_v$, implying that Mowst$^\star$ always produces an identical output to that of a GNN expert alone.

**Cases for Mowst.** Similar reasoning can be applied to the Mowst design, where we can use the $G$ function defined in Equation 36 to let the Mowst system always predict based solely on the MLP expert or the GNN expert (see Algorithm 1 for the inference operation). $\qquad\square$

**Theorem D.16.** *(Originally Theorem 2.7)* `Mowst-GCN` *and* `Mowst*-GCN` *are more expressive than the GCN expert alone.*

*Proof.* We consider unweighted and undirected graphs for this theorem. According to Proposition 2.6, we know that `Mowst-GCN` and `Mowst*-GCN` are at least as expressive as a standalone GCN. To show that `Mowst-GCN` and `Mowst*-GCN` are more expressive than GCN, we compare the function class consisting of all possible $K$-layer GCN models and the function class consisting of all possible $K$-layer `Mowst*-GCN` models. Then we will construct an example graph $\mathcal{G}$ with a pair of nodes $u$ and $v$. On this $\mathcal{G}$, for *any* GCN model $\mathcal{M}$, we can find a corresponding `Mowst*-GCN` model $\mathcal{M}'$, such that

(1) $\mathcal{M}$ cannot distinguish $u$ from $v$,

(2) $\mathcal{M}'$ can distinguish $u$ from $v$ and,

(3) if $\mathcal{M}$ can distinguish some nodes, then $\mathcal{M}'$ can also distinguish them.

Denote $\mathcal{N}_v^k$ as the set of neighbor nodes that are $k$ hops away from $v$ (specifically, $\mathcal{N}_v^0 = \{v\}$). We further enforce the following constraints on the *structure* of $\mathcal{G}$:

(1) The $K$-hop neighborhoods of $u$ and $v$ do not overlap, *i.e.,* $\left(\bigcup_{0 \leq k \leq K} \mathcal{N}_u^k\right) \cap \left(\bigcup_{0 \leq k \leq K} \mathcal{N}_v^k\right) = \emptyset$;

(2) For any node $w \in \mathcal{N}_v^k$ (where $0 \leq k \leq K-1$), there exists an edge $(w, w^*) \in \mathcal{E}$ where $w^* \in \mathcal{N}_v^{k+1}$ and $w^*$ does not connect any node in $\mathcal{N}_v^k$ other than $w$; Similar constraint applies to node $u$'s neighborhood $\mathcal{N}_u^k$;

(3) The structures (i.e., not considering node features) of the $K$-hop neighborhood of $u$ and $v$ are isomorphic. i.e., For the two subgraphs induced by $\bigcup_{0 \leq k \leq K} \mathcal{N}_v^k$ and $\bigcup_{0 \leq k \leq K} \mathcal{N}_u^k$, we can find an isomorphic node mapping $F$ where $F(u) = v$, and a node $v' \in \mathcal{N}_v^k$ is mapped by $F$ to a node $u' \in \mathcal{N}_u^k$.

Next, we discuss how to set the node *features* for $\mathcal{G}$. First, we let $u$ and $v$ have different node features, and thus $u$ and $v$ should be distinguished if the model is powerful enough. Then, we consider the features of the neighbors of $u$ and $v$. Recall the operation of a GCN layer. For a node $w$ in layer $k$, the GCN performs weighted sum of the embedding vectors of $w$'s direct neighbors in GCN's layer $(k-1)$. Denote $\boldsymbol{h}_w^{(k)}$ as the output embedding vector of $w$ in layer $k$. Then,

$$\boldsymbol{h}_w^{(k)} = \sigma\left(\left(\boldsymbol{W}^{(k)}\right)^{\mathsf{T}} \cdot \sum_{w' \in \mathcal{N}_w^1 \cup \{w\}} \frac{1}{\sqrt{(\deg(w)+1) \cdot (\deg(w')+1)}} \boldsymbol{h}_{w'}^{(k-1)} + \boldsymbol{b}^{(k)}\right) \qquad (37)$$

where $\sigma$ is the activation function, $\boldsymbol{W}^{(k)}$ and $\boldsymbol{b}^{(k)}$ are the learnable weight matrix and bias vector of layer $k$, and $\deg(\cdot)$ denotes the degree of the node.

Denote $\boldsymbol{x}_* = \boldsymbol{h}_*^{(0)}$ as the raw node feature. We can construct the graph features such that $\sum_{w' \in \mathcal{N}_w^1 \cup \{w\}} \frac{1}{\sqrt{(\deg(w)+1) \cdot (\deg(w')+1)}} \boldsymbol{h}_{w'}^{(0)} = \boldsymbol{0}$ for all $w \in \bigcup_{0 \leq k \leq K-1} \mathcal{N}_v^k$. This can always be achieved due to the constraints (1) and (2) above: for each $w \in \mathcal{N}_v^k$, we can find a $w^*$ (as described in constraint (2)) and set $\boldsymbol{h}_{w^*}^{(0)}$ s.t. it "counter-acts" the aggregated features of all $w$'s other neighbors.

In this case, by Equation 37, $\boldsymbol{h}_w^{(1)} = \sigma\left(\left(\boldsymbol{W}^{(0)}\right)^{\mathsf{T}} \cdot \boldsymbol{0} + \boldsymbol{b}^{(0)}\right) = \sigma\left(\boldsymbol{b}^{(0)}\right)$ for all $w \in \bigcup_{0 \leq k \leq K-1} \mathcal{N}_v^k$. Thus, the layer 1 output features are *identical* for all $(K-1)$-hop neighbors of both $u$ and $v$. i.e., after GCN's layer 1 aggregation, we *lose all information* of the node features in the neighborhoods of $u$ and $v$. For layer 2 and onwards, the GCN sees two completely isomorphic (including both structure

and feature) neighborhood subgraphs of $u$ and $v$ (due to constraint (3) above). And thus, regardless of the GCN's model parameters, the GCN will always output identical embedding vectors for $u$ and $v$. And no GCN can distinguish $u$ from $v$.

Now consider `Mowst-GCN` and `Mowst*-GCN`. Recall that $u$ and $v$ have different self-features, $\boldsymbol{x}_u \neq \boldsymbol{x}_v$. In addition, for all other nodes in $\mathcal{V} - \{u\} - \{v\}$ that we want to predict, they have self-features different from both $\boldsymbol{x}_u$ and $\boldsymbol{x}_v$. Consider the following confidence function:

$$G(x) = \begin{cases} 0, & \text{when } x \leq 0 \\ 1, & \text{when } x > 0 \end{cases} \tag{38}$$

Due to universal approximation theory (Hornik et al., 1989), we can have an MLP expert that differentiates nodes $u$ and $v$ based on their input features $\boldsymbol{x}_u$ and $\boldsymbol{x}_v$ (and produces meaningful predictions not equal to $\frac{1}{n}\mathbf{1}$), while always producing a $\frac{1}{n}\mathbf{1}$ prediction for all other nodes. In this scenario, for $u$ and $v$, the MLP's prediction exhibits positive dispersion, leading to $G = 1$. Consequently, the confidence function acts as a binary gate, completely disabling the GCN expert on $u$ and $v$. Otherwise, the MLP's prediction has zero dispersion, resulting in $G = 0$. In this case, the confidence function entirely disables the MLP expert and relies solely on the GCN expert. With this configuration, both `Mowst-GCN` and `Mowst*-GCN` can differentiate all nodes that a standalone GCN model can, and they can also distinguish nodes that any GCN model cannot ($u$ *vs.* $v$). This demonstrates that `Mowst-GCN` and `Mowst*-GCN` are strictly more expressive than a GCN alone. $\quad\square$

### D.2.3 PROOF OF PROPOSITION 2.8

**Proposition D.17.** *(Originally Proposition 2.8)* `Mowst` *and* `Mowst*` *are at least as expressive as any expert alone.*

*Proof.* The proof follows the idea of proving Proposition 2.6.

The logic to prove the case of `Mowst` is identical to that of `Mowst*`, and thus we only discuss the `Mowst` case in detail. Consider a confidence function $C = G \circ D$ defined as follows:

$$G(x) = \begin{cases} 0, & \text{when } x \leq 0 \\ 1, & \text{when } x > 0 \end{cases} \tag{39}$$

Equation 44 defines the general form for `Mowst` consisting of $M$ experts. The coefficient $\tau_m^v := \left( \prod_{1 \leq i < m} (1 - C_i^v) \right) \cdot C_m^v$ in front of the loss $L_m^v$ represents the probability of activating expert $m$ during inference. This setup allows us to easily generalize Algorithm 1 based on Equation 44. To ensure that the entire `Mowst` system produces identical results as the $m$-th expert, we need to approximately show that $\tau_m^v = 1$ for all $v$, and $\tau_{m'}^v = 0$ for $m' \neq m$ (with some exceptions to be discussed separately).

For $m' \neq m$, we always let the expert $m'$ to generate random guess of $\frac{1}{n}\mathbf{1}$ for all $v$. Note that a $\frac{1}{n}\mathbf{1}$ prediction corresponds to a 0 dispersion, and thus the corresponding $G$ and $C$ are also 0. A prediction not equal to $\frac{1}{n}\mathbf{1}$ leads to positive dispersion, and thus a confidence of 1. Given the binary nature of our confidence values, to achieve $\tau_{m'}^v = 0$ for some $1 \leq m' \leq M$, we need $C_{m'}^v = 0$ or at least one preceding expert to have $C_{m''}^v = 1$ for some $m'' < m'$. To obtain $\tau_{m'}^v = 1$ for some $1 \leq m' \leq M$, all preceding experts must have $C_{m''}^v = 0$ for all $m'' < m'$, and $C_{m'}^v = 1$.

According to the above categorization, for expert $m'$ where $m' < m$, we always have their $\tau_{m'}^v = 0$ since $C_{m'}^v = 0$. If expert $m$ generates a non-$\frac{1}{n}\mathbf{1}$ prediction, then $\tau_m^v = 1$ and thus the whole `Mowst` system reduces to a single expert $m$. If expert $m$ predicts $\frac{1}{n}\mathbf{1}$, then $\tau_M^v = 1$ (since by definition, $C_M^v = 1$ for all $v$). Since the expert $M$ also predicts $\frac{1}{n}\mathbf{1}$, the system's output is also equivalent to expert $m$'s output.

Therefore, both `Mowst` and `Mowst*` can produce identical results as any individual expert. $\quad\square$

# E ADDITIONAL ALGORITHMIC DETAILS

## E.1 DERIVATION OF OPTIMIZATION PROBLEM 2

We re-write $L_{\texttt{Mowst}}$ (Equation 1) as follows:

$$
\begin{aligned}
L_{\texttt{Mowst}} =& \frac{1}{|\mathcal{V}|} \sum_{v \in \mathcal{V}} \left( C\left(\boldsymbol{p}_v\right) \cdot L\left(\boldsymbol{p}_v, \boldsymbol{y}_v\right) + \left(1 - C\left(\boldsymbol{p}_v\right)\right) \cdot L\left(\boldsymbol{p}'_v, \boldsymbol{y}_v\right) \right) \\
=& \frac{1}{|\mathcal{V}|} \sum_{v \in \mathcal{V}} \left( C\left(\boldsymbol{p}_v\right) \cdot \left( L\left(\boldsymbol{p}_v, \boldsymbol{y}_v\right) - L\left(\boldsymbol{p}'_v, \boldsymbol{y}_v\right) \right) + L\left(\boldsymbol{p}'_v, \boldsymbol{y}_v\right) \right) \\
=& \frac{1}{|\mathcal{V}|} \sum_{v \in \mathcal{V}} C\left(\boldsymbol{p}_v\right) \cdot \left( L\left(\boldsymbol{p}_v, \boldsymbol{y}_v\right) - L\left(\boldsymbol{p}'_v, \boldsymbol{y}_v\right) \right) + \frac{1}{|\mathcal{V}|} \sum_{v \in \mathcal{V}} L\left(\boldsymbol{p}'_v, \boldsymbol{y}_v\right)
\end{aligned}
\tag{40}
$$

Since we optimize the MLP parameters by fixing the GNN, $L\left(\boldsymbol{p}'_v, \boldsymbol{y}_v\right)$ remains constant throughout the process. Therefore,

$$
\arg \min_{\{\boldsymbol{p}_v\}} L_{\texttt{Mowst}} = \arg \min_{\{\boldsymbol{p}_v\}} \sum_{v \in \mathcal{V}} C\left(\boldsymbol{p}_v\right) \cdot \left( L\left(\boldsymbol{p}_v, \boldsymbol{y}_v\right) - L\left(\boldsymbol{p}'_v, \boldsymbol{y}_v\right) \right)
\tag{41}
$$

Now we simplify it as follows:

$$
\begin{aligned}
& \sum_{v \in \mathcal{V}} C\left(\boldsymbol{p}_v\right) \cdot \left( L\left(\boldsymbol{p}_v, \boldsymbol{y}_v\right) - L\left(\boldsymbol{p}'_v, \boldsymbol{y}_v\right) \right) \\
=& \sum_{1 \leq i \leq m} \sum_{v \in \mathcal{m}_i} C\left(\boldsymbol{p}_v\right) \cdot \left( L\left(\boldsymbol{p}_v, \boldsymbol{y}_v\right) - L\left(\boldsymbol{p}'_v, \boldsymbol{y}_v\right) \right) \\
=& \sum_{1 \leq i \leq m} \sum_{v \in \mathcal{m}_i} C\left(\boldsymbol{p}_v\right) \cdot L\left(\boldsymbol{p}_v, \boldsymbol{y}_v\right) - \sum_{1 \leq i \leq m} \sum_{v \in \mathcal{m}_i} C\left(\boldsymbol{p}_v\right) \cdot L\left(\boldsymbol{p}'_v, \boldsymbol{y}_v\right) \\
=& \sum_{1 \leq i \leq m} C\left(\hat{\boldsymbol{p}}_i\right) \cdot \left( \sum_{v \in \mathcal{m}_i} L\left(\hat{\boldsymbol{p}}_i, \boldsymbol{y}_v\right) \right) - \sum_{1 \leq i \leq m} C\left(\hat{\boldsymbol{p}}_i\right) \cdot \left( \sum_{v \in \mathcal{m}_i} L\left(\boldsymbol{p}'_v, \boldsymbol{y}_v\right) \right) \\
=& \sum_{1 \leq i \leq m} C\left(\hat{\boldsymbol{p}}_i\right) \cdot \left( \sum_{v \in \mathcal{m}_i} L\left(\hat{\boldsymbol{p}}_i, \boldsymbol{y}_v\right) - \sum_{v \in \mathcal{m}_i} L\left(\boldsymbol{p}'_v, \boldsymbol{y}_v\right) \right) \\
=& \sum_{1 \leq i \leq m} C\left(\hat{\boldsymbol{p}}_i\right) \cdot \left( |\mathcal{m}_i| \cdot \hat{L}_{\boldsymbol{\alpha}_i}\left(\hat{\boldsymbol{p}}_i\right) - |\mathcal{m}_i| \cdot \mu_i \right) \\
=& |\mathcal{m}_i| \sum_{1 \leq i \leq m} C\left(\hat{\boldsymbol{p}}_i\right) \cdot \left( \hat{L}_{\boldsymbol{\alpha}_i}\left(\hat{\boldsymbol{p}}_i\right) - \mu_i \right)
\end{aligned}
\tag{42}
$$

where Equation 42 is (almost) exactly our objective in optimization problem 2, with the only difference being the scaling factor $|\mathcal{m}_i|$. We can use the "decomposible" argument in §D.1.3 to easily derive that this scaling factor does not affect the optimizer $\hat{\boldsymbol{p}}_i^*$.

## E.2 EXTENDING TO MORE THAN TWO EXPERTS

Following the notations in §2.7, we first have the case of 3 agents:

$$
\begin{aligned}
L_{\texttt{Mowst}} =& \frac{1}{|\mathcal{V}|} \sum_{v \in \mathcal{V}} \left( C_1^v \cdot L_1^v + \left(1 - C_1^v\right) \cdot \left( C_2^v \cdot L_2^v + \left(1 - C_2^v\right) \cdot L_3^v \right) \right) \\
=& \frac{1}{|\mathcal{V}|} \sum_{v \in \mathcal{V}} \left( C_1^v \cdot L_1^v + \left(1 - C_1^v\right) \cdot C_2^v \cdot L_2^v + \left(1 - C_1^v\right)\left(1 - C_2^v\right) \cdot L_3^v \right)
\end{aligned}
\tag{43}
$$

In general, for $M$ experts, we have the following loss:

$$L_{\texttt{Mowst}} = \frac{1}{|\mathcal{V}|} \sum_{v \in \mathcal{V}} \sum_{1 \le m \le M} \left( \prod_{1 \le i < m} (1 - C_i^v) \right) \cdot C_m^v \cdot L_m^v \qquad (44)$$

where we define $\prod_{x \le i < x} (1 - C_i^v) = 1$ for any integer $x$.

We further define the following term:

$$L_{\ge q}^v = \sum_{q \le m \le M} \left( \prod_{q \le i < m} (1 - C_i^v) \right) \cdot C_m^v \cdot L_m^v \qquad (45)$$

Therefore, $L_{\texttt{Mowst}} = \frac{1}{|\mathcal{V}|} \sum_{v \in \mathcal{V}} L_{\ge 1}^v$ and $L_{\ge q}^v = C_q^v \cdot L_q^v + \left(1 - C_q^v\right) \cdot L_{\ge q+1}^v$ defines the basic recursive formulation.

### E.3    CALCULATION OF COMPUTATION COMPLEXITY

We provide more details to support the analysis on computation cost in §2.6. We derive the specific equations for computation complexity for various model architectures, including

- Vanilla MLP
- Vanilla GNN (with GCN and GraphSAGE as examples)
- GNN with skip connections
- `Mowst` consisting of an MLP expert, a GNN expert, and an optional MLP for confidence computation

We further discuss how state-of-the-art techniques for scalable and efficient GNNs improve the computation complexity of a vanilla GNN, but still fall short in making a GNN as lightweight as an MLP. In summary, our analysis reveals the following:

- A GNN, even when scaled up, is significantly more computationally expensive than an MLP.
- A `Mowst` has a similar computational complexity to that of its corresponding GNN expert alone.
- A GNN with skip connections remains substantially more expensive than a `Mowst` with an equivalent number of model parameters.

Our analysis justifies the claim that `Mowst` **is efficient**, and our **baseline comparison criteria is fair** (§A.5).

### E.3.1    SETUP

We follow the same setup for computation complexity analysis as Zeng et al. (2021), where we analyze *the total number of arithmetic operations needed to generate prediction for one target node during inference*. Note that:

- We focus on inference since the exact equations for training are hard to derive without strong assumptions about the specific training algorithm and convergence behavior. The conclusions from our following analysis apply to the training costs as well.
- Batch processing on a group of target nodes may help reduce the computation complexity but its benefits tend to diminish on *large*, realistic graphs (Zeng et al., 2021). In addition, the benefits of batch processing strongly depend on both the graph connectivity pattern, and the neighborhood similarity among the same-batch target nodes (Fey et al., 2021). Thus, similar to Zeng et al. (2021), we only consider the case for each individual target node.

**Notations.**    Denote $\ell$ as the total number of layers, and $f$ as the hidden dimension (for simplicity, we assume that all layers have the same hidden dimension, and the raw node features are also of dimension $f$).

Denote $\mathcal{N}^v$ as the set of direct neighbors of node $v$, excluding $v$ itself (*i.e.,* a node in $\mathcal{N}^v$ is connected to $v$ via an edge). Denote $\mathcal{N}_i^v$ as the set of $v$'s neighbors within $i$ hops, *i.e.,* $\mathcal{N}_i^v$ consists of all nodes that can reach $v$ in *no more than $i$* hops. For instance, $\mathcal{N}_1^v = \mathcal{N}^v \cup \{v\}$. Denote $b_i^v = |\mathcal{N}_i^v|$ as the number of $v$'s neighbors within $i$ hops (*e.g.,* $b_0^v = 1$ and $b_1^v$ equals $v$'s degree plus 1). Denote $\boldsymbol{x}_{i-1}^v \in \mathbb{R}^{f \times 1}$ as $v$'s embedding vector input to layer $i$. Denote $\boldsymbol{W}_i \in \mathbb{R}^{f \times f}$ as the weight parameter matrix of layer $i$. Denote $\gamma$ as the computation cost measured by total number of multiplication-accumulation operations.

**Simplifications.**    In the following calculation, we omit the non-linear activation and normalization layers, as their computation cost is negligible compared with the GNN and MLP layers.

### E.3.2    COMPUTATION COST OF MLP

Each layer $i$ performs the following computation:

$$\boldsymbol{x}_i^v = \boldsymbol{W}_\ell \cdot \boldsymbol{x}_{i-1}^v \tag{46}$$

As a result, the number of multiplication-accumulation operations (corresponding to matrix multiplication) equals $f^2$. For all $\ell$ layers, the total computation cost equals

$$\gamma_{\text{MLP}} = f^2 \cdot \ell \tag{47}$$

### E.3.3    COMPUTATION COST OF GNN

The GNN performs recursive neighborhood aggregation to generate the embedding for the target node $v$. Specifically,

- The $\ell$-th (*i.e.,* last) layer aggregates information from $v$'s 1-hop neighbors $\mathcal{N}_1^v$, and outputs a single embedding $\boldsymbol{x}_\ell^v$ for $v$ itself.
- The $(\ell - 1)$-th layer aggregates information from $v$'s 2-hop neighbors $\mathcal{N}_2^v$, and outputs $b_1^v$ embeddings for each of $v$'s 1-hop neighbors $\mathcal{N}_1^v$.
- ...
- The first layer aggregates information from $v$'s $\ell$-hop neighbors $\mathcal{N}_\ell^v$, and outputs $b_{\ell-1}^v$ embeddings for each of $v$'s $(\ell - 1)$-hop neighbors $\mathcal{N}_{\ell-1}^v$.

Mathematically, we formulate the layer operation as follows:

$$\boldsymbol{x}_i^v = \text{UPDATE}\left(\boldsymbol{x}_{i-1}^v, \text{AGGREGATE}\left(\{\boldsymbol{x}_{i-1}^u | u \in \mathcal{N}^v\}\right)\right) \tag{48}$$

where the function $\text{AGGREGATE}\left(\cdot\right)$ aggregates the previous-layer embedding vectors of $v$'s neighbors, and the function $\text{UPDATE}\left(\cdot\right)$ performs transformation on $v$'s own embedding vector from the last layer, as well as the aggregated neighbor embedding. Different GNNs have difference choices of the $\text{UPDATE}\left(\cdot\right)$ and $\text{AGGREGATE}\left(\cdot\right)$ functions. In addition, to implement **skip connection**, we can let the $\text{UPDATE}\left(\cdot\right)$ function specifically operate on $\boldsymbol{x}_{i-1}^v$.

Note that for layer $i$, each of its output nodes in $\mathcal{N}_{\ell-i}^v$ will execute Equation 48. Therefore, we describe the general layer operation as follows: the $i$-th GNN layer

(1) aggregates information from nodes in $\mathcal{N}_{\ell-i+1}^v$ into $b_{\ell-i}^v$ intermediate embeddings,

(2) updates the intermediate embeddings and self-embeddings of $\mathcal{N}_{\ell-i}^v$, and

(3) outputs embeddings for nodes in $\mathcal{N}_{\ell-i}^v$.

We are now ready to study the computation cost for two specific GNN architectures, GCN (Kipf & Welling, 2016) and GraphSAGE (Hamilton et al., 2017).[5]

Both GCN and GraphSAGE implement AGGREGATE $(\cdot)$ as a weighted sum of the neighbor embeddings. Thus, layer $i$ gathers the previous-layer embeddings of $\mathcal{N}_{\ell-i+1}^v$, and reduces them into $b_{\ell-i}$ aggregated embeddings, via vector summation. Since AGGREGATE $(\cdot)$ only involves addition but no multiplication, its cost is much lower than UPDATE $(\cdot)$, and should be ignored according to our computation cost definition (recall that for computation cost, we only count one multiplication-accumulation as one unit cost, which is consistent with Zeng et al. (2021)).

After AGGREGATE $(\cdot)$, both GCN and GraphSAGE implements UPDATE $(\cdot)$ as a *linear transformation* via the layer's weight matrix. The difference is that for GCN, each layer only has a single weight matrix operating on the aggregated embedding (*i.e.,* output of AGGREGATE $(\cdot)$). For GraphSAGE, each layer $i$ has two weight matrices, operating on the aggregated embedding vector and $x_{i-1}^v$, respectively. The two embedding vectors after GraphSAGE's linear transformation are added to generate the final $x_i^v$.[6] The computation cost for layer $i$ of GCN's UPDATE $(\cdot)$ function is thus $f^2 \cdot b_{\ell-i}^v$. The computation cost for GraphSAGE is doubled as $2 \cdot f^2 \cdot b_{\ell-i}$ due to the operation from two weight matrices. For all $\ell$ layer, we omit the cost of AGGREGATE $(\cdot)$ as discussed before. Therefore, the total computation cost equals

$$\gamma_{\text{GCN}} \approx f^2 \cdot \sum_{1 \leq i \leq \ell} b_{\ell-i} \tag{49}$$

$$\gamma_{\text{GraphSAGE}} \approx 2f^2 \cdot \sum_{1 \leq i \leq \ell} b_{\ell-i} \tag{50}$$

where $b_{\ell-i}$ denotes the average number of $(\ell-i)$-hop neighbors among all target nodes.

In §2.6, we perform a further simplification of Equations 49 and 50: note that $\sum_{1 \leq i \leq \ell} b_{\ell-i} = b_{\ell-1} + \sum_{2 \leq i \leq \ell} b_{\ell-i} \geq b_{\ell-1} + \sum_{2 \leq i \leq \ell} 1 = b_{\ell-1} + \ell - 1$. Thus, the computation cost is asymptotically

$$\Omega\left(f^2 \cdot (\ell + b_{\ell-1})\right) \tag{51}$$

which is the same as the results in §2.6.

In large, realistic graphs, the number of $\ell$-hop neighbors of a target node can grow exponentially with $\ell$. For instance, in a social network, if one person has 10 friends, then he/she will have $10^2$ friends of friends, and $10^3$ friends of friends of friends. Such an exponential growth is commonly referred to as "*neighborhood explosion*" (Chen et al., 2018; Hamilton et al., 2017; Zeng et al., 2020; Fey et al., 2021) in the GNN literature. Consequently, **a realistic GNN has much higher computation cost than an MLP**: $\gamma_{\text{GNN}} \gg \gamma_{\text{MLP}}$ since $b_{\ell-1} \gg \ell$ even for $\ell$ as small as 2 or 3.

### E.3.4 COMPUTATION COST OF GNN WITH SKIP CONNECTION *vs.* GNN WITH MLP EXPERT

First, let us derive the computation cost of `Mowst` with an MLP expert and a GNN expert. For a target node, the MLP expert only operate on the node itself, resulting in a cost of $f^2 \cdot \ell$ as per Equation 46. If there is a learnable confidence function implemented by another MLP, its cost is also $f^2 \cdot \ell$ assuming this MLP has the same architecture as the MLP expert (as set up in §3). The cost of the GNN expert is given by Equations 49, 50 and 51. Therefore, the computation cost of `Mowst` can be estimated as:

---

[5]Complexity of other architectures can be derived similarly. We omit the details to avoid redundancy.

[6]The original GraphSAGE (Hamilton et al., 2017) paper has proposed variants of AGGREGATE $(\cdot)$ and UPDATE $(\cdot)$. The version described here is the most common one (see for instance, the default implementation from PyTorch Geometric `https://pytorch-geometric.readthedocs.io/en/latest/generated/torch_geometric.nn.conv.SAGEConv.html`), and also used in our experiments.

$$\gamma_{\texttt{Mowst-GCN}} \approx 2f^2 \cdot \ell + f^2 \cdot \sum_{1 \le i \le \ell} b_{\ell-i} \approx \gamma_{\text{GCN}} \tag{52}$$

$$\gamma_{\texttt{Mowst-SAGE}} \approx 2f^2 \cdot \ell + 2f^2 \cdot \sum_{1 \le i \le \ell} b_{\ell-i} \approx \gamma_{\text{GraphSAGE}} \tag{53}$$

$$\tag{54}$$

where the second "$\approx$" is again due to $b_{\ell-1} \gg \ell$.

Now let us consider a single GNN model **with a skip connection** in each layer. As shown by the description regarding Equation 48, a skip connection can be implemented via a specific $\texttt{UPDATE}(\cdot)$ function. If the skip connection implements a linear transform on the self-features generated by the previous layer (*i.e.*, $\boldsymbol{x}_{i-1}^v$ of Equation 48), then GraphSAGE discussed in §E.3.3 already implements the skip connection.

Note that adding a skip connection will increase the cost of $\texttt{UPDATE}(\cdot)$ due to additional linear transformation. The more expensive $\texttt{UPDATE}(\cdot)$ will be applied on all the $(\ell-1)$-hop neighbors of the target node $v$. Thus, **adding a skip connection significantly increases the computation cost of a GNN**. Specifically, note from Equations 49 and 50 that GraphSAGE is twice as expensive as GCN. The "2" factor is exactly due to the skip connection. Similarly, if we modify the GCN architecture by adding a skip connection in each GCN layer, we will have $\gamma_{\text{GCN-skip}} = 2\gamma_{\text{GCN}}$.

**Remark.** From the architecture perspective, adding a skip connection in each GNN layer can be seen as *breaking down the MLP expert of Mowst and fusing each MLP layer with each GNN layer*. However, from the computation cost perspective, **a $\texttt{Mowst}$ model with MLP + GNN has much lower cost than GNN + skip connection**. The fundamental reason behind the gap in computation cost is that **the skip connection increases the cost on all neighbors, while the MLP expert only introduces overhead on the target node itself.**

### E.3.5 Scalable GNN techniques

Note that many techniques have been proposed to improve the scalability of GNN, including neighbor sampling (Hamilton et al., 2017; Ying et al., 2018), subgraph sampling (Zeng et al., 2020; Gasteiger et al., 2022) and historical embedding reuse (Fey et al., 2021; Shi et al., 2023). Scalable GNN techniques may reduce the GNN computation cost derived in §E.3.3, but **a scalable GNN will still be much more expensive than an MLP**. We give a brief summary of the reasons:

- Even with aggressive sampling, the neighborhood size will still be much larger than 1.
- Many sampling techniques (Hamilton et al., 2017; Shi et al., 2023; Ying et al., 2018; Fey et al., 2021; Zeng et al., 2020) aim to *approximate* the aggregation on the *full* neighborhood. Thus, sampling trade offs accuracy for efficiency. In addition, many sampling algorithms impose strong assumptions on the neighbor aggregation function.
- Many scalable GNN techniques apply only to the training phase (Zeng et al., 2020; Shi et al., 2023; Fey et al., 2021) and, as such, do not address the computation challenges during inference.
- GPUs are inefficient in processing graph data due to the sparsity and irregularity of edge connections. As a result, GPU utilization is significantly lower when running a GNN (whether scalable or not) compared to running an MLP.

## F  A Study on "Failure Cases"

We present an analysis on how $\texttt{Mowst}$ can handle two typical "failure cases" by adjusting the predictions of the MLP expert and the shape of the confidence function during training.

### F.1  What are the two typical "failure cases"?

Our goal here is **not** to exhaustively address all possible corner cases. Instead, we focus on **typical** failure cases that may seem controversial to aid our understanding the interactions between experts.

Considering the overall training objective in Equation 1, the intuitive "**success cases**" should be:

- Low MLP loss with high confidence: **confident & correct MLP predictions**
- High MLP loss with low confidence: **unconfident & incorrect MLP predictions**

Therefore, the typical failure cases are essentially the opposites of these success cases: (1) Confident & incorrect MLP predictions, and (2) Unconfident & correct MLP predictions.

Given that we are considering two experts, we can refine the **failure cases** by taking into account the relative strengths of the experts:

- **Case 1**: Confident & incorrect MLP predictions + correct GNN predictions;
- **Case 2**: Unconfident & correct MLP predictions + incorrect GNN predictions.

*Side note*: Nodes associated with Case 1 should have different self-features compared to those associated with Case 2, otherwise their confidence levels would be identical.

Since the balance between the experts is ultimately determined by the loss, we quantify each term in Equation 1:

- **Case 1**: High $C$; high $L_{\mathrm{MLP}}$; low $(1 - C)$; low $L_{\mathrm{GNN}}$.
- **Case 2**: Low $C$; low $L_{\mathrm{MLP}}$; high $(1 - C)$; high $L_{\mathrm{GNN}}$.

where $L_{\mathrm{MLP}}$ and $L_{\mathrm{GNN}}$ denote the MLP loss and GNN loss terms in Equation 1, respectively.

### F.2  HOW DOES Mowst ADDRESS THE "FAILURE CASES"?

**Observation**: The commonality between the two failure cases is that for one expert, both its loss and the weight coefficient in front of the loss are high. To address the failure case, the Mowst training should be able to either *reduce the loss*, OR *reduce the weight coefficient*.

In summary, Mowst will take the following steps:

(1) Update the MLP model to make the predictions for the **case 1** nodes closer to a random guess.
(2) Update the confidence function to give it an "over-confident" shape, increasing the $C$ value for the **case 2** nodes.

These steps are not independent. Step 1 is straightforward for an MLP, so it will not significantly affect predictions for case 2 nodes. In Step 2, an "over-confident" $C$ means *it is easier for the MLP to achieve high confidence*. For discussion, consider a simple function where $C(\boldsymbol{p}) = 0$ if the dispersion of $\boldsymbol{p}$ is less than $\tau$, and $C(\boldsymbol{p}) = 1$ if the dispersion is greater than $\tau$. We can make $C$ more "over-confident" with a smaller $\tau$. This update affects $C$ for both case 1 and case 2 nodes.

After executing both steps, we analyze the joint effect on case 1 and case 2 nodes:

In **step 2**, we can reduce $\tau$ until the dispersion of the MLP's case 2 predictions is higher than $\tau$ (we can always do so since MLP's case 2 predictions are correct by definition). Simultaneously, in **step 1**, we need to push the MLP's case 1 predictions towards a random guess until their dispersion is lower than $\tau$ (we can always do so since random guess has 0 dispersion).

**Net effect of reduced Mowst loss.**    Each term in the loss changes after executing steps 1 and 2. For case 1, $C$ will reduce (to 0 under our example confidence function). $L_{\mathrm{MLP}}$ will increase. The net effect is that the *overall loss is reduced*: we now have $L'_{Mowst} = 0 \cdot L'_{\mathrm{MLP}} + (1 - 0) \cdot L_{\mathrm{GNN}} = L_{\mathrm{GNN}}$ (by definition of case 1, $L_{\mathrm{GNN}}$ is low). For case 2, $C$ will increase (to 1 under our example confidence function). $L_{\mathrm{MLP}}$ will remain the same. The net effect is that the *overall loss is also reduced*: $L'_{Mowst} = 1 \cdot L_{\mathrm{MLP}} + (1 - 1) \cdot L_{\mathrm{GNN}} = L_{\mathrm{MLP}}$ where $L_{\mathrm{MLP}}$ is low by definition of case 2.

**Net effect of improved prediction behaviors.**    After jointly executing steps 1 & 2:

- For case 1, the MLP has unconfident, incorrect predictions.

- For case 2, the MLP has confident, correct predictions.

Thus, through `Mowst` training, we have **successfully converted the two typical failure cases into two success cases**.

## G    DESIGN CONSIDERATIONS

### G.1    ORDER OF EXPERTS: MLP-GNN *vs.* GNN-MLP

With one weak expert and one strong expert, two ordering possibilities for the mixture exist: MLP-GNN (our design) and GNN-MLP (alternative design).

In general, let us consider experts A & B. According to the analysis in §2.3 (especially the last paragraph), when we compute confidence based on expert A, `Mowst` will be biased towards the other expert B. In other words, on some training nodes, if expert B achieves a lower loss, then `Mowst` will likely accept predictions from B and ignore those from A. Conversely, when expert A achieves a lower loss, `Mowst` may still have some non-zero probability (controlled by the learnable $C$) to accept B's prediction. When B is the stronger expert with better generalization capability, the above bias is desirable when applying `Mowst` to the test set. Since GNN generalizes better (Yang et al., 2023), we prefer the MLP-GNN order (current design) over GNN-MLP (alternative design).

### G.2    NON-CONFIDENCE-BASED GATING

Consider a learnable confidence function $C$ implemented by an MLP. To create a non-confidence-based learnable gating module, we can modify the MLP for $C$ by replacing its dispersion-based input with the raw input features of the target node. This modified gating module would then output weights for each expert instead of the confidence $C$.

The gating modules in existing Mixture-of-Expert systems (*e.g.,* Shazeer et al. (2017); Wang et al. (2023)) resemble the above proposed gating module. Theoretically, such a gate can also simulate our confidence function – the initial layers of this alternative gating neural network can learn to precisely generate the prediction logits of the MLP expert, while the remaining layers can learn to calculate dispersion and the $G$ function. In this sense, our confidence module can be seen as a specific type of gate, which is significantly simplified based on the inductive bias of the weak-strong combination. Our confidence-based gating makes the model explainable (§2.3, §2.4), expressive and efficient (§2.6). More importantly, it enables `Mowst` to achieve significantly higher accuracy than GNNs based on traditional gating (*e.g.,* GraphMoE (Wang et al., 2023), Table 1) due to our simplified design.

## H    LIMITATIONS & BROADER IMPACT

### H.1    LIMITATIONS

Our design is fundamentally driven by the goal of improving model capacity without compromising computation efficiency and optimization quality. Therefore, there are no apparent limitations to applying our model. Due to the low computation complexity of the weak MLP expert, the overall complexity of `Mowst` is comparable to that of a single GNN model, ensuring that the increased model capability does not come at the cost of more computation. Furthermore, the GNN expert can also be optimized individually following Algorithm 2, allowing the convergence of `Mowst` to be as good as the baseline GNN. Additionally, since our confidence mechanism is applied after executing the entire expert models, there are minimal restrictions on the experts' model architecture. For instance, on very large graphs, we can easily apply existing techniques to scale up the `Mowst` computation, such as neighborhood (Hamilton et al., 2017; Ying et al., 2018) or subgraph sampling (Zeng et al., 2021; Gasteiger et al., 2022) commonly seen in scalable GNN designs.

An interesting question to explore is under what types of graph structures `Mowst` would be most effective. For instance, understanding the properties of the graph that determine the experts' specialization and measuring the relative importance of features and structures would be valuable. Moreover, identifying suitable choices for weak and strong experts in non-graph domains (*e.g.,* time-series analysis, computer vision, *etc.*) is an intriguing direction for future research.

## H.2 BROADER IMPACT

In §2.7, we have discussed interesting extension of `Mowst` into multiple (more than 2) experts. Here, we explore the potential of broader impact of this direction:

**Graph learning task.** Within the graph learning domain, there are two approaches to selecting "progressively stronger" experts for a multi-expert version of `Mowst`:

(1) One simple way is to progressively make the GNN deeper. For instance, an MLP can be considered as a 0-hop GNN, so we can implement expert $i$ as a GNN that aggregates $i$-hop neighbor information.

(2) Another approach is from the architectural perspective. Some GNN architectures are theoretically more expressive than others. For instance, simplified GNN models like SGC (Wu et al., 2019) could serve as an intermediate expert between a weak MLP and a strong GCN. Alternatively, following general theoretical frameworks (*e.g.,* (Zhao et al., 2021)) to construct GNNs with progressively stronger expressive power is possible. In this case, a stronger expert does not necessarily have more layers.

The choice of progressively stronger GNN experts should depend on the graph's properties. For example, if information from distant neighbors is still useful (Alon & Yahav, 2021), it makes sense to follow direction 1 and create deeper experts. Otherwise, if most useful information is concentrated within a shallow neighborhood (Zeng et al., 2021), following direction 2 to define stronger experts with more expressive layer architectures may be more appropriate.

**Other domains.** The concept of weak and strong experts perfectly holds in other domains, such as natural language processing and computer vision, *e.g.,* experts may take various forms of Transformers when considering NLP tasks. From our theoretical understanding in 2, we know that the design of the many-expert `Mowst`is not tied to any specific model architecture, suggesting significant potential for generalizing `Mowst` beyond graph learning. Moreover, the benefits of a multi-expert `Mowst` could be more pronounced when dealing with complex data containing multiple modalities (*e.g.,* graphs with multimedia features (Lyu & Luo, 2022), spatio-temporal graphs (Guo et al., 2019), *etc.*).

**Hierarchical mixture.** Besides the model extension proposed in 2.7, another straightforward way to integrate multiple experts is to construct a hierarchical mixture using the 2-expert `Mowst`. The strong expert in the 2-expert `Mowst` can be any existing MoE model, containing several "sub-experts" controlled by traditional gating modules (*e.g.,* symmetric softmax gating). The interaction between the weak expert and the strong expert remains governed by the confidence-based gating.

