# OpenReview forum: "Mixture of Weak and Strong Experts on Graphs"
_ICLR.cc/2024/Conference — ICLR 2024 poster_

### Official Review · Reviewer_jetB · 2023-10-29

**Soundness:** 3 good
**Presentation:** 1 poor
**Contribution:** 2 fair
**Rating:** 5
**Confidence:** 3

**Summary:**

In this paper, the authors leverage the idea of using mixture-of-experts models to improve the model capacity for graph neural networks (GNNs). Since the GNNs are both expensive and hard to optimize, the authors propose mixing a light-weight multi-layer perceptron (MLP), with an off-the-shelf GNN rather than modeling each expert as a GNN. Here, the MLP specializes in extracting the rich self-features of nodes and it is referred to a weak expert, while the GNN exploits the structure of neighborhood and is called a strong expert. To aggregate these two models, the authors introduce a biased gating function towards the GNN predictions based on a novel "confidence" mechanism, and name this model "Mowst"

For inference, the authors run the weak expert first and obtain a prediction. If the confidence score of that prediction exceeds some random threshold, then it is selected as the final prediction. Otherwise, they continue to run the strong expert, and use the prediction of that expert.

For training, they propose the loss function $L_{Mowst}$ by firstly computing the losses of two experts separately, and then combining them via the confidence function $C$.

In addition, the authors also introduce a variant of the Mowst model in which they use the confidence function $C$ to combine the predictions of the two experts, and then calculate a single loss.

The authors demonstrate that both models are at least as expressive as the MLP or GNN alone but with a comparable computional cost. Finally, they empirically show that these models have significant accuracy improvement on 6 standard node classification benchmarks.

**Strengths:**

1. Originality: the idea of using mixture of weak and strong experts with the confidence mechanism is novel.
2. Quality: the authors provide both theoretical and empirical results to demonstrate the effectiveness of the proposed model "Mowst".

**Weaknesses:**

1. Clarity: the paper is not well-written.
- The authors should emphasize the problem they would like to solve more clearly in the introduction section.
- The presentation of all propositions and theorems is informal. Additionally, the statements of Propositions 2.5, 2.6, 2.7 are so confusing (see Question section).
- The notations used in this paper are difficult to digest (see Theorem 2.4 and Corollary 2.4.1).

2. As far as I understand, with the confidence gating function, we can only leverage two experts in the Mowst without being able to use multiple experts as in previous work. Therefore, the ability to scale up the model capacity is quite limited.

3. The experiments in Section 4 do not show the ability to scale up the model capacity of the Mowst model and its variant.

**Questions:**

1. Below Proposition 2.2, the authors should either present formal formutions of variance and negative entropy functions or give references for these functions.

2. In Section 2.3, are indistinguishable self-features are necessarily the same? The authors should explain more about the term 'indistinguishable'.

3. In Proposition 2.5, does the loss function $L_{Mowst}$ upper bound its counterpart $L^*_{Mowst}$ for any choice of confidence function $C$?

4. In Proposition 2.6 and Theoremm 2.7, the authors should illustrate the concept of expressiveness mathematically.

5. At the end of page 1, the authors claim that the sparse gating function may make the optimization harder due to its discontinuity. I would like to emphasize that not all sparse gating functions are discontinuous. For instance, a temperature softmax gating function in [1] is a sparse yet continuous gating function.

6. Could the authors please explain more clearly why the MLP denoises for the GNN during training?

7. In Algorithm 2, what methods do the authors use to learn the MLP weights and the GNN weights? And what are the convergence rates of those parameters?

8. Why do the authors need the confidence $C$ to be quasiconvex? What happens if $C$ is not quasiconvex?

9. What are the main challenges of using the Mowst model?

10. In Section 3, the authors should cite more relevant papers regarding symmetric gating functions in mixture-of-experts models, namely [1], [2], [3], [4].

**Minor issues**:

1. The abbreviation 'GCN' in page 6 has not been introduced.
2. Grammatical errors: 'the number graph convolution' (Section 1).
3. After the statement of each result, the authors should give references to location (within the paper) of the corresponding proofs.

**References**

[1] X. Nie. Dense-to-Sparse Gate for Mixture-of-Experts.
[2] H. Nguyen. A General Theory for Softmax Gating Multinomial Logistic Mixture of Experts.
[3] H. Nguyen. Statistical Perspective of Top-K Sparse Softmax Gating Mixture of Experts.
[4] H. Nguyen. Demystifying Softmax Gating Function in Gaussian Mixture of Experts.

---

> ### Author Response · Authors · 2023-11-16
> **Authors' Response (Part 1)**
>
> We greatly appreciate the valuable feedback from the reviewer. We are actively preparing a revision to address the questions and include the suggested citations. Detailed responses to each question / concern are presented in the following.
>
> ## Weakness 1
>
> > The authors should emphasize the problem they would like to solve more clearly in the introduction section.
>
> Thanks for the suggestion. We will update the presentation accordingly in our revision.
>
> As summarized in the first sentence of paragraph 3 in *Introduction*, our goal is “*to improve [GNN] model capacity without significant trade-offs in computation overhead and optimization difficulty*”. The two challenges of “computation overhead” and “optimization difficulty” are illustrated in paragraph 2 of *Introduction*.
>
> > The presentation of all propositions and theorems is informal.
>
> > The notations used in this paper are difficult to digest
>
> Our theoretical statements are formal and all results have been **rigorously proven** in *Appendix B*. Nevertheless, we appreciate the feedback that some terms could be more clearly described. See our response to the "Questions" section below.
>
> We will also adjust the structure of *Section 2.3* to facilitate readers' better understand.
>
> ## Weakness 2
>
> > with the confidence gating function, we can only leverage two experts in the Mowst without being able to use multiple experts as in previous work.
>
> > Therefore, the ability to scale up the model capacity is quite limited.
>
> The confidence function is **not** an issue. In *Section 2.7*, we have described the detailed methodology to generalize Mowst to more than 2 experts (mathematical derivation is in *Appendix C.2*). In this case, we have a series of “progressively stronger” experts rather than just a weak and a strong one.
>
> To summarize the idea: the generalization to multiple experts follows a **recursive formulation**, where the 2-expert case is the fundamental building block. We use a 3-expert example for illustration (expert 1 weaker than expert 2, and expert 2 weaker than expert 3). We first combine experts 2 and 3 according to the 2-expert Mowst design. Then we regard the sub-system of experts 2 and 3 as a “meta-model”. Finally, we combine expert 1 with the “meta-model” again according to the 2-expert Mowst design. Under such recursion, the **confidence mechanism still applies**: we first compute expert 1’s confidence to decide if we want to proceed to experts 2 & 3. If so, we compute expert 2’s confidence to decide between the predictions of experts 2 & 3.
>
> The exact **Mowst loss function under any number of experts** is shown in *Equation 40, Appendix C.2*. Finally, the analysis on the 2-expert case (*Section 2.3* to *2.6*) can be automatically generalized to the multi-expert case. Thus, we **already have a comprehensive design for the multi-expert Mowst**.
>
> *Side note*: In addition to the above recursive formulation, there is another straightforward way to integrate many experts. We can use the 2-expert Mowst to construct a hierarchical mixture. The strong expert of the 2-expert Mowst can be an existing MoE model. Thus, our strong expert now contains many “sub-experts” which can be controlled by existing symmetric gating modules.
>
>
> ## Weakness 3
>
> > The experiments in Section 4 do not show the ability to scale up the model capacity of the Mowst model and its variant.
>
> Due to resource and time constraints, we have only empirically evaluated the 2-expert case. Even without the many-expert results, we believe the 2-expert Mowst has already demonstrated a significant enhancement to the existing literature. Currently, there lacks a good MoE design in the graph learning domain, state-of-the-art GNN-based MoE models (e.g., GraphMoE in *Table 1*) or GNN-based ensemble models (e.g., AdaGCN in *Table 1*) do not show significant accuracy improvements on realistic graphs, and may even significantly increase the computation complexity compared with the vanilla GNNs. We have shown that even the basic 2-expert Mowst can significantly and consistently improve the accuracy of state-of-the-art, with computation cost comparable to a single vanilla GNN. Thus, the current experiments have provided **solid evidence on significantly improved model capacity**.
>
> Nevertheless, we thank the reviewer for mentioning the evaluation on multi-expert Mowst. We agree that it is a promising direction given the good performance already achieved by the 2-expert version. We will leave such evaluation as future work if we don’t have enough time or resources during the rebuttal.

---

> ### Author Response · Authors · 2023-11-16
> **Authors' Response (Part 2)**
>
> ## Q1
>
> > ​​formal formulations of variance and negative entropy
>
> Please refer to *Appendix B.1.1* for the exact mathematical expression of variance and negative entropy. We omitted the equations in the main text because none of our analysis in *Section 2* relies on a specific form of $D$ (*Definition 2.1* has specified all requirements on $D$).
>
> ## Q2
>
> > are indistinguishable self-features necessarily the same?
>
> *Theoretically*, the self-features being indistinguishable means that they are exactly the same. This is because, theoretically, MLPs are universal approximators (Hornik et al., 1989 in original submission), capable of approximating any function to an arbitrary precision. Thus, even if two nodes have slightly different self features, the MLP can in theory distinguish them and classify them as having *any* two labels.
>
> Therefore, we will clarify in *Section 2.3* that here we refer to identical self-features.
>
> As a side node (*not affecting theoretical analysis*), in practice, an MLP tends to generate similar predictions for similar features (due to the generalization of neural networks). So practically, “indistinguishable” refers to features that are so similar that the MLP generates the same predictions. Thus, the insights from *Theorem 2.4* and *Corollary 2.4.1* help us understand the practical behaviors of the two experts on clusters of nodes with similar self-features.
>
> ## Q3
>
> > In Proposition 2.5, does the loss function $L_{Mowst}$ upper bound its counterpart $L_{Mowst}^*$ for any choice of confidence function $C$?
>
> Yes. The upper bound holds as long as the output of $C$ is between 0 and 1. The proof is in *Appendix B.1.4*. In short, $C$ and $1-C$ lead to a convex combination of the experts’ loss (for Mowst) or prediction logits (for Mowst*). The upper bound is thus a property of the cross-entropy loss function (which is a convex function).
>
> Since $C$ defined by *Definition 2.1* satisfies the above condition, we didn’t specify any additional condition on $C$ in the statement of *Proposition 2.5*.
>
> ## Q4 (Updated from authors' original response)
> > In Proposition 2.6 and Theorem 2.7, the authors should illustrate the concept of expressiveness mathematically.
>
> Thanks for the suggestion. We will clarify the following definition in our revision:
>
> The "expressive power" **follows the standard definition** in the literature (e.g., [a]) to describe *the neural network's ability to approximate functions*.
>
> In our case (*Propositions 2.6, 2.7, 2.8*):
> * If model A is as expressive as model B, then for any model B applied on any graph, we can find a corresponding model A that generates *the same predictions* as model B.
> * If model A is more expressive than model B, then 1) model A is at least as expressive as model B, and 2) there *exists* a graph such that we can find a model A whose predictions are different from the predictions of any model B.
>
> In the proof (*Appendix B*), to show "as expressive as", we construct confidence functions such that Mowst generates exactly the same outputs as any of its experts. Then to show "more expressive than", we construct a graph and a corresponding Mowst-GCN such that the Mowst-GCN can classify more nodes correctly than any GCN alone.
>
> Here GCN refers to "Graph Convolution Network" (Kipf et al., 2016 in original submission).
>
> Reference
>
> [a] Lu et al. The Expressive Power of Neural Networks: A View from the Width. In NeurIPS 2017.
>
> ## Q5
>
> > not all sparse gating functions are discontinuous. For instance, a temperature softmax gating function in [1] is a sparse yet continuous gating function.
>
> Thanks for mentioning the related work [1]. We will cite it in our revision and clarify the statement regarding sparsity. [1] addresses the discontinuity in sparse gating by implementing a novel dense gate whose output adaptively evolves to a one-hot distribution when training progresses.
>
> However, we would like to clarify that the discontinuity and optimization difficulty are still **prevailing issues in popular MoE designs**. For example, see the discussion in a recent review paper by Google Brain: “A review of sparse expert models in deep learning” (cited as Fedus et al., 2022 in the original submission).

---

> ### Author Response · Authors · 2023-11-16
> **Authors' Response (Part 3)**
>
> ## Q6
>
> > Could the authors please explain more clearly why the MLP denoises for the GNN during training?
>
> The denoising process is described in *Section 2.4*, and we are happy to explain this interesting behavior from another perspective.
>
> Let’s start from the fundamentals. According to theoretical analysis in *Section 2.3*, under the confidence-based loss (*Equation 1*), the MLP expert will be assigned to nodes on which it can outperform the GNN. i.e., such nodes contain rich self-feature information but may also have structural noises. The remaining nodes are assigned to the GNN and they would contain useful structural information. Let the sets of nodes assigned to the MLP and GNN be $V_1$ and $V_2$, respectively. There are two potential benefits provided by Mowst:
> * **Specialization**: We match different parts of the data to the best-fit expert model. So naturally, $V_1$ is matched with MLP and $V_2$ with GNN.
> * **Denoising**: Without Mixture-of-Experts, a single GNN model will be shared between both $V_1$ and $V_2$. The noises in $V_1$ will negatively affect the shared GNN model (e.g., generating harmful gradients as described in *Section 2.4*), resulting in degraded performance even on the clean data $V_2$. When the MLP filters out $V_1$ from the GNN training set, the GNN can improve its performance on $V_2$ since it is no longer affected by the harmful gradients from $V_1$.
>
> In the normal case, we would expect that $V_1$ and $V_2$ both contain a large number of nodes and thus they each capture a meaningful distribution of the training data. In this case, both “specialization” and “denoising” contribute to the model performance boost.
>
> **In our case, we observe another possibility**. Due to the weak-strong combination, it can be much harder for the MLP to take charge of a node than the GNN. Consequently, $V_1$ may possibly only consist of a very small amount of data -- e.g., a few outliers or noisy nodes in the training set. Thus, $V_1$ does **not** really represent a meaningful data distribution. Optimizing an MLP on $V_1$ just results in **overfitting** (mentioned in *Section 2.4*). In other words, the MLP does not truly “specialize”. Even so, the interesting part is that we still observe significant accuracy improvement comparing Mowst with the baseline GNN. This means **“denoising” by MLP plays a critical role to improve the model quality of the GNN expert**.
>
> *Remark*: Whether $V_1$ consists of a significant portion of the training data or not, the MLP-GNN interaction is governed by the same mechanism theoretically described in *Section 2.3*. When MLP overfits a small $V_1$ set, the GNN should have already converged to a reasonable model (otherwise, the GNN would not be powerful enough to dominate on almost the entire training set). Therefore, we use “**fine-tuning**” in *Section 2.4* to describe the further GNN model improvement due to denoising.
>
>
> ## Q7
> > In Algorithm 2, what methods do the authors use to learn the MLP weights and the GNN weights? And what are the convergence rates of those parameters?
>
> Since the loss defined by *Equation 1* is end-to-end differentiable, we use the standard gradient-based optimizer to learn the MLP and GNN weights. Specifically, in our experiments, we use the Adam optimizer with dropout. See *Appendix A.2 & A.3* for the parameter search methodology.
>
> In practice, the MLP training converges much faster than the GNN training (e.g., on Flickr, the MLP takes 134 iterations to converge while the GNN takes 280, both averaged by 10 runs). The difference in the convergence speed is the main motivation that we use the “in-turn” training strategy in *Algorithm 2*.
>
> ## Q8
> > Why do the authors need the confidence C to be quasiconvex? What happens if C is not quasiconvex?
>
> We have the “quasiconvexity” requirement mainly for ease of theoretical analysis. If $C$ is not quasiconvex, the bound in *Theorem 2.4* does not necessarily hold. The experts will still interact in a meaningful & collaborative way, but less explainable.
>
> Without quasiconvexity, the following theoretical results still hold: the bound of *Proposition 2.5*, and the conclusions on expressive power by *Proposition 2.6*, *Theorem 2.7* and *Proposition 2.8*.
>
> Quasiconvexity is easy to satisfy. Following *Section 2.2*, $C$ is decomposed as the $D$ and $G$ functions. Popular dispersion functions such as variance and negative entropy can satisfy the convexity (and thus quasiconvexity) requirement of $D$. So to make $C$ quasiconvex, we only need $G$ to be a monotonically non-decreasing scalar function (which is very intuitive & reasonable -- model should be more confident if its prediction has higher dispersion $D$).

---

> ### Author Response · Authors · 2023-11-16
> **Authors' Response (Part 4)**
>
> ## Q9
>
> > What are the main challenges of using the Mowst model?
>
> There is no apparent tradeoff to apply Mowst. Due to the low computation complexity of the weak MLP expert, the overall complexity of Mowst is close to that of a single GNN model. Thus, the increased model capability does not come at the cost of more computation. Further, since our confidence mechanism is applied after executing the entire expert model, we have very little restriction on the experts’ model architecture. For example, on a very large graph, we can easily apply techniques to scale up the Mowst computation (e.g., neighborhood or subgraph sampling commonly seen in scalable GNN designs).
>
> One question worth exploring is, on what types of graphs would Mowst work most effectively. For example, what are the properties of the graph that determines the experts’ specialization, and how to measure the relative importance of features and structures. Another question is, what are the suitable choices for weak & strong experts in non-graph domains (e.g., time-series analysis, computer vision, etc. ). These are interesting questions to explore as future work.
>
>
> ## Q10
> > In Section 3, the authors should cite more relevant papers regarding symmetric gating functions in mixture-of-experts models, namely [1], [2], [3], [4].
>
> We are happy to include the mentioned citations in our revision. We have discussed [1] in the response to Q5. [2,3,4] all present deep theoretical understanding on the symmetric softmax gating. [2] proposes a novel class of modified softmax gating functions to address the slow parameter estimation rate under the vanishing experts’ parameters scenario. [3] performs novel theoretical analysis on the top-k sparse gating, which shows no tradeoff between model capacity and model performance (computation cost & convergence rate), efficacy of the top-1 gate on parameter estimation and the implication on estimating true number of experts. [4] proposes a novel aspect to analyze the convergence rates under Gaussian MoE and proposes a novel Voronoi loss function.
>
> The above citations are all about symmetric gating. They are clearly different from Mowst because of Mowst’s unique weak-strong combination.

---

> ### Author Response · Authors · 2023-11-20
> **Follow-up from Authors: New Revision**
>
> Dear reviewer,
>
> We would like to follow up with you regarding our new revision.
>
> We have significantly improved our paper according to you suggestion. We have:
> * revised Introduction to state the target problem and main motivation more clearly
> * improved description throughout the technical section (Section 2) to make the theoretical statements more readable and easier to understand
> * added missing definitions for theoretical statements
> * rewritten the description on the "denoising" process
>
> We have also included all the suggested references to make our "Related Work" more complete.
>
> Please do not hesitate to let us know of any further comments or questions.
>
> Thanks!
>
> Authors

---

> > ### Comment · Reviewer_jetB · 2023-11-21
> >
> > Dear Authors,
> >
> > Thanks for your detailed response, which have addressed many of my concerns. The revision of your paper looks better than the original version. However, based on your response, I am still not convinced about the ability to scale up the model capacity of the proposed model, which is the main goal of this work. I agree that the accuracies for the 2-expert Mowst model and its variant are slightly improved compared to previous methods, but we cannot guarantee that these accuracies will be significantly improved when the number of experts increases. Therefore, I suggest that the authors should study the multi-expert Mowst more extensively, and conduct more experiments on that model to strengthen the paper. For those reasons, I decide to keep my score unchanged.
> >
> > Thank you,
> >
> > Reviewer jetB

---

> > > ### Author Response · Authors · 2023-11-22
> > > **Authors' Follow-up on Criteria**
> > >
> > > Thanks for going over our response and revisions. We are glad that we have addressed many concerns.
> > >
> > > We encourage the reviewer to be open-minded about **contributions already made** in the paper. While the question on many-expert extension is valid, we believe it is a **future work** direction and should not be a significant drawback under the current scope.
> > >
> > > Our objective is to “*improve model capacity without significant tradeoffs in computations overhead and optimization difficulty*” (Introduction, paragraph 3), instead of “*scaling up*” w.r.t. number of experts (based on reviewer's response).
> > >
> > > ## Main focus & scope: 2-expert model
> > >
> > > We have made it clear that we mainly focus on the 2-expert model. e.g.:
> > > * Title: “Weak & strong” indicates 2 experts. Multiple experts are referred to as “progressively stronger” in the short Sec 2.6.
> > > * Abstract: we describe our techniques with one MLP and one GNN, without mentioning many experts.
> > > * Introduction: we do not use the word “*scale up*” which can be ambiguous to imply many experts. Instead, we *“**improve** model capacity*”, where experiments indeed have validated such improvement.
> > > * Sec 2, first sentence: “*our discussion mainly focuses on the 2-expert Mowst*”.
> > >
> > > While we discuss the generalization to multiple experts in a brief Sec 2.7, the main purpose is to make the paper more complete. It should be clear that our current scope is on the 2-expert Mowst.
> > >
> > > Nevertheless, we appreciate the suggestion. We agree multi-expert extension is a promising **future** direction.
> > >
> > > ## Significant contribution by the 2-expert design
> > >
> > > > the accuracies for the 2-expert Mowst model and its variant are slightly improved compared to previous methods
> > >
> > > While the standard for “accuracy improvement” can be subjective, we believe the accuracy gain by the 2-expert Mowst is **clearly significant, strong & consistent**, by comparing with similar papers in the field.
> > >
> > > ### **Strong evidence on significant model capacity improvements**
> > >
> > > We evaluate Mowst with 4 backbone GNNs (plus 1 GNN variant) on 6 large graphs. In Tables 1, 4 & 5, comparing Mowst with its corresponding GNN baseline, the accuracy improvement averaged over all datasets are:
> > > * GraphSAGE: **1.04** (Table 1)
> > > * GCN: **0.73** (Table 1)
> > > * GIN: **6.20** (Table 5)
> > > * GIN-skip: **1.19** (Table 5)
> > > * H2GCN: **1.05** (Table 4)
> > >
> > > Comparing the best accuracy among all 11 baselines with the best accuracy among all 5 Mowst models, the average accuracy improvement is high: **1.24**.
> > >
> > > |  | Flickr | ogbn-arxiv | ogbn-products | Penn94 | pokec | twitch-gamer | Avg gain  |
> > > |---|---|---|---|---|---|---|---|
> > > | Best of all baselines | 53.86±0.37 | 78.50±0.14 | 71.88±0.32 | 82.71±0.67 | 80.89±0.16 | 65.70±0.20   |  |
> > > | Best of all Mowst | 55.48±0.32 | 79.38±0.44 | 72.52±0.07 | 84.56±0.31 | 83.02±0.30 | 66.03±0.16   | |
> > > | (Gain) | (+1.62) | (+0.88) | (+0.64) | (+1.85) | (+2.13) | (+0.33) | **+1.24** |
> > >
> > > Our accuracy improvement is higher than many well-cited papers [1,2,3,4,5,6] in top conferences. Such cross-paper comparison is reasonable since all papers perform the same node classification task on similar graphs.
> > >
> > > ### **Contributions on model design & understanding**
> > >
> > > Combining a weak & a strong expert is a new concept in the field of **both MoE and graph learning**. It provides a new aspect to understand GNN's capacity and the interaction between features and graph structure. Such new understanding is valuable and has been sufficiently conveyed via the design and analysis (covering optimization behavior, expressive power & computation cost) of the 2-expert Mowst.
> > >
> > > ---
> > >
> > > ## State of affairs in mixture & ensemble in graph learning
> > >
> > > A reader in the traditional MoE field may tend to think state-of-the-art mixture models should by default contain many experts. This is reasonable since MoE designs have been successfully scaled up (w.r.t. number of experts) in many NLP and CV tasks.
> > >
> > > However, the situation is **different in the field of graph learning**. Currently, there still **lacks** a widely-used mixture or ensemble design for GNNs with **many experts**. Graph learning imposes **unique challenges** due to
> > > * Convoluted interaction between feature and structure information, and
> > > * Computation and optimization challenges mentioned in paragraph 2 of Introduction.
> > >
> > > Therefore, we believe even with only 2 experts, Mowst has already **significantly pushed the frontiers of mixture models in the field of graph learning**.
> > >
> > >
> > > ---
> > >
> > > [1] Cluster-GCN: An Efficient Algorithm for Training Deep and Large Graph Convolutional Networks. In KDD 2019.
> > >
> > > [2] Predict then Propagate: Graph Neural Networks meet Personalized PageRank. In ICLR 2019.
> > >
> > > [3] DropEdge: Towards Deep Graph Convolutional Networks on Node Classification. In ICLR 2020.
> > >
> > > [4] Decoupling the depth and scope of graph neural networks. In NeurIPS 2021.
> > >
> > > [5] AdaGCN: Adaboosting Graph Convolutional Networks into Deep Models. In ICLR 2021.
> > >
> > > [6] How Attentive are Graph Attention Networks? In ICLR 2022.

---

### Official Review · Reviewer_SJRD · 2023-10-31

**Soundness:** 2 fair
**Presentation:** 3 good
**Contribution:** 3 good
**Rating:** 6
**Confidence:** 4

**Summary:**

This paper introduces a novel prediction strategy designed for scenarios where the model inputs are graphs.  The authors try to improve the model's capacity in graph neural networks (GNN) without raising the computational costs. For making predictions, they introduce two experts: a basic Multi-Layer Perceptron (MLP) and an off-the-shelf GNN. The collaboration between these experts follows a system known as the mixture of experts (MoE), which is referred to as the confidence mechanism.  This gating scheme determines which expert's prediction should be chosen. The proposed solution employs two algorithms for both inference and training. During inference, a prediction is generated for each node, while during training, the MLP and GNN models are updated. The authors demonstrate the superiority of their proposed solution on node classification benchmarks for both homophilous and heterophilous graphs.

**Strengths:**

The paper addresses a significant and challenging problem in graph neural networks: model capacity. It demonstrates good quality, offering clear explanations for mathematical aspects, and the appendix provides helpful content. The technical details of the proposed solution are well-explained. The idea of converting the prediction task into an ensemble technique (here mixture of experts, MoE) is impressive. Although the conventional MoE is not a novel method, model ensemble on graphs has received much attention in recent years.

**Weaknesses:**

The paper, in its current state, exhibits several deficiencies that require attention from the authors. The most significant issue with the paper is the absence of a clear presentation of the details of the proposed solution.

- A Minor Issue: A minor issue I've observed in the paper is that on two pages, a significant portion of the content is dedicated to citations and references rather than the core content. On page 1, approximately 40 percent of the introduction is occupied by references, and a similar issue is present on page 6, where content presentation heavily relies on citations. While I understand that this issue may be related to the template used, it can inconvenience readers who may not wish to review all the references. It would greatly benefit the paper if the authors could pay attention to this issue to enhance the reader's understanding.

- An important source of ambiguity arises in Algorithms 1 and 2. Algorithm 1 is responsible for generating predictions using the MLP or GNN, whereas Algorithm 2 focuses on estimating the parameters of the networks. The paper lacks clarity regarding the process when the MLP and GNN have not been trained, and Algorithm 1 does not contain an initialization step for their parameters or hyperparameters. It is imperative that the paper defines the interaction and collaboration between these two algorithms to address this issue. In particular, it is crucial to explain how Algorithm 1 can produce predictions when the experts have not yet been trained.

- The loss functions in (Eq.1) and (Eq.3) play a crucial role in the training process, as they are meant to be minimized. However, an issue arises in these loss functions because all elements within them are known. Specifically, the confidence, and predictions of both experts and the true labels are all given. Algorithm 1 is responsible for generating confidence and predictions, while the true labels are known. This results in the loss function having a fixed value, and it remains unclear with respect to which parameters the minimization task is intended. As a result, the loss function appears to be independent of the experts' parameters. The paper should address and clarify this issue to ensure a proper understanding of the optimization process.

- Confidence mechanism C: The paper introduces a novel gating variable, referred to as confidence, which holds promise for providing appropriate weightings in the Mixture of Experts (MoE) framework. However, a significant concern arises with the use of the random value q. The strategy resembles sampling techniques such as Metropolis-Hastings, where an acceptance ratio is compared with a generated uniform random number. Nevertheless, the fundamental problem here differs as the model's goal is to choose between experts. While the paper presents an innovative confidence mechanism, it ultimately compares it with a completely random value. This approach may not be a precise method for selecting one of the experts, as the confidence mechanism is inherently tied to the experts' predictions, while the ratio q is an independent value. The paper should address this issue to ensure a more reliable method for expert selection.

- Limitations of the proposed solution based on MLP and GNN: It's important to note that available baselines often face challenges when dealing with graph inputs and have their own limitations. The paper introduces two experts, MLP and GNN. However, the use of MLP as an expert should be explored more thoroughly in the paper, particularly due to its potentially weak prediction quality. For example, if Algorithm 1 consistently generates small values for q, resulting in MLP being selected in most cases, questions arise about the guarantee for the prediction quality of the $M_{owst}$ method. Additionally, the paper should delve into the main advantages of using MLP compared to other potential candidates. The current discussion in the paper does not adequately address this scenario. On the other hand, the limitations of using an off-the-shelf GNN have not been sufficiently discussed. The paper should elaborate on how employing GNN as an expert can effectively address its known limitations. A more comprehensive discussion in these areas is necessary to provide a well-rounded understanding of the model's capabilities and potential challenges.

- Computational complexity: While the paper briefly touches upon the complexity of the proposed model, it is imperative to provide a more comprehensive analysis of its computational efficiency, especially within the experimental context. The Abstract highlights the efficiency of the $M_{owst}$ method, making it essential for the authors to include sensitivity analyses related to the model's complexity in the experiments. This would help in quantifying the trade-offs between model performance and computational resources. Furthermore, the cost associated with the confidence mechanism, even if estimated using a simple MLP, should be clearly explained in the paper. Precise details on the computational cost will aid readers in understanding the practical implications of implementing this mechanism. Providing a more in-depth discussion and analysis of these aspects will enhance the paper's completeness and help readers assess the practical feasibility of the proposed approach.

**Questions:**

See discussions in the weakness section.

---

> ### Author Response · Authors · 2023-11-17
> **Authors' Response (Part 1)**
>
> We thank the reviewer for carefully going through our paper and we greatly appreciate the valuable feedback. We cherish this opportunity to resolve the misunderstanding on our algorithms and to clarify some missing context.
>
> ## Citation formatting issue
>
> > on two pages, a significant portion of the content is dedicated to citations and references rather than the core content.
>
> We thank the reviewer for carefully thinking about how to make our paper more readable. We agree that this is partially an issue with the template. We are working on a revision accordingly.
>
> ## Relation between Algorithms 1 & 2
>
> > The paper lacks clarity regarding the process when the MLP and GNN have not been trained, and Algorithm 1 does not contain an initialization step for their parameters or hyperparameters.
>
> > It is imperative that the paper defines the interaction and collaboration between these two algorithms to address this issue.
>
> > it is crucial to explain how Algorithm 1 can produce predictions when the experts have not yet been trained.
>
> We thank the reviewer for raising this clarity issue. We see that the *numbering* of the two algorithms may have caused this confusion. We will revise this in the revision.
>
> *Algorithm 1* will only be applied **after** the Mowst system has been trained via *Algorithm 2*. And *Algorithm 1* will only be applied on the test set for prediction generation, while *Algorithm 2* will only be applied on the training set for model parameter update. In other words, *Algorithm 1* does **not need an initialization step**. It just executes the forward path of a **trained MLP**, obtains its confidence, and then optionally executes the forward path of a **trained GNN**.
>
> The relation between *Algorithms 1 & 2* are summarized as follows:
> * *Algorithms 1 & 2* do not interact with each other. Execution of *Algorithm 2* (training) does not involve execution of *Algorithm 1* (inference), and vice versa.
> * *Algorithm 2* is mathematically consistent with *Algorithm 1*, in the sense that the training of *Algorithm 2* optimizes the **expected loss** incurred during the inference of *Algorithm 1*. See more details from our response to the "Confidence $C$" question.
>
> *Another aspect of understanding*: *Algorithm 1* defines how to combine the predictions of the MLP and GNN, in order to generate **one single** final prediction for each target node. On the other hand, for training in *Algorithm 2* to proceed, the predictions separately generated by the MLP and GNN are sufficient to compute the loss (as well as the confidence) defined by *Equation 1*, and we don’t need to combine the two predictions into a single one via *Algorithm 1* to execute gradient descent.
>
> *Presentation clarity*: We appreciate that the reviewer raised this question. We will update the description in *Section 2.1* to avoid confusion. Our rationale to describe the inference *Algorithm 1* before the training *Algorithm 2* is that the training objective is designed according to the inference procedure (see response to "Confidence $C$").
>
> *Typo correction*: in *Algorithm 2*: $L$ should be $L_{Mowst}$ defined in *Equation 1*.

---

> ### Author Response · Authors · 2023-11-17
> **Authors' Response (Part 2)**
>
> ## Training procedure & parameters
>
> > an issue arises in these loss functions because all elements within them are known. Specifically, the confidence, and predictions of both experts and the true labels are all given.
>
> > Algorithm 1 is responsible for generating confidence and predictions, while the true labels are known.
>
> > This results in the loss function having a fixed value, and it remains unclear with respect to which parameters the minimization task is intended.
>
> We hope that the response in the above section has helped clarify the confusion here. In summary, *Algorithm 1* is not involved in the training process. The training is to optimize the loss in *Equation 1*, where $p_{v}$ and $p_{v}'$ are generated by the 2 experts. See the description in the "Training" paragraph of *Section 2.1*: "$p_{v} = MLP(x_{v}; \theta)$ and $p_{v}' = GNN(x_{v}; \theta')$" and "*$\theta$ and $\theta'$ are the experts’ model parameters*".
>
> Thus, **$\theta$ and $\theta'$ are our learnable parameters**. This is consistent with our statement in *Algorithm 2* that we "*update MLP (or GNN) weights to $\theta_{r}$ (or $\theta_{r}'$)*". The **loss is not fixed**: update on $\theta$ and $\theta'$ will change the predictions $p_{v}$ and $p_{v}'$, and will subsequently change the confidence $C(p_{v})$. Eventually, it will change the loss $L_{Mowst}$.
>
> *Remark*: We can optionally implement a learnable confidence function $C$ (see the end of *Section 2.2*). In this case, the learnable parameters will additionally include the model weights for the confidence neural network. At the end of the "Training" paragraph in *Section 2.1*, we described that "*if $C$ is learnable, we update its parameters together with MLP’s $\theta$*".
>
> ## Confidence $C$: Sampling via an independent random number is precise
>
> > a significant concern arises with the use of the random value q.
>
> > While the paper presents an innovative confidence mechanism, it ultimately compares it with a completely random value. This approach may not be a precise method for selecting one of the experts, as the confidence mechanism is inherently tied to the experts' predictions, while the ratio q is an independent value.
>
> We thank the reviewer for asking this question and making a connection with Metropolis-Hastings. We would like to clarify that the goal is **not** to estimate a probability distribution. The functionality of *Algorithm 1* is very straightforward: with probability $C$, we accept the MLP’s prediction. With probability $(1-C)$, we reject MLP’s prediction and accept GNN’s prediction. Thus, the independent random variable $q$ under uniform distribution is used to **exactly** realize such functionality, and so *Algorithm 1* is **precise**.
>
> We will clarify in the revision that the random number is used so that MLP is activated with probability $C$.
>
> Finally, we would like to provide some additional context regarding why **the training *Algorithm 2* aligns with the inference *Algorithm 1*.** As stated in *Section 2.1*, the fundamental objective of training is to "*minimize the **expected** loss of inference*". During inference, since the MLP’s prediction is used with probability $C(p_{v})$, the loss incurred by the MLP is **on expectation** $C(p_{v}) L(p_{v}, y_{v})$. Similarly, the expected loss incurred from the GNN side is $(1 - C(p_{v})) L(p_{v}', y_{v})$. This leads to the total expected loss of Mowst being *Equation 1*.

---

> ### Author Response · Authors · 2023-11-17
> **Authors' Response (Part 3)**
>
> ## Mowst discourages the weak MLP from making poor predictions
>
> > if Algorithm 1 consistently generates small values for q, resulting in MLP being selected in most cases, questions arise about the guarantee for the prediction quality of the method.
>
> Such an extreme case is theoretically possible, but **the probability is so low that it does not actually have practical implications**.
>
> Since $q$ is sampled under a uniform random distribution, the probability of $q < C$ is exactly $C$. In addition, $q$ is independently sampled for each node. So if we consider two nodes $u$ and $v$, the probability that MLP is selected on both nodes is equal to the probability that $q_{u} < C_{u}$ **and** $q_{v} < C_{v}$, which is exactly $C_{u} \cdot C_{v}$. This means it is **extremely unlikely** that we select MLP simultaneously on these two nodes if both have low confidence. For example, $C_{u} \cdot C_{v} = 0.01$ if $C_{u} = C_{v} = 0.1$.
>
> **Empirical evidence**: In Table 1, we can compare the variance of the model accuracy under 10 runs. It should be clear that the variance of Mowst-GCN and Mowst-SAGE is not larger than that of the baseline GCN and GraphSAGE. Thus, **the randomness introduced by $q$ is not of practical concern**.
>
>
> > the use of MLP as an expert should be explored more thoroughly in the paper, particularly due to its potentially weak prediction quality.
>
> Our confidence mechanism **discourages the poor MLP predictions from being accepted** by Mowst. In fact, one of our fundamental design principles is that “*the weak expert should only be cautiously activated to avoid accuracy degradation*” (paragraph 1, *Section 2*).
>
> Given some target nodes, the MLP predictions being poor means that the MLP has higher loss than the GNN. Thus, Mowst has the incentive to decrease the confidence so that the bad MLP contributes less and the overall loss $L_{Mowst}$ (*Equation 1*) is reduced. Such intuition has **rigorous theoretical guarantee**. According to the analysis in *Section 2.3*, on the part of data where GNN achieves lower loss, the MLP will have 0 confidence -- i.e., the MLP takes no effect. On the part of data where MLP achieves lower loss (e.g., on nodes with noisy neighborhood information), the MLP will have positive confidence, but its value may still be less than 1 (exact $C$ value depends on the learnt confidence function) so that the GNN can still play some role. Thus, our gating is not fair between the two experts, and **inherently favors the GNN** (described as “*desirable bias*” in *Section 2.1 & 2.3*).
>
>
> ## MLP as a suitable weak expert
>
> > the paper should delve into the main advantages of using MLP compared to other potential candidates.
>
> There are several reasons for choosing the MLP as the weak expert:
>
> **MLP is a simplified version of GNN**. Each GNN layer generally performs two main steps: *neighbor aggregation* and *feature transformation*. The *neighbor aggregation* combines the feature vectors from the neighbor nodes into a single embedding vector. Afterwards, such an embedding vector goes through a “*transformation*” module which often just performs linear transformation. If we discard the neighbor information, the “simplified” GNN layer only performs a linear transformation on self-embedding from the previous layer, which is exactly the same as the operation of an MLP layer. More concretely, let’s see how a GCN and a GraphSAGE layer reduces to a standard MLP layer:
> * **GCN**: based on the equation here https://pytorch-geometric.readthedocs.io/en/latest/generated/torch_geometric.nn.conv.GCNConv.html, without neighbors, the term related to the graph adjacency matrix, $\hat{D}^{-1/2}\hat{A} \hat{D}^{-1/2}$ becomes the identity matrix, and so the layer simply performs $X’=X \Theta$, which is exactly the MLP layer operation before non-linear activation.
> * **GraphSAGE**: based on the equation here https://pytorch-geometric.readthedocs.io/en/latest/generated/torch_geometric.nn.conv.SAGEConv.html, without neighbors, the $W_2$ term becomes 0, and the layer performs $x_i’ = W_1 x_i$, which is again exactly the MLP layer operation before activation.
>
> Similar reduction can be applied to many other state-of-the-art GNN architectures, such as **GIN**.
>
> **MLP is efficient to compute**. Among other choices of a weak expert (e.g., a shallow GNN), MLP is probably the most efficient one to compute, since an MLP does not consider neighbors at all. More details on computation complexity can be seen in the reply in the “*Computation complexity*” section.
>
> **MLP is easy to optimize**. Due to its simple architecture, the training convergence of an MLP is both fast and stable. On the contrary, optimizing a GNN is more challenging due to issues such as oversmoothing (Chen et al., 2020a; Li et al., 2018 in the original submission) and over-squashing (Topping et al., 2022; Alon & Yahav, 2021 in the original submission). Such challenges may still exist even if we use a simplified GNN model as a weak model.

---

> ### Author Response · Authors · 2023-11-17
> **Authors' Response (Part 4)**
>
> ## Clarification on GNN expert
>
> > The paper should elaborate on how employing GNN as an expert can effectively address its known limitations.
>
> There are numerous GNN designs in the literature. Different GNN architectures handle different types of information in the neighborhood. For example, GCN performs low-path filtering on the graph signals to smooth out the neighbor features, and GIN simulates graph isomorphism tests to identify meaningful subgraph structures in the neighborhood.
>
> No matter what specific GNN architecture we choose, the following observations generally hold:
> * Noises in the neighborhood, whether they are noisy edge structures (for GIN) or noisy neighbor features (for GCN), cannot be avoided.
> * Some nodes contain sufficiently rich self-features for the classification task, and thus GNN's neighbor aggregation does not provide obvious advantage.
>
> Motivated by the above, Mowst provides a **model agnostic** solution that can generally benefit many different GNN architectures. In other words, **we do not provide a new way to aggregate neighbor information. Instead, we make the existing neighbor aggregation functionality more robust and of higher-quality**.
>
> ## More details on computation complexity
>
> > it is imperative to provide a more comprehensive analysis of its computational efficiency, especially within the experimental context.
>
>
> Empirically, we observe that
> * The computation of a GNN model is much slower than an MLP.
> * The computation of Mowst is as fast as the baseline GNN.
>
> which is consistent with our theoretical analysis.
>
> We will follow up on the exact numbers on execution time, and include them in the revision. We will also include the following details on complexity analysis in the revision.
>
> Theoretical computation complexity analysis is presented in the last paragraph of *Section 2.6*. We provide some more context here. We consider the forward path of the neural network, as the complexity analysis on backward propagation is similar. For a GNN model, as mentioned in the above “*MLP is a simplified version of GNN*” paragraph, each layer aggregates features from the direct neighbors. So consider an $\ell$-layer GNN operating on a target node $v$:
> * The $\ell$-th (i.e., last) layer aggregates information from $v$’s 1-hop neighbors, and outputs a single embedding for $v$ itself.
> * The $(\ell-1)$-th layer aggregates information from $v$’s 2-hop neighbors, and outputs embeddings for each of $v$’s 1-hop neighbors.
> * …
> * The 1st layer aggregates information from $v$’s $\ell$-hop neighbors, and outputs embeddings for each of $v$’s $(\ell -1)$-hop neighbors.
>
> As defined in *Section 2.6*, let $b_{k}$ be the number of $k$-hop neighbors. For the 1st layer, the “neighbor aggregation” step aggregates the $b_{\ell}$ features into $b_{\ell-1}$ ones. The “feature transformation” step further operates on the $b_{\ell-1}$ vectors. For simplicity, let’s ignore the cost of neighbor aggregation. If "feature transformation" is via a linear transformation (common design choice), then it performs matrix multiplication with a $f\times f$ weight matrix. So its cost is $f^2 b_{\ell-1}$.
>
> In general, for layer $k$, the cost of feature transformation is $f^2 b_{\ell-k}$. So the total cost is $\sum_{1\leq k \leq \ell} f^2 b_{\ell-k} = f^2 \sum_{1\leq k \leq \ell} b_{\ell-k} > f^2 (b_{\ell-1} + \sum_{2\leq k \leq \ell} 1 ) = f^2 (\ell + b_{\ell-1} - 1)$. So the cost of an $\ell$-layer GCN is $\Omega(f^2(\ell+b_{\ell-1}))$, as shown in *Section 2.6*.
>
> For an MLP, its complexity can be directly obtained by plugging in $b_{\ell-1} = 1$ in the above equation (see the reasoning in the “*MLP is a simplified version of GNN*” paragraph). So the MLP complexity is $O(f^2 \ell)$.
>
> To compare the complexity of an MLP and a GNN, note that $b_{\ell-1} \gg \ell$. In fact, $b_{\ell}$ can grow exponentially w.r.t. $\ell$ on realistic graphs. Such a phenomenon is well-known as “*neighborhood explosion*” in the GNN literature. In summary, consistent with our conclusion in *Section 2.6*, **the computation cost of an MLP is much less than that of a GNN**.
>
> > the cost associated with the confidence mechanism, even if estimated using a simple MLP, should be clearly explained in the paper.
>
>
> The neural network to compute confidence can be a light-weight MLP (i.e., it can have even lower computation cost than the MLP expert). Even if the confidence MLP has the same structure (and thus the same cost) as the MLP expert, the total cost of the MLP expert *plus* the confidence MLP is **still much less than** the cost of the GNN, according to our above analysis. In the *Section 4* experiments, to simplify hyperparameter search, we let the confidence MLP have an identical structure as the MLP expert. We have noticed that Mowst's execution time is still very close to that of the baseline GNN, thus validating our complexity analysis. We will follow up with the exact timing measurements in the rebuttal as well as in the revision.

---

> ### Comment · Reviewer_SJRD · 2023-11-22
> **Official Comment**
>
> I appreciate the authors for their comprehensive rebuttal. The addressed concerns, especially regarding the clarity of Alg1 and Alg2 and their connection to the loss function, have been notably improved in the new version. Consequently, I am revising my score from 5 to 6.
>
> However, some shortcomings persist. The claim about robustness and higher quality remains unconfirmed in its current state. The limitations of MLP and GNN, particularly when these experts are selected, haven't been precisely addressed. The simplicity of the model with two experts raises questions about its extensibility for a multiple-experts strategy. Additionally, the gating function is still susceptible to the randomness of the parameter q.

---

> > ### Author Response · Authors · 2023-11-23
> > **Thank you**
> >
> > Dear Reviewer SJRD,
> >
> > We are very happy to see that our response has addressed many of your concerns, and thank you for positive feedback!
> >
> > We would like to have a brief follow-up for discussion purposes:
> >
> > We think the questions on robustness, high quality, weak expert and randomness of $q$ are closely related to each other. We would like to provide another aspect in understanding the effect of the random $q$. Basically, the closer the confidence $C$ is towards 0 or 1, the less impact $q$ can have due to random sampling (e.g., if $C=0$ or $1$, then the value of $q$ does not matter at all). Theoretically, during training, the system learns better and better specialization between the two experts (Section 2.4), and the $C$ should progressively accumulate towards the end points 0 and 1 to create a clear data splitting between the two experts. Empirically, we also observe that the $C$ distribution evolves towards 0 and 1, as shown by Figure 2.
> >
> > On the other hand, on nodes with $C$ having a middle value (e.g., $C=0.5$), it is true that random $q$ can create uncertainty in experts' selection. But it is not necessarily true that $q$ will cause much uncertainty in the **final prediction**. When $C=0.5$, the two experts may be similarly powerful on such a node, and thus their predictions may be similar. So it doesn't matter that much what expert is eventually selected by $q$ because we only care about what the final predictions look like.
> >
> > Finally, the issue of random $q$ selection does not exist for our Mowst* variant (Section 2.5). Under Mowst*, we use $C$ to do a soft combination of the two experts' predictions, rather than doing a binary selection. Thus, under Mowst*, the predictions are deterministic for a given model.
> >
> > Once again, thank you for the positive feedback!
> >
> > Best,
> >
> > Authors

---

### Official Review · Reviewer_MV5e · 2023-10-31

**Soundness:** 3 good
**Presentation:** 3 good
**Contribution:** 3 good
**Rating:** 8
**Confidence:** 3

**Summary:**

The paper introduced a mixture-of-experts design to combine a weak MLP expert with a strong GNN. The proposed method relies on a confidence mechanism for controlling the contribution of each expert. The idea is to leverage the MLP for learning from nodes with self-expressive features that do not gain much from its connections and largely resort to the GNN for other cases. Their experiments show the superior performance of their method which is also well supported by several theoretical proofs.

**Strengths:**

1. The paper tackles an interesting problem and proposes a relatively simple solution of mixing a weak model like MLP with a strong expert like GNN.
2. The method is fairly intuitive and the paper is easy to follow.
3. The proposed method is very effective as shown by the experiments. The experimental setup is extensive and shows the effectiveness of the method in several aspects including ablation study, visualization of learned embeddings and training dynamics.
4. The paper is theoretically well-supported through theorems and propositions.

**Weaknesses:**

1. Although it has been briefly mentioned in supplementary section C2, it would be interesting to study the effect of the number of experts on the performance. This leads to further questions regarding the choice of multiple weak experts vs multiple strong experts. Is this even helpful in the given task? I think these are interesting follow up questions one could ask.
2. The github page with code base is private and not accessible via the provided link.
3. Minor grammatical and spelling errors which need to be improved.
4. See questions.

**Questions:**

1. Is there any correlation/relationships between the edge density of the graph dataset and performance of the MLP part of the method? Intuitively, if the graph is sparsely connected, one may assume that the MLP contributes more in that case compared to when the graph is densely connected.
2. What is the $\circ$ symbol in Proposition 2.2? It has not been defined.
3. The task for the experiments mentions "node prediction". Do the authors mean it's a node classification task?

---

> ### Author Response · Authors · 2023-11-17
> **Authors' Response (Part 1)**
>
> We greatly appreciate the positive feedback. Thank you for acknowledging the *effectiveness & simplicity of our solution*, the *clarity of our presentation*, the *significant gains from our experiments* and the *soundness of our theoretical analysis*.
>
> ## Code link
>
> Thanks for checking! We have double-checked the GitHub setting. Please check again under: https://github.com/mowstgnn/mowst
>
> ## Generalization to many experts
>
> > it would be interesting to study the effect of the number of experts on the performance. This leads to further questions regarding the choice of multiple weak experts vs multiple strong experts.
>
> We very much agree that an empirical evaluation on the many-expert scenario is valuable as future work. In addition, thanks for noticing our existing theoretical analysis on the many-expert generalization in *Section 2.7*.
>
> We are more than happy to share our understandings on the potential of the many-expert generalization:
>
> **Graph learning task**: if we limit our thinking within the graph learning domain, there are two directions to choose “progressively stronger” experts.
> 1. One simple way is to progressively make the GNN deeper. E.g., an MLP can be seen as a 0-hop GNN, and so expert $i$ can just implement a GNN aggregating $i$-hop neighbor information.
> 2. Another way is from the architecture perspective. Some GNN architectures are theoretically more expressive than the others. For example, simplified GNN models like SGC [1] can be an intermediate expert between a weak MLP and a strong GCN. Another possibility is to follow some general theoretical frameworks (e.g., [2]) to construct GNNs with progressively strong expressive power. In this case, the stronger expert do not necessarily have more layers.
>
> *The choices of progressively stronger GNN experts should be dependent on the property of the graph*. For example, if neighbors from many hops away still provide useful information [3], then it makes sense to follow the above direction 1 to make a stronger expert deeper. Otherwise, if most of the useful information concentrates within a shallow neighborhood [4], then it may be better to follow direction 2 to define stronger experts as having more expressive layer architectures.
>
> **Other domains**: the concept of weak and strong experts perfectly holds in other domains like natural language processing and computer vision. E.g., experts may become various forms of Transformers when considering NLP tasks. From our theoretical understanding in *Section 2*, we know that the design of the many-expert Mowst does not depend on any specific model architecture, and thus we see **large potential of generalizing Mowst beyond graph learning**. In addition, we believe that the benefits of many-expert Mowst would be larger when the **data is more complicated and contains more modalities** (e.g., graphs with multimedia features, spatial-temporal graphs, etc.).
>
> **Hierarchical mixture**: another straightforward way to integrate many experts is to use the 2-expert Mowst to construct a hierarchical mixture. The strong expert of the 2-expert Mowst can be an existing MoE model. Thus, the strong expert now contains many “sub-experts” which can be controlled by traditional gating modules (e.g., symmetric softmax gating). The interaction between the weak expert and the strong expert is still via the confidence-based gating.
>
> --------------------
> References
>
> [1] Felix Wu et al. Simplifying Graph Convolutional Networks. In ICML 2019.
>
> [2] Lingxiao Zhao et al. From Stars to Subgraphs: Uplifting Any GNN with Local Structure Awareness. In ICLR 2022.
>
> [3] Uri Alon and Eran Yahav. On the Bottleneck of Graph Neural Networks and its Practical Implications. In ICLR 2021.
>
> [4] Zeng et al. Decoupling the Depth and Scope of Graph Neural Networks. In NeurIPS 2021.

---

> ### Author Response · Authors · 2023-11-17
> **Authors' Response (Part 2)**
>
> ## Correlation between edge density and MLP performance
>
> > Is there any correlation/relationships between the edge density of the graph dataset and performance of the MLP part of the method? Intuitively, if the graph is sparsely connected, one may assume that the MLP contributes more in that case compared to when the graph is densely connected.
>
> Thanks for this interesting question! We also believe that there may be some graph properties correlated with the MLP performance. We are running additional experiments on this and will update in the revision.
>
> Intuitively, when the edge density is higher, there could be more neighborhood information. However, it may not be sufficient to just use the edge density alone to reflect the amount of neighborhood information. For example,
> * **Noises in neighbor features** (e.g., heterophily): given two graphs with *identical* edge connections (thus, the same edge density). In graph A, the neighbor node features are useful. In graph B, the neighbor node features are noisy. Then the MLP expert of Mowst will perform better on graph B because the GNN expert is relatively weaker in that case. Realistically, the graph B scenario may correspond to graphs with high heterophily.
> * **Subgraph topology**: the structure / topology of the neighborhood may contain useful information. The amount of structural information, however, is not proportional to the edge density. For example, in a complete graph (each node is connected with all other nodes), the edge density is maximized. However, there is very little structural information, since the neighborhood structures of any two different nodes are identical. On the other hand, if most nodes in the graph have low degree, but there is a small cluster where nodes are densely connected with each other, then the overall edge density is low, but the amount of structural information is high. In this case, it is not the absolute value of edge density but the relative differences in different neighborhood structures that provides useful information.
>
> ## Clarification on math symbol
>
> > What is the $\circ$ symbol in Proposition 2.2? It has not been defined.
>
> It means the “function composition” operation. So $C = G\circ D$ means $C(x) = G(D(x))$ for input $x$. We will clarify this in the revision.
>
> ## Clarification on experimental task
>
> > The task for the experiments mentions "node prediction". Do the authors mean it's a node classification task?
>
> Correct. In the node classification task, the model predicts the class that each node belongs to.

---

> > ### Comment · Reviewer_MV5e · 2023-11-20
> > **Acknowledgement of response**
> >
> > I thank the authors and acknowledge their responses.

---

> > > ### Author Response · Authors · 2023-11-20
> > > **Thank you**
> > >
> > > We appreciate the reviewer's positive feedback and thanks for the reply!

---

### Official Review · Reviewer_DN88 · 2023-11-06

**Soundness:** 2 fair
**Presentation:** 3 good
**Contribution:** 3 good
**Rating:** 6
**Confidence:** 4

**Summary:**

This paper introduces a new method named Mowst for processing graphs, which decouples node features and neighborhood structures by employing a mixture of MLP and GNN. These two models are coordinated through a "confidence" mechanism that activates the GNN when MLP is uncertain, indicated by the spread of prediction logits.  Empirically, Mowst has proven to enhance accuracy on various node classification benchmarks and operates with a computational cost on par with a single GNN model.

**Strengths:**

This authors introduced a new method named Mowst that structures by employing a mixture of MLP and GNN.

The proposed method outperform existing methods in the experiments.

**Weaknesses:**

The manuscript would be improved if the authors could more clearly explain the reasons behind the superior performance of their proposed method compared to existing ones. They attribute their success to the integration of weak and strong experts; yet, it appears that the true advantage may lie in the combination of Multilayer Perceptrons (MLPs) and Graph Neural Networks (GNNs). The distinction is that while MLPs leverage features of individual nodes, GNNs also capitalize on information transmitted across edges. This, I believe, might be the actual contributing factor to their method's effectiveness. To substantiate their claims, the authors should consider conducting additional experiments. Specifically, they could compare the performance of a combination of shallow and deep MLPs, as well as  a combination of shallow and deep GNNs. Such experiments would provide a more convincing validation of their results.

Additionally, the introduction to the datasets used in the study requires enhancement. The authors should provide a detailed description of how each network dataset is constructed and clarify the specific features attributed to the nodes within these datasets.

**Questions:**

1. Are there experimental results show that the superior performance was due to the integration of weak and strong experts?

2. How is the network in each dataset constructed? What are the features attributed to the nodes within these datasets.?

---

> ### Author Response · Authors · 2023-11-17
> **Authors' Response (Part 1)**
>
> We greatly appreciate the valuable feedback from the reviewer. We thank the reviewer for acknowledging the *novelty* of our method as well as the *significant improvements in experimental evaluations*.
>
> ## Choices on weak & strong experts
>
> > The manuscript would be improved if the authors could more clearly explain the reasons behind the superior performance of their proposed method compared to existing ones.
>
> > yet, it appears that the true advantage may lie in the combination of Multilayer Perceptrons (MLPs) and Graph Neural Networks (GNNs).
>
>
> Thanks for thinking deeply into the reasons behind the performance improvement. To **rephrase the question** based on our understanding, we believe that the reviewer is asking why we pick two different architectures for the two experts, rather than the weaker & stronger versions of the same architecture (e.g., a shallow GNN & a deep GNN).
>
> Our conclusion is that **an MLP does not have a fundamentally different architecture than a GNN, and an MLP is indeed a weaker version of a GNN**. Thus, the current results can support our claim that the performance gains come from the combination of weak & strong experts.
>
> More specifically, an MLP is a special version of a GNN -- **it is a shallow GNN with 0-hop aggregation**. Let’s take a closer look at the two Mowst variants evaluated in the experiments in *Section 4*, Mowst-GCN and Mowst-SAGE. We show in the following how **a multi-layer GCN and GraphSAGE both reduce to the standard MLP when we perform a 0-hop, shallow neighbor aggregation**:
> * **GCN**: based on the equation here https://pytorch-geometric.readthedocs.io/en/latest/generated/torch_geometric.nn.conv.GCNConv.html, a GCN layer performs $X' = \hat{D}^{-1/2}\hat{A} \hat{D}^{-1/2} X\Theta$, where $X$ and $X'$ are node feature matrices before and after the GCN operation, $\Theta$ is the layer weight matrix, $\hat{D}$ and $\hat{A}$ are both related to the graph structure (i.e., degree & adjacency matrix). Without neighbors (i.e., 0-hop aggregation), the term related to the graph structure, $\hat{D}^{-1/2}\hat{A} \hat{D}^{-1/2}$ becomes the identity matrix, and so the layer performs $X’=X \Theta$, which is exactly the MLP layer operation (here we consider layer operation before non-linear activation).
> * **GraphSAGE**: based on the equation here https://pytorch-geometric.readthedocs.io/en/latest/generated/torch_geometric.nn.conv.SAGEConv.html, a GraphSAGE layer performs $x_i' = W_1 x_i + W_2\cdot mean_{j\in N(i)}(x_j)$, where $W_1$ and $W_2$ are both weight matrices, and $N(i)$ is the set of neighbors of node $i$. Without neighbors, the $W_2$ term becomes 0, and the layer performs $x_i’ = W_1 x_i$, which is again exactly the MLP layer operation.
>
> A similar reduction procedure can be applied to many other state-of-the-art GNN architectures, such as **GIN**. For this reason, we will also include the Mowst-GIN results in the revision.
>
> > Are there experimental results show that the superior performance was due to the integration of weak and strong experts?
>
> While we believe the current MLP + GNN results can well support our claim on weak & strong experts, we greatly appreciate the reviewer’s feedback and are actively running the suggested experiments. We will clarify the above points in the revision.

---

> ### Author Response · Authors · 2023-11-17
> **Authors' Response (Part 2)**
>
> ## Details on datasets
>
> > How is the network in each dataset constructed? What are the features attributed to the nodes within these datasets.?
>
>
> We thank the reviewer for the suggestion on enhancing dataset introduction. We will incorporate a more detailed discussion on how the datasets were constructed in the revision. Our benchmarks cover a **diverse set of applications with different kinds of node features**. A brief description is as follows:
>
> **Flickr** is from [1]. Each node in the graph represents one image uploaded to the Flickr online community https://flickr.com. If two images share some common properties (e.g., same geographic location, same gallery, commented by the same user), an undirected edge is drawn between the two corresponding nodes. Node features are 500-dimensional bag-of-word representation of the images.
>
> **ogbn-arxiv** represents a paper citation network which is curated by [2]. Each node is an arXiv paper and each directed edge stands for that one paper cites another one. Node features are the 128-dimensional feature vectors obtained by averaging the embeddings of the words in its title and abstract. The embeddings are generated by the word2vec model trained over the MAG corpurs [3].
>
> **ogbn-products** represents an Amazon products co-purchasing Network which is also curated by [2]. Each node is a product sold in Amazon and edges between two nodes indicate that the two corresponding products are purchased together. Node features are 100-dimensional bag-of-words vectors of the product descriptions.
>
> **Penn94** [4, 5] is a friendship network from the Facebook 100 networks of university from 2005. Each node represents a student. The node features are major, second major/minor, dorm/house, year, and high school.
>
> **Pokec** [4, 6] is the friendship graph of a Slovak online social network, where nodes are users and edges are directed friendship relations. Node features are profile information such as geographical region, registration time, and age.
>
> **twitch-gamer** [4, 7] is a connected undirected graph of relationships between accounts on the streaming platform Twitch (https://www.twitch.tv). Each node represents a Twitch user. An edge means that they are mutual followers. Node features include number of views, creation and update dates, language, life time, and whether the account is dead.
>
> -------------
> References:
>
> [1] Zeng et al. GraphSAINT: Graph Sampling Based Inductive Learning Method. ICLR 2020.
>
> [2] Hu et al. Open Graph Benchmark: Datasets for Machine Learning on Graphs. NeuIPS 2020.
>
> [3] Wang et al. Microsoft Academic Graph: When Experts Are Not Enough. Quantitative Science Studies 2020.
>
> [4] Lim et al. Large Scale Learning on Non-Homophilous Graphs: New Benchmarks and Strong Simple Methods. NeuIPS 2021.
>
> [5] Traud et al. Social Structure of Facebook Networks. Physica A: Statistical Mechanics and its Applications 2012.
>
> [6] Leskovec and Krevl. Snap Datasets: Stanford Large Network Dataset Collection. 2014.
>
> [7] Rozemberczki and Sarkar. Twitch Gamers: A Dataset for Evaluating Proximity Preserving and Structural Role-based Node Embeddings. arXiv 2021.

---

> > ### Author Response · Authors · 2023-11-22
> > **Brief update**
> >
> > Dear reviewer,
> >
> > We have concluded the experiments on alternative weak-strong combinations. We are preparing the revision to include the results and will keep you posted today.
> >
> > Thanks,
> >
> > Authors

---

> ### Author Response · Authors · 2023-11-23
> **Revision with new experimental results**
>
> Dear reviewer DN88,
>
> We have uploaded a revision containing new experimental results on a Mowst variant consisting of a 2-layer GCN (weak expert) and a 3-layer GCN (strong expert) in Appendix B.3. We observe that this weak-strong variant can still lead to accuracy improvement, but the scale of such improvement is overall smaller than the original Mowst design with MLP + GNN.
>
> We believe such an observation is consistent with our analysis in the main text. Our interpretations are as follows:
>
> In the original Mowst design, when the two experts specialize, the MLP expert would specialize to nodes with rich self-features but high level of structural noises. The GNN expert would specialize to nodes with a large amount of useful structure information. Following the above logic, in the newly implemented weak-strong variant, the 2-layer GCN expert will specialize to nodes with rich information in the 2-hop neighborhood, but noisy connections in the 3-hop neighborhood. However, in realistic graphs, a shallow neighborhood (e.g., 2-hop) already contains most of the useful structural information [1] (consequently, the accuracy boost by upgrading a 2-layer GNN to a 3-layer one is much smaller than the boost by upgrading an MLP to a GNN). As a result, the 3-hop noises in a 3-layer GNN can be much less harmful than the 1-hop or 2-hop noises. Thus, it would be challenging for the 2-layer GCN to find nodes that it can specialize on. The intuitive explanation is that, the functionality of the 2-layer GCN expert overlaps significantly with the 3-layer GCN expert, and thus the benefits reduces when mixing two experts that play a similar role.
>
> Once again, thank you for your positive review!
>
> Best,
>
> Authors
>
> ---------
>
> [1] Zeng et al., Decoupling the depth and scope of graph neural networks. In NeurIPS 2021.

---

### Official Review · Reviewer_3gAn · 2023-11-06

**Soundness:** 3 good
**Presentation:** 4 excellent
**Contribution:** 2 fair
**Rating:** 5
**Confidence:** 4

**Summary:**

The Mowst approach is a novel way of handling self-features and neighborhood structures in GNNs, using a mixture of weak and strong experts with a confidence mechanism to activate the strong expert in low-confidence regions. The authors demonstrate Mowst's effectiveness on multiple benchmark datasets and provide an analysis of its expressive power and computational efficiency. The contributions of the paper encompass an innovative mechanism for expert collaboration, a variant of Mowst designed for directed graphs, and insights into the training dynamics that foster expert specialization and the gradual partitioning of the graph.

**Strengths:**

- The paper is well-written and organized, with clear explanations of the Mowst approach and its variants.
- The experiments are comprehensive, with ablation studies on several design choices

**Weaknesses:**

**Failure cases**: MLPs can exhibit both incorrect predictions and high confidence levels. This is particularly common when dealing with graph datasets featuring imbalanced classes or containing a small number of outlier nodes that may not be sufficiently representative. The authors didn't discuss the limitations of such cases and how mowest may or may not solve them.

**Clarity:** There are several different design choices other than the proposed version. For instance, the authors do not provide a justification for positioning the weak expert at a lower order while placing the strong expert at a higher order. An alternative approach could involve computing confidence using the GNN and then determining the weight for further combining the MLP results. Another version might not involve confidence computation but instead rely on a learnable module to decide which expert is more suitable. Supervision can be obtained through self-supervision, i.e., comparing with a previously trained MLP and a GNN in terms of predictive accuracy on each instance.

**Experimental justification:** The paper lacks results for Mowst*-GIN in Table 1. Including these results would be valuable for assessing whether Mowst can further enhance performance, especially since GIN is the top-performing model on Penn94. Moreover, demonstrating how Mowst can complement different GNN architectures would strengthen the assessment of its utility.

**Experimental comparison:** It may not be fair to compare Mowst with either MLP or GNN in isolation. A more informative comparison could be made between Mowst and a GNN with an MLP skip connection since both have similar expressiveness (theoretically, when the strong expert is activated) and the same number of model parameters.

**Minor:**

In the abstract, the authors mentioned "GNN" before "... the strong expert is an off-the-shelf Graph Neural Network (GNN)."

**Questions:**

- How does mowest solve the scenarios where MLPs can exhibit both incorrect predictions and high confidence levels?
- How do the alternative design choices  (in Weakness 2) compare to mowest?
- What are the performances of Mowst*-GIN and GNN+MLP sc on Table 2?

---

> ### Author Response · Authors · 2023-11-15
> **Authors' Response (Part 1)**
>
> We greatly appreciate the valuable feedback from the reviewer. We thank the review for acknowledging our presentation quality and overall contributions. In addition to the response below, we are actively preparing additional experiments according to the suggestion.
>
> ## Mowst handles the “failure case” well
>
> > MLPs can exhibit both incorrect predictions and high confidence levels … when dealing with graph datasets featuring imbalanced classes or containing a small number of outlier nodes.
>
> We are glad that the reviewer asked about this interesting scenario. Recall that it is the relative performance of the two experts rather than the accuracy of the MLP itself that determines the MLP’s confidence (*Section 2.3*, second last paragraph). Based on this fact, we conclude the following:
> * **Mowst discourages such a failure case by moderating the MLP-GNN interaction** (case 1 below).
> * When the reviewer’s proposed scenario is unavoidable, **we achieve an overall win of the entire Mowst system at the cost of small failure of the MLP expert** (case 2 below).
>
> We use a case study to illustrate the idea. Detailed theoretical analysis can be found in *Section 2.3*. Let’s say the MLP generates the same prediction on a group of 100 nodes, where 90 nodes belong to class 1 and the remaining 10 belong to class 2. If we consider the MLP model alone, it will learn the dominant class very well and its prediction will be close to $[1, 0]$ -- Let’s say the MLP predicts $[0.9, 0.1]$ on all the 100 nodes.
>
> Thus, the MLP is highly confident since $[0.9, 0.1]$ has high dispersion. But its predictions on the 10 minority-class nodes are wrong. In other words, **such 10 nodes correspond to the reviewer’s failure case**. For sake of discussion, let $L_{1}$ be the average loss on the 100 nodes corresponding to the $[0.9, 0.1]$ prediction.
>
> Now let’s see how the Mowst loss in *Equation 1* changes the MLP and GNN behaviors. There are two possibilities for the GNN:
>
> ### Case 1
>
> The GNN can outperform MLP on the 100 nodes. For example, 5 of the 10 minority-class nodes contain useful neighborhood information, and so the GNN can correctly predict 90 + 5 = 95 nodes. Let $L_{2}$ be the GNN loss on the 100 nodes. Reasonably, $L_{2} < L_{1}$.
>
> To minimize the total loss of *Equation 1*, the MLP has the incentive to reduce its confidence: while doing so will hurt the MLP’s own performance on the 90 majority-class nodes, it will reduce the overall loss of the MLP + GNN system. For simplicity, let’s consider the extreme case: the MLP degrades the initial $[0.9, 0.1]$ prediction to a trivial prediction $[0.5, 0.5]$ (i.e., random guess). Then the MLP loss $L_{1}$ will increase to $L_{1}'$, but $C$ will decrease to $C'=0$ (by *Definition 2.1*). The overall loss becomes $C' \cdot L_{1}' + (1-C')\cdot L_{2} = L_{2}$. On the other hand, the loss corresponding to MLP’s original $[0.9, 0.1]$ prediction is $C\cdot L_{1} + (1-C)\cdot L_{2} > L_{2}$ since $L_{1} > L_{2}$. Therefore, decreasing MLP’s confidence will reduce the total loss of Mowst, and **the failure case is addressed**.
>
> ### Case 2
>
> The GNN cannot outperform MLP on the 100 nodes. For example, some nodes may contain structural noises. In this case, $L_{2} \geq L_{1}$.
>
> Following the analysis on case 1, if the GNN loss $L_{2}$ is *significantly* higher than the MLP loss $L_{1}$, then the MLP may even increase its confidence further to reduce the total Mowst loss. This seemingly exacerbates the MLP failure. However, the interesting part is that **the MLP sacrifices its performance on the 10 minority-class nodes in exchange for the GNN’s greater success on other nodes in the graph**. When a strong expert performs worse than a weak one, this means that the 100 nodes contain a significant amount of noise. Such noise may directly degrade the GNN model quality, resulting in worse prediction on all nodes in the graph (not just the 100 nodes that contain the noise). The strategy of Mowst is to filter out such noise via high MLP confidence, so that the harm from the MLP's failure case is justified by a higher quality GNN model.
>
> ### Remark
>
> in the above cases, why is the tradeoff between the MLP and GNN always worthwhile? This is guaranteed by the design of our loss function (*Equation 1*) and the confidence function (*Definition 2.1*). Since $C$ is learnable (*Section 2.2*), there always exists a set of Mowst parameters such that the overall loss of *Equation 1* is no larger than the loss of an MLP alone or a GNN alone. See the construction of such $C$ in the proof of *Proposition 2.6 & 2.8*.
>
> In summary, Mowst does not always prevent “highly confident wrong predictions” from happening. However, Mowst always handles it in a beneficial way that reduces the overall prediction loss.

---

> ### Author Response · Authors · 2023-11-15
> **Authors' Response (Part 2)**
>
> ## Clarity on alternative designs
>
> We appreciate the reviewer's suggestion on alternative designs. We have also thought about them and here are our reasonings:
>
> ### Order of experts (i.e., MLP-GNN vs. GNN-MLP)
>
> > the authors do not provide a justification for positioning the weak expert at a lower order while placing the strong expert at a higher order. An alternative approach could involve computing confidence using the GNN and then determining the weight for further combining the MLP results
>
> When one expert is more powerful than the other, the order of experts matters. Consider Mowst with two experts A & B. According to the analysis in *Section 2.3* (especially the last paragraph), **when we compute confidence based on expert A, Mowst will be biased towards the other expert B**. In other words, on some training nodes, if expert B achieves a lower loss, then Mowst will simply accept predictions from B and ignore those from A. Otherwise, when expert A achieves lower loss, Mowst may still have some non-zero probability (controlled by learnable $C$) to accept B’s prediction.
>
> When B is the stronger expert with **better generalization capability**, the above **bias is desirable** when applying Mowst on the *test* set. Since GNN generalizes better (Yang et al., 2023 cited in original submission), we prefer the MLP-GNN order (current design) over GNN-MLP (alternative design).
>
> ### Non-confidence based learnable module
>
> > Another version might not involve confidence computation but instead rely on a learnable module to decide which expert is more suitable.
>
> First, we would like to point out that **our confidence function is also learnable**. Basically, all our analysis (*Sections 2.3* to *2.7*) is based on a general class of $C$ defined by *Definition 2.1*, which is decomposed as a dispersion function $D$ and a simple scalar function $G$. We fix $D$ as standard dispersion functions such as variance or negative entropy, and use a light-weight neural network to learn $G$ (see end of *Section 2.2*).
>
> Second, the reviewer’s proposal is reasonable, and we think it is similar to the gating modules in existing Mixture-of-Expert systems (e.g., Shazeer et al., 2017; Wang et al., 2023 cited in original submission). For example, the gating module can be a neural network whose input is the target node feature, and outputs are the weights in front of each expert (the weights can further go through a softmax to ensure the sum across all experts equals 1). Theoretically, such a gate can also simulate our confidence function -- the initial layers of the gating neural net can learn to exactly generate the prediction logits of the MLP expert, and the remaining layers can learn to calculate dispersion and the $G$ function. In this sense, our confidence module can be seen as a specific gate, which is significantly simplified based on the inductive bias of the weak-strong combination. Our confidence-based gating makes the model explainable (*Sections 2.3 & 2.4*), expressive and efficient (*Sections 2.6 & 2.7*). More importantly, it enables Mowst to achieve significantly higher accuracy than GNNs based on traditional gating (e.g., GraphMoE, *Table 1*) due to our simplified design.
>
> ## Experimental justification
>
> > The paper lacks results for Mowst*-GIN in Table 1.
>
> We thank the reviewer for mentioning the Mowst*-GIN results. We agree that including Mowst*-GIN adds value to the paper and we are actively running experiments.

---

> ### Author Response · Authors · 2023-11-15
> **Authors' Response (Part 3)**
>
> ## Experimental comparison
>
> > A more informative comparison could be made between Mowst and a GNN with an MLP skip connection
>
> When selecting baselines, we actually have similar consideration as the reviewer that the baselines should have some flavor of residual connection. We would like to clarify that **existing baselines** such as GraphSAGE, GIN, GAT and GPR-GNN all implement some architectural components to **facilitate the propagation of the node’s self-features**. Nevertheless, we are running additional experiments according to the reviewer's suggestion.
>
> Specifically, for each of the above mentioned baselines:
> * **GraphSAGE**: in each layer, there are two parallel branches (see equation here: https://pytorch-geometric.readthedocs.io/en/latest/generated/torch_geometric.nn.conv.SAGEConv.html). Branch 1 performs linear transformation on the self-feature from the previous layer. Branch 2 performs another linear transformation on the mean-aggregated neighbor features from the previous layer. Results of the two branches are summed and sent to non-linear activation, to generate the layer output. Branch 1 can be seen as a **per-layer residue connection**. In addition, we can understand **an $L$-layer MLP as embedded into an $L$-layer GraphSAGE**, if we see all branch 1 of the L layers as a whole.
> * **GAT**: similar to GraphSAGE, each GAT layer contains two parallel branches (see equation here: https://pytorch-geometric.readthedocs.io/en/latest/generated/torch_geometric.nn.conv.GATConv.html). The main difference from GraphSAGE is that GAT contains additional “attention-based” scaling factors in front of each branch. Likewise, GAT can also be seen as having **dedicated residual connections** integrated into the model.
> * **GIN**: in each layer, the features of all neighbors are summed, while the self-feature is scaled by $(1 + \epsilon)$, where $\epsilon$ is learnable (see equation here: https://pytorch-geometric.readthedocs.io/en/latest/generated/torch_geometric.nn.conv.GINConv.html). Although GIN does not have an explicit residual connection, the model can learn a large $\epsilon$ to **facilitate the propagation of self-features**.
> * **GPR-GNN**: this model generates prediction via $\sum_{k=0}^{K}\gamma_{k} H_{k}$ where $H_{k}$ is the embedding generated from the $k$-hop neighborhood and $\gamma_{k}$ is a scaling factor. GPR-GNN is like Mowst (with a $K$-layer GNN expert) when $k$ only takes values of $0$ and $K$. Thus, GPR-GNN includes an **explicit residual connection** from the $k=0$ branch. The main difference from Mowst is that GPR-GNN's scaling factor $\gamma_{k}$ is applied on all graph nodes, while our confidence-based weighting is customized for each node.

---

> ### Comment · Reviewer_3gAn · 2023-11-21
>
> Thank you for the detalied response! My concerns remain as follows:
>
> For my comment #1, I think you may oversimplify the scenarios. For example, GNN and MLP can be both biased but towards different directions. What if MLP is right about a low-confidence instance while GNN is wrong? That would be another failure case I meant. I think the cases you mentioned are scenarios where one expert absolutely outweighs another.
>
> For my comment #2, I am still not so sure about the design choice here. Another concern raised is that, in the revision, "Our main motivation is to improve model capacity without significant trade-offs in computation overhead and optimization difficulty" - The computation of GNNs in general is not intensive, and if the authors are mostly motivated by the model capacity, there are lots of works on automatic graph learning, e.g., [1], which I think you should compare with. This could be important to justify the usefulness of this algorithm. And they are closely related the way I see it.
>
> For my comment #3: Would be interesting to see the experimental results. I feel skip connection, (depends on the implementation ofc) could be very different from self-features propagation in terms of preventing oversmoothing and localization.
>
> For my comment #4: Does the GNN in your implementation also facilitate the propagation of the node’s self-features? If so, I found it a bit confusing since you pointed out that "However, many widely-used GNNs have a fundamental limitation since they are designed by global properties of the graph. ....". Moreover, what are the number of parameters of Mowst and the baselines methods. If the baselines already have local attention on the node features, then another fair comparison should be that they have the same or close number of parameters.
>
> [1] Design Space for Graph Neural Networks. Jiaxuan You, Rex Ying, Jure Leskovec. NeurIPS 2020.

---

> ### Author Response · Authors · 2023-11-21
> **Authors' follow-up (Part 1)**
>
> Thank you for the follow-up questions.
>
> We would like to provide an update regarding the additional experiments based on our new revision, as well as clarify your main concerns in your follow-up comments.
>
> ## Computation of GNN is intensive
>
> Computation complexity is one of the major blockers in deploying GNNs on realistic large graphs (here we consider large graphs like social networks or citation networks evaluated in our experiments, rather than graphs in the science domain like small molecules). The fundamental reason behind the high computation complexity lies in the recursive neighbor aggregation, which results in the well-know challenge of "neighborhood explosion".
>
> * There are many works in the recent literature aiming to reduce the computation cost of GNNs (e.g., [1,2,3,4,5])
> * One representative realistic example is shown by the seminal work PinSAGE [5], where the GNN model is very small (2 layers, with hidden dimension of 1024 to 2048) and the GNN computation is very expensive (training a single instance takes 3 days on 16 GPUs). The main factor contributing to its high computation cost is the *neighborhood explosion* on a large Pinterest graph.
> * In **Appendix D.3** of the new revision, we have explained in detail
>     * why the GNN computation cost is high
>     * why adding a skip connection is much less efficient than adding an MLP expert in our design, in terms of computation cost.
>
>
> ## Skip connection results
>
> First, we agree that "skip connection in each GNN layer" is very different from "self-feature propagation via an MLP expert", both from the architecture perspective and from the computation cost perspective. The fundamental reason is that skip connection in **intermediate** GNN layers will take effect on the neighbors of the target node as well (in fact, for an $\ell$-layer GNN, skip connection in the layer 1 will operate on the features of the $(\ell-1)$-hop neighbors), while the self-feature propagation via an MLP expert will only operate on the target node itself. For this reason, a GNN augmented with skip connection is in fact more expensive than a Mowst with an MLP expert.
>
> We have shown the detailed derivation of the above argument in **Appendix D.3**
>
> Second, we have implemented new baselines, GCN-skip, GIN-skip as well as H2GCN. For GCN and GIN, we add an additional skip connection on top of GCN and GIN. For H2GCN, it already includes skip connection (specifically, 0-hop, 1-hop and 2-hop propagation in each layer) in its native definition. Results show that
>
> * GCN-skip does not improve accuracy of GCN.
> * GIN-skip improves accuracy of GIN significantly on 3 benchmarks.
> * **Mowst-GIN boosts the accuracy of GIN significantly**
> * **Mowst-GIN-skip & Mowst-H2GCN further boost the accuracy significantly** compared with their corresponding skip-connection baselines.
>
> Please refer to **Tables 4 & 5** in **Appendix B** for details.
>
> ## Fair comparison criteria
>
> We understand your point that total number of parameters can be a reasonable criteria for fair comparison. However, we also believe that the overall computation cost can be another reasonable criteria. The issue here is that there is not a perfect criteria that makes the baseline and Mowst simultaneously have the similar model size and computation cost.
>
> In **Appendix A.4** (with the support from **Appendix D.3**), we reasonably conclude that "keeping the model computation cost similar" is more important in practice (please refer to the PinSAGE example).
>
> We will attach the model size numbers soon.
>
> -------------------------
> [1] Fey et al., GNNAutoScale: Scalable and Expressive Graph Neural Networks via Historical Embeddings. In ICML 2021.
>
> [2] Zeng et al., Decoupling the depth and scope of graph neural networks. In NeurIPS 2021.
>
> [3] Shi et al., LMC: Fast Training of GNNs via Subgraph Sampling with Provable Convergence. In ICLR 2023.
>
> [4] Chen et al., Stochastic Training of Graph Convolutional Networks with Variance Reduction. In ICML 2018.
>
> [5] Ying et al., Graph Convolutional Neural Networks for Web-Scale Recommender Systems. In KDD 2018.

---

> > ### Author Response · Authors · 2023-11-22
> > **Authors' follow-up (Part 2)**
> >
> > ## Deep dive into “failure cases”
> >
> > > For example, GNN and MLP can be both biased but towards different directions.
> >
> > > GNN and MLP can be both biased but towards different directions. What if MLP is right about a low-confidence instance while GNN is wrong?
> >
> > This is a good question! Let us combine together your “failure case” questions in the original and follow-up comments for a joint analysis. We believe you are looking for a conceptual illustration. Thus, in the following, we will **not** focus on the theoretical results in Section 2.3.
> >
> > ----
> > ### **What are the failure cases?**
> >
> > Let’s comprehensively construct possible failure cases as the reviewer suggested. Since this question is mainly for conceptual understanding, our goal is **not** to rigorously cover all possible corner cases. Rather, we use **typical** failure cases that seem controversial to aid our understanding of the experts’ interactions.
> >
> > Looking at the overall training objective of Eq 1, the intuitive “**success cases**” should be:
> > * The MLP loss is low, and its confidence is high: **confident & correct MLP predictions**
> > * The MLP loss is high, and its confidence is low: **unconfident & incorrect MLP predictions**
> >
> > Thus, the typical failure cases are just the opposite of the above success cases: 1. Confident & incorrect MLP predictions, 2. Unconfident & correct MLP predictions
> >
> > Further, since we are dealing with two experts, we can have a more specific definition by considering the experts’ relative strengths. So the **failure cases** can be refined as:
> > * **Case 1**: Confident & incorrect MLP predictions + correct GNN predictions
> > * **Case 2**: Unconfident & correct MLP predictions + incorrect GNN predictions
> >
> > Side note: nodes corresponding to Case 1 should have different self-features than the nodes corresponding to Case 2. Otherwise, their confidence would be the same.
> >
> > Since the balance between the experts are ultimately controlled by the loss, we quantify each term in Eq 1.
> >
> > * **Case 1**: high $C$; high $L_{MLP}$; low $(1-C)$; low $L_{GNN}$.
> > * **Case 2**: low $C$; low $L_{MLP}$; high $(1-C)$; high $L_{GNN}$.
> >
> >
> > ---
> > ### **How to address failure cases?**
> >
> > **Observation**: The commonality between the two failure cases is that for one expert, both its loss and the weight coefficient in front of the loss are high. To address the failure case, the Mowst training should be able to either *reduce the loss*, OR *reduce the weight coefficient*.
> >
> > In summary, Mowst will take the following steps:
> > 1. Update the MLP model to make the predictions on the **case 1** nodes closer to random guess, and
> > 2. Update the confidence function to make it have an “over-confident” shape, so that the MLP expert has higher $C$ on the **case 2** nodes.
> >
> > The two steps do not independently take effect. For step 1, it is a simple task for an MLP to “learn” a random guess. So executing step 1 will not have much effect on the predictions on the case 2 nodes. For step 2, an “over-confident” $C$ means that *it is easier for the MLP to achieve high confidence*. For sake of discussion, let’s manually construct a simple function as an example, where
> > * $C(p) = 0$ if dispersion of $p$ is less than $\tau$, and
> > * $C(p) = 1$ if dispersion of $p$ is larger than $\tau$.
> >
> > If we just consider the above special function, we can make $C$ more “over-confident” with a smaller $\tau$. Note that updating the confidence function in step 2 affects $C$ on both case 1 and case 2 nodes. So we next analyze the joint effect on the case 1 & 2 nodes after executing both steps 1 & 2.
> >
> > In **step 2**, we can reduce $\tau$ until the dispersion of the MLP’s case 2 predictions is *higher than $\tau$* (we can always do so since MLP’s case 2 predictions are correct by definition). Simultaneously, in **step 1**, we need to push MLP’s case 1 predictions more and more towards random guess, until the predictions have a dispersion *lower than $\tau$* (we can always do so since random guess has 0 dispersion).
> >
> > ------
> > ### **Net effect of reduced Mowst loss**
> >
> > Each term in the loss will change after executing steps 1 & 2. For case 1, $C$ will reduce (to 0 under our example confidence function). $L_{MLP}$ will increase. The net effect is *reduced overall loss*: we now have $L_{Mowst}' = 0\cdot L_{MLP}' + (1-0)\cdot L_{GNN} = L_{GNN}$ (by definition of case 1, $L_{GNN}$ is low). For case 2, $C$ will increase (to 1 under our example confidence function). $L_{MLP}$ will remain the same. The net effect is also reduced overall loss: $L_{Mowst}' = 1\cdot L_{MLP} + (1-1)\cdot L_{GNN} = L_{MLP}$ where $L_{MLP}$ is low by definition of case 2.
> >
> > ### **Net effect of improved prediction behaviors**
> >
> > After jointly executing steps 1 & 2:
> > * For case 1, MLP has unconfident incorrect predictions.
> > * For case 2, MLP has confident correct predictions.
> >
> > So through Mowst training, we have **converted the two typical failure cases into two success cases**.
> >
> > ----
> >
> > Please do not hesitate to let us know of any further questions.

---

> > > ### Author Response · Authors · 2023-11-22
> > > **Authors' follow-up (Part 3)**
> > >
> > > ## Relation to automatic graph learning (AGL)
> > >
> > > From the very high level, both our work and AGL perform model selection to improve capacity. However, the approaches are fundamentally different and thus we believe AGL methods are **not directly comparable** with us.
> > >
> > > Let's take GraphGym (mentioned by the reviewer) as an example.
> > >
> > > GraphGym performs model selection in two ways:
> > > 1. Thorough and efficient model architecture search in a large design space consisting of major architecture components.
> > > 2. Model architecture transfer among tasks based on task similarity measure.
> > >
> > > The first way selects a **single** GNN model with the best architecture. This approach is fundamentally different from ours. GraphGym selects a **single model that is applied on all target nodes**, while Mowst adaptively **customizes the suitable expert for each node**. GraphGym's approach belongs to neural architecture search, while our approach falls within the scope of MoE and model ensemble. These are two **very different directions not directly comparable**. We believe suitable baselines falling within our scope are other GNN based MoE or ensemble models (i.e., GraphMoE, AdaGCN in Table 1).
> > >
> > > The second way of GraphGym’s model selection focuses on inter-task transfer. Since Mowst only focuses on a single task. This component of GraphGym is apparently irrelevant to our evaluation.
> > >
> > > ----
> > > ## Further clarification on GNN computation cost
> > >
> > > In the Part 1 response, we have explained technical details on why GNN is expensive for our task. We would like to add some more context here, using GraphGym as an example (we noticed that the reviewer mentioned GraphGym after mentioning computation efficiency).
> > >
> > > Part 1 and Appendix D.3 mention the “neighborhood explosion” challenge. We further clarify that the extent of “**neighborhood explosion**” largely depends on the graph size, as well as the learning task. In the GNN literature, typically, the datasets for graph classification tasks are very different from the datasets for node classification (focus of our paper).
> > > * A graph classification dataset often consists of many graphs, each having a small number of nodes. For example, ogbg-molhiv evaluated in GraphGym consists of 41127 molecule graphs, each having only 25.5 nodes on average.
> > > * A node classification dataset often consists of a single, large graph. For example, ogbn-products evaluated by us contains 2.4M nodes.
> > >
> > > GNN computation cost for node classification on a single large graph (e.g., a graph with 1M nodes) is **much higher** than the cost for graph classification on many small graphs (e.g., 10K graphs each having 100 nodes), for 2 reasons:
> > > * Neighborhood does not “explode” within a small graph. Any target node in a 100-node graph have at most 100 neighbors, but a node in a 1M graph can have thousands to tens of thousands neighbors (e.g., see Table 4 of [1]).
> > > * Node classification on the 1M graph requires 1M node predictions, while graph classification on 10K graphs requires only 10K graph predictions.
> > >
> > > Therefore, GNNs may be efficient if executing graph classification on small graphs, but are **expensive when executing node classification on a large graph**. We focus on node classification and our benchmarks are proposed by papers on large-scale GNNs [2, 3, 4].
> > >
> > > Many graphs evaluated by GraphGym are graph classification tasks on small graphs (e.g., ogbg-molhiv, DD, PROTEINS). The graphs for node classification evaluated by GraphGym are also small (e.g., Cora with only 2708 nodes). The small scale of the graphs is one key reason that GraphGym can afford extensive search in the architecture and hyperparameter space.
> > >
> > > The scalable GNNs mentioned in Part 1 response also all focus on the node classification task on large graphs.
> > >
> > > ---
> > >
> > > ## Additional note on fair comparison criteria
> > >
> > > We have mentioned in Part 1 response and Appendix A.4 that, we pick the “computation cost” criteria for a fair baseline comparison. This criteria is justified by the high computation cost of GNNs.
> > >
> > > We would like to add that **“computation cost” is a common metric for fair comparison** in the GNN literature. e.g., in GraphGym, it mentions in Sec 6:
> > > * ***"Here the computational budgets for all the models are controlled to ensure a fair comparison."***
> > > * *“Specifically, we use a GNN with a stack of 1 pre-processing layer, 3 message passing layers, 1 post-processing layer and 256 hidden dimensions to set the computational budget. For all the other GNN designs in the experiments, the number of hidden dimensions is adjusted to match this computational budget.”*
> > >
> > >
> > > --------
> > > [1] Zeng et al., Decoupling the depth and scope of graph neural networks. NeurIPS 2021.
> > >
> > > [2] Hu et al., Open Graph Benchmark: Datasets for Machine Learning on Graphs. NeurIPS 2020.
> > >
> > > [3] Zeng et al.,  GraphSAINT: Graph sampling based inductive learning method. ICLR 2020.
> > >
> > > [4] Lim et al., Large scale learning on non-homophilous graphs: New benchmarks and strong simple methods. NeurIPS 2021.

---

> > > > ### Author Response · Authors · 2023-11-22
> > > > **Authors' follow-up (Part 4)**
> > > >
> > > > ## Self-feature propagation, global property, localization & specialization
> > > >
> > > > ### **Concern #4**
> > > >
> > > > > Does the GNN in your implementation also facilitate the propagation of the node’s self-features? If so, I found it a bit confusing since you pointed out that "However, many widely-used GNNs have a fundamental limitation since they are designed by global properties of the graph. ....".
> > > >
> > > > For the question: the GNN expert of Mowst always has identical architecture as the corresponding GNN baseline. So if the GNN baseline implements some form of skip connection (e.g., GraphSAGE, or GIN-skip), then Mowst will also have skip connection in the GNN expert.
> > > >
> > > > To address the confusion, we believe it is important to understand that, Mowst is fundamentally designed by **two main ideas**
> > > > * decoupling of the self-features and neighbor structures -- achieved by setting the experts' architecture as MLP and GNN;
> > > > * specialization, localization or "diversified treatments on a per-node basis" (paragraph 1, Introduction) -- achieved by confidence-based gating that controls on/off of each expert on each target node.
> > > >     * **Specialization** of Mowst's MLP and GNN experts on **local** graph regions is discussed in Section 2.4.
> > > >
> > > > The reviewer's confusion seems to come from the relation between self-feature decoupling and localization. We use GPR-GNN as a typical example to illustrate that **self-feature propagation can be based on global properties**.
> > > >
> > > > ----
> > > > ### **Global self-feature propagation**
> > > >
> > > > **GPR-GNN**'s main architecture has been described in our "original" response (Part 4). It implements $h^v = \sum_{0\leq k\leq K} \gamma_k h_k^v$ for each target node $v$, where $h_k^v$ is the embedding vector generated by aggregating $v$'s $k$-hop neighbors, and $\gamma_k$ is a scaling factor for each hop $k$, but **shared** by all nodes $v$.
> > > >
> > > > GPR-GNN is a typical example that self-feature propagation is facilitated (and similar to Mowst, completely decoupled via the $h_0^v$ branch), but the model capacity is still limited by the **global property** of the graph. "Global property" refers to the non-personalized $\gamma_k$ shared by all nodes. This is in clear contrast to Mowst, where our MLP's contribution varies significantly across different target nodes.
> > > >
> > > > -------------
> > > > ### **Further discussion on skip connection (in addition to "follow-up (Part 1)")**
> > > >
> > > > We can generalize the GPR-GNN analysis to other GNN baselines. As mentioned in our "original" response, Part 4, GraphSAGE and GAT (and H2GCN in Appendix B.1 and Table 4) also implement standard skip connection in each layer. In addition, we have implemented skip-connection variants of GIN and GCN (Appendix B.2 and Table 5). What's in common for all these GNN variants is that, their skip connections facilitate self-feature propagation, but the same skip connection functionality (e.g., the same residual weight matrix) is applied to neighborhoods of all target nodes in the same way. This is different from Mowst where our explicit self-feature propagation via the MLP expect can be enabled or disabled depending on the target node.
> > > >
> > > > What contributes to **Mowst's improvement over skip-connection GNN baselines** are as follows:
> > > > * Localization / specialization based on self-feature information achieved activating / deactivating the MLP expert on a per-node basis.
> > > > * The more explicit & thorough decoupling of self-features from neighbor structures via the MLP expert. Specially, skip connections of a GNN may still be largely affected by neighbor information:
> > > >     * Residual embedding in an intermediate GNN layer will be combined with neighbor aggregated embedding before fed into the next layer.
> > > >     * Skip connection in a GNN layer $i$ operates on both the target node features and the $(\ell-i)$-hop features (see details in Appendix D.3.4).
> > > >
> > > > -----
> > > > -----
> > > >
> > > > ## Additional clarification on baselines
> > > >
> > > > > If the baselines already have local attention on the node features, then another fair comparison should be that they have the same or close number of parameters.
> > > >
> > > > Among all baselines on top of which we build a Mowst (GCN, GraphSAGE, GIN, H2GCN), none of them implement attention. Among all baselines in Table 1, GAT implements edge-wise attention.
> > > >
> > > > The fair comparison criteria has been thoroughly discussed in "follow-up" Part 1 & 3.
> > > >
> > > > ----
> > > >
> > > > ### **Differentiating attention \& localization**
> > > >
> > > > Even though edge-attention (e.g., GAT) generates different weights for different neighbors, it is still a **global** operation since the **same attention function** (e.g., the attention weight vector in GAT) is applied on **all neighbors** of a node, **regardless** what the target node is. Thus, attention is different from Mowst's gating which conditionally activates part of the neural network architecture based on the target node.

---

> > > > > ### Author Response · Authors · 2023-11-22
> > > > > **Follow up from authors**
> > > > >
> > > > > Dear Reviewer 3gAn,
> > > > >
> > > > > Thanks again for engaging in the discussion!
> > > > >
> > > > > We value your feedback very much, and thus have spent tremendous efforts addressing your questions in the original and follow-up reviewers. We have also conducted additional experiments as requested, on GIN, Mowst-GIN, GCN-skip, GIN-skip and Mowst-GIN-skip. Results in Table 5 (Appendix B.2) show **significant and consistent accuracy boost across all datasets**.
> > > > >
> > > > > |                   | Flickr      | ogbn-arxiv  | Penn94      | pokec        | twitch-gamer | Avg gain  |
> > > > > |-------------------|-------------|-------------|-------------|--------------|--------------|-----------|
> > > > > | GCN               | 53.86±0.37  | 71.74±0.29  | 82.17±0.04  | 76.01±0.49   | 62.42±0.53   |           |
> > > > > | Mowst-GCN         | 54.62±0.23  | 72.52±0.07  | 83.19±0.43  | 77.28±0.08   | 63.74±0.23   |           |
> > > > > | (Gain)            | (**+0.76**) | (**+0.78**) | (**+1.02**) | (**+1.27**)  | (**+1.32**)  | **+1.03** |
> > > > > | GCN-skip          | 52.98±0.00  | 69.56±0.00  | 76.58±0.53  | 73.46±0.04   | 61.05±0.23   |           |
> > > > > | GIN               | 53.71±0.35  | 69.39±0.56  | 82.68±0.32  | 53.37±2.15   | 61.76±0.60   |           |
> > > > > | Mowst-GIN         | 55.48±0.32  | 71.43±0.26  | 84.56±0.31  | 76.11±0.39   | 64.32±0.34   |           |
> > > > > | (Gain)            | (**+1.77**) | (**+2.04**) | (**+1.88**) | (**+22.74**) | (**+2.56**)  | **+6.20** |
> > > > > | GIN-skip          | 52.70±0.00  | 71.28±0.00  | 80.32±0.43  | 76.29±0.51   | 64.27±0.25   |           |
> > > > > | Mowst-GIN-skip    | 53.19±0.31  | 71.79±0.23  | 81.20±0.55  | 79.70±0.23   | 64.91±0.22   |           |
> > > > > | (Gain)            | (**+0.49**) | (**+0.51**) | (**+0.88**) | (**+3.41**)  | (**+0.64**)  | **+1.19** |
> > > > > | Best of GNN       | 53.86±0.37  | 71.74±0.29  | 82.68±0.32  | 76.29±0.51   | 64.27±0.25   |           |
> > > > > | Best of Mowst-GNN | 55.48±0.32  | 72.52±0.07  | 84.56±0.31  | 79.70±0.23   | 64.91±0.22   |           |
> > > > > | (Gain)            | (**+1.62**) | (**+0.78**) | (**+1.88**) | (**+3.41**)  | (**+0.64**)  | **+1.67** |
> > > > >
> > > > > As the rebuttal period is approaching to its end, we would like to confirm if there is any remaining concern / question regarding our follow-up responses (4 parts in total). We cherish the opportunity to discuss with you.
> > > > >
> > > > > Please do not hesitate to let us know your thoughts. If you are satisfied with our follow-up responses and the new experimental results, please consider raising your score.
> > > > >
> > > > > Thank you very much!
> > > > >
> > > > > Best,
> > > > >
> > > > > Authors

---

> ### Author Response · Authors · 2023-11-23
> **Second follow-up from authors**
>
> Dear reviewer 3gAn,
>
> We have uploaded a new revision after incorporating our responses to your second-round comments. Specifically:
>
> * Appendix E: illustration of how Mowst handles the two failure cases proposed (see "follow-up response", Part 2)
> * Appendix F: explanation of potential alternative designs (see response to original review, Part 2)
>
> As the rebuttal period is about to end, we hope that our **two rounds of comprehensive responses** to your initial and follow-up reviews have addressed your major concerns. We also hope that you find our **new results on Mowst-GIN and the skip connection variants** significant. If so, please kindly consider increasing your score.
>
> Once again, thank you very much for your time during the review and rebuttal period. We appreciate your feedback and look forward to your reply!
>
> Best,
>
> Authors

---

> > ### Comment · Reviewer_3gAn · 2023-11-23
> >
> > Thanks the author for the response. Sorry for didn't make my points clear.
> >
> > By "failure cases", I didn't only mean the modeling ability during training time - surely you can avoid the two failure cases in your discussion. However, the failure case can always happen if MLP takes unseen features, and this is conditionally independent with the training process your have.
> >
> > By "local attention", I didn't mean attention mechanism, but in general, either the self-feature prop or the locationzation by MLP, as opposed to the message-passing integration.
> >
> > The authors did address my concern about Experimental justification and partly about Experimental comparison. I appreciate the authors' response and I know it takes large effort. However, my major concerns remain not fully addressed. I changed my evaluation on the soundness but inclined to maintain my initial rating. I will discuss with other reviewers as well.  Besides, I would encourage the authors to keep the response more concise in the future.

---

> > > ### Author Response · Authors · 2023-11-23
> > > **Round 3: Follow up from authors (Part 1)**
> > >
> > > Dear Reviewer 3gAn,
> > >
> > > Thanks a lot for your reply. We greatly appreciate your time for reading through our long replies, as we understand you are busy.
> > >
> > > We are glad that we have addressed the reviewer’s concerns on experiments.
> > >
> > > ## Failure case follow-up: OOD generalization?
> > >
> > > First, thank you for acknowledging that Mowst can handle the failure cases during training.
> > >
> > >
> > > > However, the failure case can always happen if MLP takes unseen features, and this is conditionally independent with the training process your have.
> > >
> > > By "unseen features" and "conditionally independent", we speculate that you mean the out-of-distribution (OOD) generalization problem. We would like to clarify that it is a very different topic on closing the generalization gap when training and test data distributions differ. Developing a technique to close the generalization gap is out of our current scope.
> > >
> > > We answer the reviewer’s question from the following two perspectives:
> > > 1. Analyzing the test-time model behavior requires a lot of assumptions on the training & test data distribution. Since the reviewer did not specify the training/testing distributions, and how they mismatch, we are unable to quantify the model behavior on the failure cases.
> > > 2. However, we do have a built-in mechanism to alleviate the bad OOD generalization performance of the MLP expert. Our model by design is very conservative about activating the MLP. As discussed in Section 2.3, our confidence-gate is biased, meaning that when the MLP and GNN have similar training performance, Mowst weighs the GNN predictions more. Such bias is motivated by the fact that GNNs generalize better on the test set (Yang et al., 2023).
> > >
> > > Thus, even though we don't know the exact behavior of Mowst on an arbitrarily different test set, we do know that Mowst has its unique mechanism to handle the failure cases for OOD generalization.
> > >
> > > *Side note*: depending on the relative generalization performance of the MLP and the GNN, we may tune the shape of the confidence function to control the bias in the gating as mentioned in the above Point 2. Of course, such tuning is data dependent and can be a future work direction.

---

> > > > ### Author Response · Authors · 2023-11-23
> > > > **Round 3: Follow up from authors (Part 2)**
> > > >
> > > > ## Self-feature propagation & localization
> > > >
> > > > > By "local attention", I didn't mean attention mechanism, but in general, either the self-feature prop or the locationzation by MLP, as opposed to the message-passing integration.
> > > >
> > > > Thanks for clarifying this point.
> > > >
> > > > Self-feature propagation and localization are two different concepts. Specialization / localization relies on self-feature information. However, **an architecture facilitating self-feature propagation may not necessarily localize**. We can see this via an intuitive example:
> > > >
> > > > Suppose nodes with feature $x$ all have class $y$, but these nodes all have very noisy neighborhoods. An effective localization strategy is to gate on self-feature $x$, so that nodes with $x$ will be processed by MLP and nodes with other features will be processed by the GNN. In this case, the GNN will have high quality, since it is not affected by the noises from $x$’s neighborhood. However, if we have a traditional GNN with skip connection, then it will inevitably be affected by the neighborhood noises for nodes with $x$, because neighbor aggregation is **universally applied on all nodes**, including those with $x$.
> > > >
> > > > Thus, **specialization / localization needs to conditionally activate different parts of the architecture**. This is also why we categorize the existing GNNs mentioned in Introduction, paragraph 1 as being based on global property.
> > > >
> > > > * Many **global** models facilitate **self-feature** propagation by different architecture components (e.g., layer-wise skip connection in GraphSAGE, and end-to-end decoupled branches in GPR-GNN).
> > > > * **Localized** models do **not** necessarily have dedicated **self-feature** propagation (e.g., GraphMoE baseline in Section 4).
> > > >
> > > > We have explained the above first point via the GPR-GNN example in Round 2 follow-up response, Part 4. For the second bullet, GraphMoE is a mixture-of-experts system consisting of multiple GNN experts, where the GNN can be any off-the-shelf model not necessarily having skip connections (e.g., vanilla GCN). Just like any other MoE models, each GNN expert in GraphMoE also localizes / specializes on different parts of the graph.
> > > >
> > > > -------
> > > >
> > > > ## Experimental setup
> > > >
> > > > ### **Baselines**
> > > >
> > > > |               | Self-feature propagation | Localization | Table   |
> > > > |---------------|--------------------------|--------------|---------|
> > > > | GCN           | No                       | No           | Table 1 |
> > > > | GCN-skip      | Yes                      | No           | Table 5 |
> > > > | GraphSAGE     | Yes                      | No           | Table 1 |
> > > > | GAT           | Yes                      | No           | Table 1 |
> > > > | GIN           | No                       | No           | Table 1 |
> > > > | GIN-skip      | Yes                      | No           | Table 5 |
> > > > | H2GCN         | Yes                      | No           | Table 4 |
> > > > | GPR-GNN       | Yes                      | No           | Table 1 |
> > > > | AdaGCN        | No                       | No           | Table 1 |
> > > > | GraphMoE-GCN  | No                       | Yes          | Table 1 |
> > > > | GraphMoE-SAGE | Yes                      | Yes          | Table 1 |
> > > >
> > > > ### **Mowst**
> > > >
> > > > For all Mowst models, we do not modify the architecture of the GNN expert at all.
> > > >
> > > > * GNN expert of Mowst: if the “self-feature propagation” for the baseline GNN is Yes (or No), then “self-feature propagation” for Mowst’s GNN expert is also Yes (or No). Likewise for “localization”.
> > > > * Entire system of Mowst: regardless of the GNN architecture, Mowst always have both “self-feature propagation” and “localization”.
> > > >     * “Self-feature propagation” is because we use MLP as an expert.
> > > >     * “Localization” is because of the confidence-gating that selects different expert models (i.e., activates different architecture) for different target nodes.
> > > >
> > > > -----
> > > >
> > > > ## Thanks
> > > >
> > > > As the rebuttal period is ending, we would like to thank the reviewer for engaging in the discussion. We have had a fruitful conversation. We hope that the review can take our last round of reply into consideration for the final rating.
> > > >
> > > > Thank you very much!

---

### Official Review · Reviewer_BGLX · 2023-11-09

**Soundness:** 3 good
**Presentation:** 2 fair
**Contribution:** 2 fair
**Rating:** 6
**Confidence:** 4

**Summary:**

The heterogeneity of node classes raise unique challenges to message passing graph neural networks. Iterative aggregating neighbors can magnify the negative impact of dissimilar nodes on the representation quality. This work tries to deal with this problem by decoupling node presentation into MLP and GNN. MLP mainly captures node-self property and receives much more less impact from neighbor than GNNs. With the help of confidence value calculated based on the output logits of MLP, the proposed approach tends to obtain a better balance on the predictions between homophilous and heterophilous nodes.

**Strengths:**

Strengths:
1. This work proposes to combine MLP and GNN for balancing the predictions for homophilous and heterophilous nodes.
2. Authors add theoretical analysis that are trying to fold out more insight to show the superiority of the proposed loss.
3. Experiments are conducted on various scale of datasets to demonstrate the performance of the proposed method.

**Weaknesses:**

Weaknesses:
1. The presentation quality still has lots of space to improve. That's also the main drawback of this work. The motivation is not clear enough. I guess that the main motivation of this work is about how to overcome the challenges raised by heterophily since most of GNNs work well on homophily network. But there is no literature overview on overcoming heterohpily in the related work section. Moreover, no baselines for dealing with heterophily are included in the experimental section. It's better to illustrate the diagram of learning objective to show how the MLP and GNN collaborate to each other, since they are not applied to mode layers.

Apart from the missing literature, many claims in this work are not clear. Here a incomplete list.
  - "GNNs are both expensive to compute and hard to optimize". By far, many efficient way to run GNNs are already proposed and applied to solve industrial problems. It should not be the main drawback of advanced GNN layer or model. Comparing with MLP, GNN indeed requires more computation cost, but we still have solutions to make it applicable to deal with large scale data. With the claim, what is it relation to the proposed method? It's difficult see that the proposed method has dominated advantage in terms of time complexity since it still relies on the GNN.
   - "a cluster of high homophily, the neighbors are similar, and thus aggregating their features via a GNN layer may not be more advantageous". The "similar neighbors" should be more clear. Assume that this work mainly focuses on node classification according to the experimental tasks, similar neighbors tend to have the same class, but not mean that they should have close similar feature distributions. In this case, aggregating neighbors could bring missing information to the target node for better classification performance.
  - "The MLP can help “clean up” the dataset for the GNN, enabling the strong expert to focus on the more complicated nodes whose neighborhood structure provides useful information for the learning task. " Could you please explain why a weak classifier can provide high-quality dataset for GNNs? From the information presented in this work, I guess it's just use the classification confidence distribution. Suppose that the cleaned nodes by MLP are those the low-quality nodes that have heterophilous neighbors, can the GNNs still distinguish them?
  -  "since the confidence score solely depends on MLP’s output, the system is biased and inherently favors the GNN’s prediction. However, such bias is desirable since under the same training loss". It's difficult to follow the statement. Could you please explain the system is biased by what? why it favors the GNN prediction and the bias is desirable?
  -  "Second, our model-level (rather than layer-level) mixture makes the optimization easier and more explainable." What is the difference between model- and layer-level mixture, and why the optimization is more easier and explainable?
  - "During training, the system dynamically creates a soft splitting of the graph based on not only the nodes’ characteristics but also the two experts’ relative model quality." The splitting of graph is difficult to follow, is the graph divided by nodes or edges?

2. The theoretical analysis can be further improved. For me, it's difficult to follow the conclusion from the series of theorems. It comes with weak connections to the proposed learning objective. It's better to make the point stand out with a clear statement, such as the convergence, stability of confidence value, insight about how to select probable confidence value function. By the way, the exact definition of the confidence value function seems to be missed.

3. The experimental results are incrementally improved comparing to previous methods. Typical baselines for overcoming heterophilous issue are ignored, such as but limited to $H_2$$GCN [1].

References:
1. Beyond Homophily in Graph Neural Networks: Current Limitations and Effective Designs, NeurIPS 2020.

**Questions:**

please refer to question above.

---

> ### Author Response · Authors · 2023-11-14
> **Authors' response (Part 1)**
>
> We appreciate the valuable feedback from the reviewer. We are actively preparing a paper revision according to the suggestions. The main concerns are in the presentation clarity -- specifically, most bullets in “Weakness” are regarding statements in *Introduction*. In the following, we respond to each question based on the **main text of our original submission**.
>
> ## Motivation
> > I guess that the main motivation of this work is about how to overcome the challenges raised by heterophily since most of GNNs work well on homophily network.
>
> The main motivation (*Section 1, paragraph 3*) is to **improve GNN’s model capacity without increasing the computation cost**, where the GNN can be a standard GNN or a scalable one.
>
> Our motivation is not limited to heterophilous graphs. Generally speaking, we target graphs with varying local properties (*Section 1, paragraph 1*). They include
> * graphs with hybrid local-homophily / local-heterophily patterns
> * homophilous graphs with different signal mixing rate in different homophilous local regions
> * graphs, whether they are homophilous or heterophilous, containing some noisy nodes (e.g., outliers, mislabels, etc.)
>
> The reviewer has the right intuition on how Mowst handles Case 1 (heterophily). We further complete the full picture on the remaining 2 cases:
> * Case 2 (homophily): the GNN needs to learn complicated signal filtering functions for various homophilous regions. The MLP expert simplifies the learning task for the GNN expert by eliminating nodes with rich self-features. Such specialization between the MLP and GNN is in principle consistent with the “non-universal treatment” idea of NDLS (cited in *Sections 1 & 3*). Yet, Mowst eliminates NDLS’s strong assumption on “mixing time” and thus relies less on manually tuned heuristics.
> * Case 3 (noises): our confidence mechanism leads to a unique MLP-GNN collaboration mode, elaborated in detail in “denoised fine-tuning” in *Section 2.4*. In short, the small amount of noises in a realistic graph can generate harmful gradients that are significant enough to trap the GNN model in sub-optimality. Our confidence mechanism allows the MLP expert to “overfit” on such noisy nodes, thus providing the GNN with a cleaner dataset to escape its local optimum and further fine-tune.
>
> ## Related work & experiments on heterophilous GNNs
>
> > there is no literature overview on overcoming heterohpily in the related work section. Moreover, no baselines for dealing with heterophily are included in the experimental section.
>
> > Typical baselines for overcoming heterophilous issue are ignored, such as but limited to H2GCN
>
> We appreciate the reviewer’s suggestion on H2GCN. **We have discussed a number of representative heterophilous GNNs**, including H2GCN, in “Related Work” (*Section 3*). **We have also compared Mowst with state-of-the-art heterophilous GNN baselines** in “Experiments” (*Section 4*). We are working on adding the H2GCN baseline in Experiments.
>
> Note that our Related Work (*Section 3*) has covered a broader & more general literature **beyond** just the heterophilous GNNs. This is consistent with our motivation (see above) that our target graphs include but are not limited to those with high heterophily. Specifically, the heterophilous GNNs in *Section 3* adopt different ways to decouple the information from self-features and (multi-hop) structures. This is the rationale behind Section 3’s current subtitle. Nevertheless, in our revision, we will explicitly point out that these related works are suitable for heterophilous graphs.
>
> Justification to current experiment baselines: Platonov et al., 2023 (citation in original submission) shows that even on heterophilous graphs, the “standard” GNNs like GCN, GraphSAGE and GAT achieve comparable or better performance than popular heterophilous GNNs. In addition, GPR-GNN is widely considered as a state-of-the-art heterophilous GNN in the literature (e.g., see Lim et al., 2021 and Platonov et al., 2023 in the original submission). As our benchmarks include both homophilous and heterophilous graphs, we believe the **current set of baselines** (MLP, GCN, GraphSAGE, GAT, GIN, GPR-GNN, GraphMoE, AdaGCN) is **reasonable and comprehensive**. Nevertheless, we will add the H2GCN results in our revision.

---

> ### Author Response · Authors · 2023-11-14
> **Authors' response (Part 2)**
>
> ## Scalable GNNs
>
> > By far, many efficient way to run GNNs are already proposed and applied to solve industrial problems. It should not be the main drawback of advanced GNN layer or model.
>
> > It's difficult see that the proposed method has dominated advantage in terms of time complexity since it still relies on the GNN.
>
> Our claim is to improve the model capacity **at a comparable cost as** the backbone GNN. Here the GNN can be a standard GNN, as well as a scalable GNN designed for large graphs (it should be clear from the analysis in *Section 2* that the GNN expert of Mowst can also be an off-the-shelf “scalable” GNN).
>
> *Clarifying reviewer’s misunderstanding*: we do not improve scalability of existing GNNs. Instead, we **improve their model capacity without hurting their scalability**. The reviewer also agrees that to date, GNNs are still very expensive. Even though there have been successful industrial applications, their model computation costs are still very high (e.g., training of the seminal PinSAGE [a] model (2 layers) takes more than 3 days on 16 GPUs).
>
> In summary, our description, “*GNNs are .. expensive to compute*”, is valid even with the numerous scalable GNN designs existing in the literature. Likewise, it is a valid motivation to improve model capacity without tradeoff in increased computation complexity.
>
> [a] Ying et al., Graph Convolutional Neural Networks for Web-Scale Recommender Systems. In KDD 2018.
>
>
> ## Clarification on similar neighbors
>
> > similar neighbors tend to have the same class, but not mean that they should have close similar feature distributions.
>
> > aggregating neighbors could bring missing information to the target node for better classification performance.
>
> We thank the reviewer for considering various scenarios of homophilous graphs. It is certainly possible that nodes with similar labels can have dissimilar features. We are thus happy to clarify in the revision that “similar neighbors” here refers to neighbors with similar features, which is also a reasonable scenario for homophilous neighborhoods.
>
> To provide more context: the main objective of this sentence in *Introduction* is to help readers build up intuitions for typical use cases. When the neighbor features are dissimilar and provide valuable information (i.e., the case mentioned by the reviewer), the MLP expert will be automatically demoted by our confidence mechanism (see *Section 2.3 & 2.4* for the theories behind this behavior).
>
> ## Clean up the dataset by the weak MLP
>
> > Could you please explain why a weak classifier can provide high-quality dataset for GNNs?
>
> > Suppose that the cleaned nodes by MLP are those the low-quality nodes that have heterophilous neighbors, can the GNNs still distinguish them?
>
> Yes, “cleaning up” the dataset means that Mowst learns a confidence distribution where nodes handled badly by the MLP expert receive low confidence scores. Correspondingly, such nodes have a high $(1-C)$ weight to influence the GNN expert.
>
> The dataset is “clean” because the GNN will not be “distracted” by the rich self-features of the nodes assigned to the MLP. Notice the following to clarify confusion:
> * “Clean nodes” may not equal to “nodes with high GNN accuracy”.
> * The benefit of “cleaning up” is to improve the accuracy of the **entire Mowst system** (MLP + GNN) rather than the accuracy of the **GNN expert alone**.
>
> For conceptual illustration, let’s consider a hypothetical case. We divide all nodes in a graph into two sets:
> * $V_1$ consists of 20% nodes whose self-features are rich enough for the classification task. Yet some $V_1$ nodes may have noisy structural information.
> * $V_2$ consists of the remaining 80% nodes, all containing insufficient self features. Some $V_2$ nodes contain useful structural information.
>
> If we train a baseline GNN on the full graph (i.e., $V_1 + V_2$), it may achieve 90% accuracy on $V_1$ and 65% accuracy on $V_2$, leading to an overall accuracy of 20% * 90% + 80% * 65% = 70%. If we train a Mowst, the MLP expert is assigned with $V_1$, and let’s say it achieves a high accuracy of 91%. The GNN is assigned with $V_2$ (i.e., the cleaned-up dataset), and it may achieve 67% accuracy. Note that the accuracy of the GNN alone drops from 70% on the full dataset to 67% on the “clean” $V_2$. However, the overall accuracy on all $V_1 + V_2$ increases from 70% (baseline GNN) to 20% * 91% + 80% * 67% = 71.8% (Mowst with MLP + GNN).
>
> In the above example, on $V_1$, the MLP expert can achieve higher accuracy than the baseline GNN because MLP is not affected by structural noise (which can exist in both homophilous and heterophilous graphs). “Cleaning up” by removing $V_1$ improves the GNN accuracy on $V_2$ because the GNN
> * does not need to learn how to extract self-features from $V_1$, and
> * is not affected by the structural noises in $V_1$.

---

> ### Author Response · Authors · 2023-11-14
> **Authors' response (Part 3)**
>
> ## Desirable bias
>
> > Could you please explain the system is biased by what? why it favors the GNN prediction and the bias is desirable?
>
> The “bias” is conceptually illustrated in the last paragraph of *Section 2.1*, and formally elaborated in the last paragraph of *Section 2.3*. “Bias” means that it is harder for the MLP to be activated than the GNN. Rephrasing the last paragraph of *Section 2.3*: given a target node, if the GNN loss is lower than the MLP, then Mowst will always take GNN’s prediction and ignore MLP’s prediction. Otherwise, if the MLP loss is lower than the GNN, there is some non-zero probability that the system will still take the GNN’s prediction.
>
> Such bias emerges from our confidence-based mixture mechanism. **The extent of such bias is learnable** based on overall training loss. *Section 2.3* theoretically explains factors controlling the bias.
>
> Right after the reviewer’s quote, we have a sentence explaining the reason why such bias is desirable (*Section 1*, last paragraph): “since under the same training loss, a GNN can generalize better than an MLP (Yang et al., 2023)”. We have again re-iterated the “generalization” explanation in *Section 2.3*.
>
>
> ## Model-level vs. layer-level mixture
>
> > What is the difference between model- and layer-level mixture, and why the optimization is more easier and explainable?
>
> Model-level mixture (our design) means that the two experts “execute independently through their last layer” (*Section 1*, last paragraph), and then the final predictions of the MLP and the GNN are mixed. In other words, the intermediate / hidden layers of the MLP and the GNN do not interact with each other.
>
> Layer-level mixture (described in *Section 3*, paragraph 1) means that in each layer $i$, each expert will generate its own embedding, and the multiple embeddings will be mixed together as the input embedding to layer $i+1$.
>
> Layer-level mixture can achieve high model capacity, but the experts’ interactions are more complicated. Model-level mixture is easier to optimize because we can completely decouple the execution of the MLP and the GNN layers until when the predictions are generated. Thus, we can fix one expert while optimizing the other (*Algorithm 2*) without worrying about the different convergence behaviors of the MLP and GNN (*Section 2.5*, last paragraph). Model-level mixture is more explainable because the mixture happens after the non-linear activations of all intermediate layers, which facilitates rich theoretical analysis in *Section 2.3 & 2.4*. In addition, it is straightforward to attribute the predictions to a particular expert under model-level mixture.
>
>
> ## Graph splitting
>
> > is the graph divided by nodes or edges?
>
> “Splitting” refers to node splitting since we study the node classification task. The splitting means some nodes are assigned with high confidence and thus handled by the MLP, and the remaining nodes are assigned with low confidence and handled by the GNN. “Soft” means the expert assignment is via a continuous confidence value between 0 and 1.
>
> ## Clarify of theoretical analysis
>
> > It comes with weak connections to the proposed learning objective.
>
> > It's better to make the point stand out with a clear statement
>
> > exact definition of the confidence value function seems to be missed.
>
> Thanks for sharing the feedback. We will update the paper to make the conclusions stand out more clearly.
>
> To clarify, we have two series of theoretical analyses:
> * Convergence behavior (*proposition 2.2, proposition 2.3, theorem 2.4, corollary 2.4.1*): we have summarized the important conclusions in 3 bullets in the last two paragraphs of *Section 2.3*. The theoretical analysis reveals factors affecting the balance between the weak & strong experts. It also provides a direct explanation on the two major collaboration behaviors described in *Section 2.4*. We thus believe the theoretical analysis **justifies our design choices (including learning objective)** in a solid way.
> * Total loss & expressive power (*proposition 2.5, proposition 2.6, theorem 2.7, proposition 2.8*): these theoretical statements are self-explanatory. They are essential to help readers understand the various important properties of Mowst.
>
> *Exact definition of confidence function*: See *Definition 2.1* for $C$’s formal definition. One good property of our design and analysis is that they apply to a general class of $C$ -- all functions decomposable as a quasiconvex dispersion function $D$ and a monotonic scalar function $G$. In practice, we set $D$ as the variance or negative entropy function, and $G$ as learnable by a light-weight neural network (last paragraph of *Section 2.2*).
>
>
> ## Experiments
>
> > Typical baselines for overcoming heterophilous issue are ignored, such as but limited to H2GCN
>
> See our "Part 1" response. We are working on the experiments for H2GCN. As mentioned, we have discussed H2GCN in “Related Work” (*Section 3*) in our original version.

---

> ### Author Response · Authors · 2023-11-20
> **Follow-up from Authors: New Results & Revision**
>
> Dear reviewer BGLX,
>
> We would like to give you an update regarding revision and new experimental results.
>
> Regarding your 3 main concerns, we have:
> * uploaded a revision to address your main concerns 1 & 2, and
> * included new results on H2GCN to address your main concern 3.
>
> In the revision, we have significantly improved the clarify of Introduction, specifically for all your bullets mentioned in Weakness 1. We have also rephrased the description on theoretical results to emphasize how they solidly justify our fundamental design choices.
>
> In Table 4 of Appendix B.1, we have included new results showing **significant and consistent accuracy improvement of Mowst-H2GCN over H2GCN** (as well as Mowst-GIN over GIN) on the three large-scale heterophilous benchmarks.
>
> For your convenience, we show the results regarding H2GCN as follows.
>
> |   | Penn94  | pokec  | twitch-gamer  | Avg gain |
> |---|---|---|---|---|
> | GCN | 82.17±0.04 | 76.01±0.49 | 62.42±0.53 | - |
> | Mowst-GCN | 83.19±0.43 | 77.28±0.08 | 63.74±0.23 | - |
> | (gain)  | (+1.02)  | (+0.29)  | (+0.83)  | (+0.71) |
> | GraphSage | 76.75±0.52 | 75.76±0.04 | 61.99±0.30 | - |
> | Mowst-Sage | 79.07±0.43 | 77.84±0.04 | 64.38±0.14 | - |
> | (gain)  | (+2.03)  | (+1.33)  | (+1.05)  | (+1.47) |
> | GIN | 82.68±0.32 | 53.37±2.15 | 61.76±0.60 | - |
> | Mowst-GIN | **84.56**±0.31 | 76.11±0.39 | 64.32±0.34 | - |
> | (gain)  | (+1.88)  | (+22.74)  | (+2.56)  | (+9.06) |
> | **H2GCN**  | 82.71±0.67  | 80.89±0.16  | 65.70±0.20  | - |
> | **Mowst-H2GCN**  | 83.39±0.43 | **83.02**±0.30  | **66.03**±0.16  | - |
> | **(gain)**  | (+0.68)  | (+2.13)  | (+0.33)  | (+1.05) |
>
> We observe that
> * H2GCN is effective in handling heterophilous graphs, as it achieves high accuracy among baselines.
> * Applying Mowst on top of H2GCN can further enable significant accuracy improvements (**+1.05** on average for the 3 heterophilous graphs).
> * We observe **consistent and significant accuracy boost** by Mowst on 4 different GNN architectures (the gains are especially large on GIN).
>
> We believe the above additional results, together with the original results in Table 1, concretely show that our **improvements are significant** rather than incremental. Please do not hesitate to let us know of any questions.
>
> Thanks!
>
> Authors

---

> > ### Comment · Reviewer_BGLX · 2023-11-21
> > **Upgrade my score based on the response**
> >
> > Dear Authors:
> >
> > Thanks so much for the detailed response. I go back to check the updated manuscript. The revised one looks much better. I'd like to update my score since most of my concerns are addressed very well.
> >
> > Minor issues:
> > It likely misses the exact kind of confidence function C that is used across the experimental section since it has different choices. It's better to indicate it out in the main content.
> >
> > Best regards

---

> > > ### Author Response · Authors · 2023-11-21
> > > **Thank you!**
> > >
> > > We are really happy to see that our response has addressed most of the reviewer's concerns. Thank you very much for the positive feedback and we appreciate your time!
> > >
> > > Best,
> > > Authors

---

### Author Response · Authors · 2023-11-18
**Looking forward to discussing with reviewers (Part 1)**

**Dear reviewers & AC**,

We greatly appreciate the valuable feedback from all reviewers. We have taken each and every comment very *seriously* and spent *tremendous* efforts in addressing them via “Authors’ Response”.

We cherish this opportunity to have an open-minded discussion. **We encourage the reviewers to take a look at our responses** so that we have sufficient time to clarify any remaining ambiguity by Nov 22, AOE.

We are actively preparing a paper revision and running suggested experiments. We will update the pdf very soon. We have seen that the current readings from all the new experiments have consistently validated & strengthened our claims (final numbers will be included in the revision). Therefore, we believe the pending experimental numbers will **not block** any discussion on our “Authors’ Response”.

## Brief summary of our responses (all based on original submission)

**Reviewer BGLX** understood our paper from the heterophilous graph perspective and asked for clarification on a number of statements in Introduction.

Our response has
* completed the picture that our model (based on original submission) goes **well beyond** heterophilous GNNs, both *theoretically* and *empirically*;
* clarified the **comprehensiveness of existing related works & baselines**, even from the perspective of heterophilous GNNs;
* clarified all statements in Introduction by referring to existing contents in the technical, related work and experiment sections.

Pending results
* We will add the H2GCN & Mowst-H2GCN results shortly. Early readings show
    * H2GCN performs well on the heterophilous benchmarks in Section 4;
    * Mowst-H2GCN further *boosts the accuracy* of H2GCN *significantly* on the heterophilous graphs.
--------------------

**Reviewer 3gAn** constructed a failure case of our model, asked for additional context for our design choice, and suggested additional experiments to strengthen our results.

Our response has
* explained from different angles why such a failure case can be **well-handled** by Mowst;
* provided justification on our current design compared with reasonable alternatives;
* shown that 4 of the existing baselines have **already implemented** various forms of residual connections.

Pending results
* We will include the Mowst-GIN results shortly. Early readings show that Mowst-GIN consistently achieves *significant accuracy improvements* compared with the GIN baseline.
* We will include additional results on alternative residual-connection baselines.

--------------------------

**Reviewer SJRD** asked about algorithmic details on the training and inference of Mowst, the mechanism to prevent over-contribution of the weak expert, the rationale behind choosing MLP and GNN as the experts, and additional details on computation complexity analysis.

Our response has
* clarified the reviewer's **misunderstanding** of our training & inference algorithms as well as the optimization process;
* explained why the random sampling step **aligns well** with our training & inference objectives;
* shown how the confidence-based gating **discourages MLP’s poor predictions** from being accepted, both *theoretically* & *empirically*;
* explained the motivation of using MLP as the weak expert, and the model-agnostic nature regarding the strong expert;
* illustrated the detailed steps to derive the computation complexity for different models.

Pending results
* We will include the execution time for MLP, GNN and Mowst. It well validates the theoretical complexity analysis and Mowst is *as efficient as* a standalone GNN baseline.

-------------------------

**Reviewer jetB** asked clarification questions on our theoretical analysis, discussed scalability of Mowst to more than 2 experts and requested more details on the “denoising” process. The reviewer also suggested related papers that can provide helpful context.

Our response has
* clearly clarified all questions on theoretical analysis, showing that all theoretical statements are **formal and rigorously proven** in Appendix B.
* explained various ways of generalizing Mowst to more than 2 experts (as **already discussed** in Section 2.7);
* illustrated the fundamental mechanism of our unique “**denoised fine-tuning**” behavior;
* acknowledged and compared the additional related works and **added to the paper revision** (*to be uploaded soon*).

-----------------------------
(Please see part 2)

---

> ### Author Response · Authors · 2023-11-18
> **Looking forward to discussing with reviewers (Part 2)**
>
> ---------------------------
>
> **Reviewer DN88** proposed a new set of experiments to provide additional justification on the choices of the weak & strong experts, and asked for dataset details.
>
> Our response has
> * clarified why the current MLP-GNN combination is a suitable choice for weak & strong experts under **a consistent model architecture**;
> * provided detailed description on dataset construction.
>
> Pending results
> * We will include additional results on weak GNN + strong GNN. Early readings justifies our intuition for selecting suitable weak & strong experts.
>
> ------------------------
> **Reviewer MV5e** discussed a promising future work direction and asked about correlation between data property and model performance.
>
> Our response has
> * provided additional thoughts on the big potential of extending our current Mowst design;
> * clarified different scenarios that affect the data-model correlation.
>
> Pending results
> * We will include profiling results on edge density and MLP performance.
>
> ------------------------
> **We are eager to have a fruitful discussion with all the reviewers and we look forward to any comments, questions and suggestions.**
>
> Thanks,
>
> Authors

---

### Author Response · Authors · 2023-11-20
**Summary of Revision 1**

Dear reviewers & AC,

We have uploaded a revision after integrating many suggestions from all the reviewers. The revision improves both the main text and the Appendix (uploaded in separate pdf files, where Appendix is under “supplementary material”).

Here is a brief summary of the changes:

**Presentation clarity**: Reviewers 3gAn, MV5e and DN88 acknowledged the high presentation quality of our original submission, while reviewers BGLX, SJRD and jetB pointed out some issues of ambiguity. We have thoroughly addressed such issues in the main text.

**Clarification on algorithms & theoretical analysis**: We have improved the descriptions throughout Section 2 to address
* Clarification questions on the training & inference algorithms (SJRD), and
* Clarification questions on theoretical statements (jetB).

**Additional experiments**: We have included additional experimental results in Appendix B (due to the 9-page space limit on main text). We have shown:
* Significant accuracy boost by applying Mowst on H2GCN (BGLX)
* Significant accuracy boost by applying Mowst on GIN (3gAn), where more results on GIN will be included in the next revision.

**Related works**: We have included suggested references from jetB, and clarified the heterophilous literature according to BGLX.

---------------------------

Main changes in the **main text**:

* Section 1: clarified the motivation (BGLX), target problem (jetB) and the key terms for describing our techniques (BGLX).
* Section 2.1: clarified the training and inference algorithms (SJRD).
* Section 2.3:
    * Restructured the paragraphs to make the conclusions more clear (BGLX, jetB)
    * Clarified the problem setup for the theoretical analysis (jetB)
    * Revised descriptions to make the analysis more readable; deferred Corollary 2.4.1 to Appendix C (jetB).
* Section 2.4: rephrased the “denoised fine-tuning” description to explain the process more clearly (jetB)
* Section 2.6: clarified definition of expressive power (jetB)
* Section 3:
    * Clarified the heterophilous GNN literature (BGLX)
    * Added suggested references (jetB)
* Section 4:
    * Clarified heterophilous GNN baselines (BGLX)
    * Clarified residual connections of baselines (3gAn)

---------------------

Main changes in **Appendix**:

* Appendix A.4: dataset description (DN88)
* Appendix B.1:
    * **Significant accuracy improvements on Mowst-H2GCN** (BGLX)
    * **Significant accuracy improvements on Mowst-GIN** (3gAn)
* Appendix C.3: formal definition of expressive power (jetB)
* Appendix G: (in addition to the design in Section 2.7) further thoughts on the many-expert generalization of Mowst (MV5e, jetB)

-----------------------

We look forward to hearing from the reviewers.

Thanks,

Authors

---

### Author Response · Authors · 2023-11-21
**Summary of Revision 2**

Dear reviewers & AC,

We have uploaded the second revision with additional experiments showing the significant and consistent accuracy gains of Mowst, and algorithmic details on the computation complexity calculation.

Here is a brief summary of the main changes:

* Appendix A.4: justification that our comparison with the baselines is fair, under the criteria of similar computation cost (3gAn).
* Appendix B.2: additional experiments on Mowst-GIN, and GCN & GIN with skip connections (where the GraphSAGE and H2GCN baselines already contain skip connection). See **Table 5** (3gAn)
    * GCN-skip does not help improve the performance of GCN.
    * GIN-skip improves accuracy of GIN on 3 benchmarks.
    * **Mowst-GIN achieves significant accuracy improvement over GIN on all graphs**.
    * **Mowst-GIN-skip further boosts accuracy of GIN-skip significantly on all graphs**.
* Appendix D: detailed calculation on computation complexity of various architectures, including: MLP, GNN, GNN with skip connection, Mowst (3gAn, SJRD). Results are consistent with our complexity analysis in Section 2.6, that
    * GNN is expensive to execute
    * Mowst is as efficient as the GNN expert alone
    * Skip connection introduces significant overhead in computation complexity (due to its operation on multi-hop neighbors)

Please do not hesitate to let us know of any questions.

Thanks,

Authors

---

### Author Response · Authors · 2023-11-23
**Summary of Revision 3**

Dear Reviewers & AC,

We have uploaded a new revision of the paper. The main changes are all in the Appendix:

* Appendix B.3: experimental results by replacing the MLP expert with a shallow GCN (DN88)
* Appendix E: illustration of how Mowst handles the two failure cases proposed by 3gAn
* Appendix F: explanation of potential alternative designs (3gAn)

Thanks,

Authors

---

### Meta-Review · Area_Chair_ij5c · 2023-12-10

**Metareview:**

This paper suggests to decouple self-features from structural signal found in graphs by employing a combination of a weak (MLP) and a strong (GNN) expert. To activate the weak or strong expert in the appropriate regions a confidence mechanism is further developed. The overall approach us dubbed "Mowst". Through experiments, Mowst is shown to improve performance of various GNNs used as backbone, effectively demonstrating that it can scale up their capacity. Theoretical results and a few ablation studies are also included.

Overall this paper is interesting because it does not only attempt to offer insights on the decoupling of the different types of signals but, additionally, proposes a MoE design with appropriate region activation mechanism. Ablation studies and theoretical results justify the majority of the decisions. Furthermore, the experiments are generally convincing.

This paper has been reviewed by 6 experts and the opinions are not unanimous. Several concerns have been raised but thankfully there have been a lot of discussions during the rebuttal period. Many concerns have been addressed and the authors have also included new experiments.

The key concern that has not been fully addressed is about analyzing deeper the limitations of MLP and GNN. The authors have stated that an MLP is a special case of a GNN and provided intuitions deriving from this fact, however it is clear that optimization dynamics do matter (this is also shown by their new ablation in appendix B.3), so still there is some confusion there.

There are a few other concerns that remain such as scaling to multiple experts of addressing failure cases (see section "why not lower score" below), however I think it is OK for future work to address those. Therefore, I think that overall this paper can be accepted as it is expected to stimulate further interesting research.

**Justification For Why Not Higher Score:**

Even after rebuttal, a few concerns by the reviewers remain. We still need to understand better the limitations of MLP and GNN. One reviewer also asked for an ablation with a strong and a weak MLP, which wasn't included in the revision (instead the weak/strong GNN combination was included).

**Justification For Why Not Lower Score:**

I find that Mowst proves the motivation for decoupling the self-features from structural signals and using MoE for graph learning. The paper also suggests a path to doing so. In my opinion that should be enough for accepting the paper even if the specific MoE architecture might not be the best (indeed, research evolves and it should be a good thing if this paper motivates some other researcher to build a multi-expert or generally more powerful MoE).

---

### Decision · Program_Chairs · 2024-01-16

Accept (poster)